# Interpolation-Based Conditioning of Flow Matching Models for Bioisosteric Ligand Design

**Yael Ziv[1,3], Martin Buttenschoen[1], Lukas Scheiblerger[3], Brian Marsden[2,3], Charlotte M. Deane[1]**

[1] Department of Statistics, University of Oxford, St Giles', OX1 3LB, Oxford, United Kingdom
[2] Nuffield Department of Medicine, University of Oxford, Old Road, OX3 7BN, Oxford, United Kingdom
[3] Centre for Medicines Discovery, Nuffield Department of Medicine, OX3 7FZ, Oxford, United Kingdom

deane@stats.ox.ac.uk

## Abstract

Fast, unconditional 3D generative models can now produce high-quality molecules, but adapting them for specific design tasks often requires costly retraining. To address this, we introduce two training-free, inference-time conditioning strategies, *Interpolate–Integrate* and *Replacement Guidance*, that provide control over E(3)-equivariant flow-matching models. Our methods generate bioisosteric 3D molecules by conditioning on seed ligands or fragment sets to preserve key determinants like shape and pharmacophore patterns, without requiring the original fragment atoms to be present. We demonstrate their effectiveness on three drug-relevant tasks: natural product ligand hopping, bioisosteric fragment merging, and pharmacophore merging.

## 1 Introduction

In drug discovery, the design of effective compounds is guided by information about the biomolecular target. This information can be direct, as in structure-based drug design (SBDD), where the target's 3D structure is known; or it can be indirect, as in ligand-based drug design (LBDD), where knowledge is inferred from molecules already known to bind the target. The difference can be illustrated with an analogy: SBDD is like having the blueprint for a lock, while LBDD is like trying to make a new key by only studying existing keys. While SBDD is powerful, the protein structure is often unknown, making LBDD a widely-used approach (Acharya et al.; Sharma et al., 2021; Palazzesi & Pozzan, 2022; Zhung et al., 2024). This work focuses on the challenges of LBDD. The central goal in LBDD is to preserve key interaction patterns (e.g., shape or pharmacophores, a specific 3D arrangement of interaction features) of a reference ligand while optimizing other properties like synthetic accessibility. This is applied in tasks like natural product ligand hopping, where complex molecules are redesigned into simpler analogues (Morrison & Hergenrother, 2014). Another application is fragment-based design, where information from multiple small, weakly-binding fragments is used to design a single, more potent molecule (Erlanson et al., 2004). A key strategy is bioisosteric fragment merging, which combines the interaction patterns from multiple fragments into a single new molecule (Wills et al., 2024). In this approach, the final product may discard the exact fragment atoms but must preserve their essential shape and pharmacophores.

Existing conditional generators that operate on ligand information alone involve specific trade-offs. For example, REINVENT (Olivecrona et al., 2017) is not 3D-native, instead optimizing SMILES with reinforcement learning toward user-defined objectives. Others, such as SQUID and Shape-Mol, generate molecules in 3D but condition only on a reference molecule's overall shape, not on more granular pharmacophore patterns (Adams & Coley, 2023; Chen et al., 2023). The state-of-the-art SHEPHERD model advances this by conditioning on a combination of shape, electrostatic potential (ESP), and pharmacophore grids (Adams et al., 2025). However, its application to multi-fragment conditioning requires a manual step of constructing the aggregate ESP and interaction profile of the fragments. Looking ahead, large-scale structural efforts such as the UK *OpenBind*

consortium—aiming to produce >500,000 protein–ligand complexes—will markedly expand fragment and pose libraries, highlighting the need for tools that can operate directly in this data-rich regime (OpenBind Consortium, 2025).

To address this gap, we adapt a pre-trained, unconditional models for controllable, goal-directed tasks. Recent advances in fast, unconditional SE(3)/E(3)-equivariant models provide realistic 3D molecule generators that support large-scale sampling (Buttenschoen et al., 2025). For molecule generation, there are already examples of this strategy (also called goal-directed or controllable generation), e.g. (Hoogeboom et al., 2022; Morehead & Cheng, 2024; Le et al., 2024; Ziv et al., 2025). By repurposing these powerful foundational models, we can avoid costly retraining and create flexible, on-the-fly conditioning mechanisms.

In this work, we introduce two modular, training-free conditioning strategies for a state-of-the-art flow-matching model, enabling precise, ligand-based bioisosteric design at inference time. Our main contributions are:

- Two complementary, inference-time conditioning methods. *Interpolate–Integrate* designed for controlled similarity to a seed structure and *Replacement Guidance* for bioisosteric merging with a user-controlled degree of relaxation. *Interpolate–Integrate* is a global similarity method without being constrained by specific local atoms, while *Replacement Guidance* is intended for "hard" conditioning where strict local preservation is required. A key feature of our framework is the ability to automatically condition on a set of multiple fragments.

- Both methods are training-free and modular, operating on a pre-trained flow-matching model. This efficient approach avoids costly retraining, making the framework adaptable for various ligand-only design scenarios where protein structures are unavailable.

- We demonstrate the practical effectiveness of our framework across three challenging drug-relevant tasks—natural product ligand hopping, bioisosteric fragment merging, and pharmacophore merging—producing high yields of valid and synthetically accessible molecules that outperform specialized, state-of-the-art baselines.

Our code is available at: `https://github.com/oxpig/cond-semla`

## 2 RELATED WORK

### 2.1 UNCONDITIONAL MOLECULAR GENERATION

The Equivariant Diffusion Model (EDM) established a foundational baseline for 3D molecular generation by using an E(3)-equivariant network to jointly denoise atom types and coordinates (Hoogeboom et al., 2022). Since EDM, several unconditional 3D generators have advanced capacity and efficiency, including EQGAT-diff (Le et al., 2024), FlowMol (Dunn & Koes, 2024), GCDM (Morehead & Cheng, 2024), GeoLDM (Xu et al., 2023), and SemlaFlow (Irwin et al., 2024). A recent head-to-head evaluation (Buttenschoen et al., 2025) on GEOM-Drugs reports that SemlaFlow achieves the best overall performance, with an 87% success rate for generating valid, unique, and novel molecules without post-processing (92.4% with post-processing), and is the fastest method compared.

### 2.2 TRAIN-TIME JOINT CONDITIONAL MODELS (3D-NATIVE AND GOAL-DIRECTED)

Ligand-based conditional design has long used 1D/2D generators guided by 3D-derived constraints (e.g., shape/pharmacophores), which still require post hoc conformer generation (Skalic et al., 2019; Imrie et al., 2021). In contrast, 3D conditional models place atoms in space directly. Both SQUID (Adams & Coley, 2023) and SHAPEMOL (Chen et al., 2023) condition generation on a target 3D shape, typically represented as a point cloud. SQUID couples an equivariant shape encoder with fragment-wise autoregressive assembly, and ShapeMol uses an equivariant shape encoder with a conditional diffusion decoder. Closest to our task, ShEPhERD learns a joint diffusion over molecules *and* interaction grids (shape, ESP, pharmacophores) on 1.6 M MOSES (Polykovskiy et al., 2020) structures (Adams et al., 2025). Despite its capabilities, ShEPhERD must be retrained for every new conditioning channel and still requires manual grid construction. As a non-joint alternative, REIN-

VENT is a non-3D-native method that optimizes a SMILES policy with reinforcement learning, requiring post hoc conformer generation (Olivecrona et al., 2017).

## 2.3 BIOISOSTERIC DESIGN AND FRAGMENT-LEVEL OPTIMIZATION.

A complementary line of research tackles bioisosteric replacement from a property–optimization perspective. Many molecular optimization methods formulate replacement as an editing task on 2D molecular representations, such as SMILES (sequence) or molecular graphs (graph-editing) (He et al., 2021; 2022; Yang et al., 2023; Jin et al., 2019; 2020). Within this category, models such as Chen et al. (2021) and DeepBioisostere (Kim et al., 2024) learn fragment substitutions that preserve biological function while explicitly optimizing scalar physicochemical properties (e.g., logP, TPSA, potency). All of these approaches require task-specific retraining to capture the desired property or substitution patterns.

## 2.4 INFERENCE-TIME CONDITIONING AND EDITING

Inference-time conditioning allows users to steer a pretrained generator without additional training. SBDD controls parallel diffusion-based inpainting and editing methods developed for images and point clouds (Lugmayr et al., 2022; Meng et al., 2022; Zeng et al., 2022). In SBDD, this is often applied to generate molecules directly in the protein pocket; examples include DiffSBDD (Schneuing et al., 2024) that apply inpainting and context injection to a pocket-conditioned SE(3) diffuser, while PILOT (Cremer et al., 2024) adds importance-sampling guidance on top of a large-scale pretrained model to meet multiple objectives. SILVR (Runcie & Mey, 2023) refines latent trajectories to satisfy pocket or fragment constraints, and MolSnapper (Ziv et al., 2025) projects explicit pharmacophore points (e.g., donors, acceptors) into the diffusion process. FLOWR (Cremer et al., 2025) uses equivariant flow matching as a faster alternative to diffusion for pocket-conditioned ligand generation, and additionally introduces a multi-task model trained on inpainting tasks for fragment-based sampling. Fragment merging/linking provides a closely related setting: DiffLinker (Igashov et al., 2024) treats the linker as a 3D-conditioned diffusion sub-problem and reports higher validity and lower RMSD than inpainting; LinkerNet co-designs linker geometry and poses jointly (Guan et al., 2023). To speed up inference, TurboHopp replaces the diffusion backbone with a consistency model, enabling pocket-conditioned scaffold hopping (Yoo et al., 2024).

## 2.5 POSITIONING OF THIS WORK.

Our work introduces *Interpolate–Integrate* and *Replacement Guidance*, two **ligand-only, inference-time** conditioning methods for **bioisosteric design**, where the goal is to generate a ligand that preserves the key binding determinants of a reference molecule, such as overall shape and pharmacophoric interactions, without need to retain the original atoms. As a training-free framework on a fast flow-matching backbone, our approach avoids the costly retraining required by many conditional models. It provides modular, fine-grained control directly at sampling while preserving the high inference speed of the base model. This allows for the efficient generation of chemically diverse molecules that maintain key pharmacophoric and shape-based interactions.

# 3 METHODS: CONDITIONING FLOW MATCHING FOR MOLECULAR GENERATION

We propose two conditioning mechanisms for fast, controlled 3D molecular generation on top of *SemlaFlow* (Irwin et al., 2024), an E(3)-equivariant flow-matching model that jointly generates coordinates and discrete chemistry. We first summarize conditional flow matching (CFM) (Lipman et al., 2022; Campbell et al., 2024) and SemlaFlow's path and solver in particular. Finally, we formulate the interpolate-integrate and replacement guidance, our two conditioning methods.

## 3.1 BACKGROUND

### 3.1.1 CONDITIONAL FLOW MATCHING (CFM)

Let $z = (x, a, b, c)$ denote a molecule with atomic coordinates $x \in \mathbb{R}^{N \times 3}$, atom types $a \in \mathcal{A}^N$, bond types $b \in \mathcal{B}^{N \times N}$, and formal charges $c \in \mathcal{C}^N$. CFM trains a neural vector field $v_\theta(t, z)$ whose flow $\phi_t$ pushes a simple prior $p_0$ to a data distribution $p_1$ by regressing to a target *conditional* vector field $u_t(z \mid z_1)$ defined along a per-example probability path $p_t(z \mid z_1)$ connecting noise to a specific data point $z_1$. The learning objective is the conditional regression

$$\mathcal{L}_{\mathrm{CFM}}(\theta) = \mathbb{E}_{\substack{t \sim \mathcal{U}[0,1], \\ z_1 \sim p_{\mathrm{data}}, \, z_t \sim p_t(\cdot | z_1)}} \left\| v_\theta(t, z_t) - u_t(z_t \mid z_1) \right\|^2,$$

which yields the same optimum as regressing the (intractable) marginal vector field on the marginal path $p_t$ obtained by integrating out $z_1$. This conditional formulation is simulation-free and admits many paths, including diffusion and optimal-transport (OT) Gaussian paths.

### 3.1.2 SEMLAFLOW INSTANTIATION FOR MOLECULES

**Joint variables and paths.** SemlaFlow models the joint $p(z)$ with: (i) a continuous OT-like interpolation for coordinates with small Gaussian jitter, and (ii) discrete conditional paths for atom and bond types; charges are predicted but do not participate in the flow. Concretely, at training time

$$x_t \sim \mathcal{N}\big(t\, x_1 + (1-t)\, x_0, \, \sigma^2 I\big), \qquad t \sim \mathrm{Beta}(\alpha, \beta),$$

$$a_t \sim \mathrm{Cat}\Big((1-t)\, \tfrac{1}{|\mathcal{A}|}\, \mathbf{1}_\mathcal{A} \,+\, t\, \delta_{a_1}\Big), \qquad b_t \sim \mathrm{Cat}\Big((1-t)\, \tfrac{1}{|\mathcal{B}|}\, \mathbf{1}_\mathcal{B} \,+\, t\, \delta_{b_1}\Big),$$

after first aligning $(x_0, x_1)$ by an equivariant OT alignment $f_\pi$ (permutation + rotation/reflection) (Klein & Noé, 2024; Song et al., 2023) that minimizes squared transport cost. SemlaFlow trains a predictor $p_\theta(z_1 \mid t, z_t)$ of clean data (rather than the vector field directly), and uses it to reconstruct the vector field needed by the ODE sampler. The sampler is Euler with 100 steps and log-spaced. Here $\mathcal{A}$ and $\mathcal{B}$ are the finite sets of atom and bond types, respectively. $\mathbf{1}_\mathcal{A} \in \mathbb{R}^{|\mathcal{A}|}$ and $\mathbf{1}_\mathcal{B} \in \mathbb{R}^{|\mathcal{B}|}$ denote all-ones vectors (so $\tfrac{1}{|\mathcal{A}|} \mathbf{1}_\mathcal{A}$ and $\tfrac{1}{|\mathcal{B}|} \mathbf{1}_\mathcal{B}$ are the uniform distributions on $\mathcal{A}$ and $\mathcal{B}$, respectively), and $\delta$ denotes the Kronecker delta. $\delta_{a_1}, \delta_{b_1}$ denote one-hot point masses at $a_1$ and $b_1$.

**SemlaFlow sampling ODE.** With the predictor $p_\theta(z_1 \mid t, z_t)$, denote $\tilde{z}_1 = \mathbb{E}[z_1 \mid t, z_t]$. For coordinates the conditional vector field implied by the linear path is

$$u_t(x_t \mid x_1) = \frac{\tilde{x}_1 - x_t}{1 - t}, \qquad \text{Euler step:} \quad x_{t+\Delta t} = x_t + \frac{\Delta t}{1 - t}\big(\tilde{x}_1 - x_t\big).$$

Following *SemlaFlow* and the *Discrete Flow Models* (DFM) framework—categorical interpolation at train time and Continuous Time Markov Chain (CTMC) sampling at test time—the discrete variables are evolved via

$$a_{t+\Delta t} \sim \mathrm{Cat}\big(\delta_{a_t} + \Delta t\, R_t(a_t, \cdot)\big), \qquad b_{t+\Delta t} \sim \mathrm{Cat}\big(\delta_{b_t} + \Delta t\, R_t(b_t, \cdot)\big). \tag{1}$$

$$R_t(i, j) = \mathbb{E}_{a_1 \sim p_\theta(a_1 | t, z_t)}\Big[\frac{\delta_{j,a_1}\,(1 - \delta_{i,a_1})}{1 - t}\Big] \quad \text{for atoms, and analogously with } b_1 \text{ for bonds.} \tag{2}$$

Intuitively, mass flows only toward the denoised label, with a strength that increases as $t \to 1$.

## 3.2 OUR CONDITIONING MECHANISMS

We now introduce two inference-time conditioning procedures. Both build on SemlaFlow's solver.

**Notation.**

- Fragment mask $M \in \{0, 1\}^N$ indicates atoms belonging to seeding fragments; $\bar{M} = 1 - M$.
- For any tensor $y$ indexed by atoms (or atom pairs), write $y[M]$ and $y[\bar{M}]$ for the masked and unmasked components respectively, and $\mathrm{Replace}_M(y; y^\star)$ for replacing $y[M] \leftarrow y^\star[M]$.
- Let $t_0 = 0 < t_1 < \cdots < t_K = 1$ be the integration grid (log-spaced as in SemlaFlow).

### 3.2.1 INTERPOLATE-INTEGRATE (SEED-GUIDED RESAMPLING)

**Goal.** Given a reference input $z_1 \sim p_{\text{data}}$, produce a sample $\tilde{z}_1$ that is controllably similar while allowing global edits.

**Key idea.** Start in the middle of the probability path at time $\tau \in [0,1]$: first interpolate from the seed toward noise to obtain $z_\tau$, then integrate the ODE forward from $t = \tau$ to 1. Large $\tau$ yields conservative edits; small $\tau$ yields diverse rewrites. Optional noise injection at $t = \tau$ decorrelates geometry without discarding the seed entirely.

**Construction.** Let $z_0 \sim p_0$ and write the SemlaFlow path for $t = \tau$

$$x_\tau = \tau\, x_1 + (1 - \tau)\, x_0 + \varepsilon, \qquad \varepsilon \sim \mathcal{N}(0, \sigma_\tau^2 I) \text{ (optional extra jitter)},$$

$$a_\tau \sim \mathrm{Cat}\Big( \tau\, \delta_{a_1} + (1 - \tau) \tfrac{1}{|\mathcal{A}|} \mathbf{1}_\mathcal{A} \Big), \qquad b_\tau \sim \mathrm{Cat}\Big( \tau\, \delta_{b_1} + (1 - \tau) \tfrac{1}{|\mathcal{B}|} \mathbf{1}_\mathcal{B} \Big).$$

From $(t, z_t) = (\tau, z_\tau)$, integrate the SemlaFlow ODE using $\tilde{z}_1 = \mathbb{E}[z_1 \mid t, z_t]$ at each step:

$$x_{t_{k+1}} = x_{t_k} + \frac{\Delta t_k}{1 - t_k} \Big( \tilde{x}_1(t_k, z_{t_k}) - x_{t_k} \Big), \qquad k : t_k \in [\tau, 1).$$

*Discrete update.* Apply the CTMC Euler step (Eqs. equation 1–equation 2):

$$a_{t_{k+1}} \sim \mathrm{Cat}\Big( \delta_{a_{t_k}} + \Delta t_k\, R_{t_k}(a_{t_k}, \cdot) \Big), \qquad b_{t_{k+1}} \sim \mathrm{Cat}\Big( \delta_{b_{t_k}} + \Delta t_k\, R_{t_k}(b_{t_k}, \cdot) \Big), \quad k : t_k \in [\tau, 1).$$

**Effect.** For $\tau \to 1$, $\tilde{z}_1$ matches $z_1$ up to small edits; for $\tau \to 0$, sampling reduces to unconditional generation. This 'mid-path restart' is conceptually related to editing techniques in diffusion models that manipulate the reverse sampling process from an intermediate timestep Zeng et al. (2022). However, our application to the deterministic ODE trajectory of a flow-matching model is distinct. To our knowledge, this is the first such interpolate-integrate conditioning scheme for flow-matching model, offering a computationally lightweight yet powerful method for controlled generation. SemlaFlow provides an OT-like linear path and the ODE needed to do so efficiently.

### 3.2.2 REPLACEMENT GUIDANCE (BIOISOSTERIC FRAGMENT MERGING)

**Goal.** Use a set of fragments to generate ligands that do not necessarily contain the exact fragment atoms but still preserve key binding interactions. As a complementary approach to the global, soft conditioning of *Interpolate-Integrate*, *Replacement Guidance* provides hard, localized control over a defined set of fragments, allowing for the generation of unconstrained parts of the molecule to merge or complete them.

**Key idea.** During ODE integration, after each Euler update, we replace the masked fragment region with the original, unperturbed fragment values. This "hard anchoring" maintains exact spatial and chemical identity of fragments throughout the trajectory, while leaving the complement $\bar{M}$ free to evolve. *Relaxation of the constraint is user-controlled:* starting from a chosen time $t_{\text{relax}} \in [0,1]$ (with the convention that the final step is always free), replacement is disabled for all subsequent steps. More generally, one may specify a relaxation set $\mathcal{T}_{\text{free}} \subseteq \{t_k\}_{k=1}^{K}$ of steps at which replacement is skipped; the default is $\mathcal{T}_{\text{free}} = \{t_K\}$.

**Masked ODE with replacement.** Let $z_{\text{frag}}$ denote the original fragments to preserve. For each step $t \to t + \Delta t$ with $t + \Delta t \notin \mathcal{T}_{\text{free}}$ (e.g., $t + \Delta t < t_{\text{relax}}$),

$$\text{(i) Euler proposal:} \quad x_{t+\Delta t}^{\text{prop}} = x_t + \frac{\Delta t}{1 - t} \Big( \tilde{x}_1(t, z_t) - x_t \Big),$$

$$\text{(ii) Replacement:} \quad x_{t+\Delta t} = \mathrm{Replace}_M \big( x_{t+\Delta t}^{\text{prop}}; x_{\text{frag}} \big),$$

$$\text{and analogously} \quad a_{t+\Delta t} = \mathrm{Replace}_M \big( a_{t+\Delta t}^{\text{prop}}; a_{\text{frag}} \big), b_{t+\Delta t} = \mathrm{Replace}_{M \otimes M} \big( b_{t+\Delta t}^{\text{prop}}; b_{\text{frag}} \big),$$

This yields a projected flow that exactly preserves the fragments up to relaxation time(s) while giving the model freedom to synthesize linkers and local substitutions thereafter, capturing the essence of bioisosteric merges. The novel components of this method are the introduction of the replacement operator and the user-controlled relaxation policy, which together adapt the pre-trained flow for bioisosteric merging.

### 3.2.3 INFERENCE RUNTIME.

All conditioning operates strictly at inference and preserves the original 100-step Euler integration schedule from *SemlaFlow*. (i) *Interpolate–Integrate* introduces a one-time $O(N)$ seed interpolation prior to integration. (ii) *Replacement Guidance* adds only a lightweight $O(N)$ masked tensor update per step, negligible relative to a model forward pass. Overall, both methods maintain inference efficiency comparable to the unconditioned baseline.

## 4 EXPERIMENTS

We demonstrate the effectiveness of our conditioning strategies on three challenging, drug-relevant tasks: natural product ligand hopping, bioisosteric fragment merging, and pharmacophore merging. Figure 4.1 provides a visual overview of the conditioning setups and showcases representative top-ranked molecules for each task, selected by a medicinal chemist after filtering. We evaluate all generated molecules on a suite of metrics designed to assess their validity, synthetic accessibility, and 3D similarity to reference compounds.

### 4.1 EVALUATION METRICS

We evaluate generated molecules on synthetic accessibility (SA) using the score from Ertl & Schuffenhauer (2009), where lower values are better. For validity, molecules must be parsable by RDKit and pass PoseBusters plausibility checks (Buttenschoen et al., 2023). We combine these and report the percentage of initial samples that pass both validity and SA score filters. We also compute 3D similarity to a reference molecule by comparing surface, electrostatic, and pharmacophore features, following the protocol of Adams et al. (2025). Finally, we calculate an AutoDock Vina docking score (Eberhardt et al., 2021), using energy minimization for conditionally-generated poses to ensure robustness (Weller & Rohs, 2024).

### 4.2 NATURAL PRODUCT LIGAND HOPPING

Natural products are complex and diverse substances produced by organisms such as plants, microbes, and marine life. Their rich structural variety and biological relevance make them valuable starting points for small molecule drug discovery (Morrison & Hergenrother, 2014). However, their inherent complexity and limited synthetic tractability often hinder the development of analogues and optimization for drug-like properties (Lee & Schneider, 2001). Ligand hopping offers a strategy for generating novel compounds that retain the biological activity of known natural products while exploring more synthetically accessible chemical space.

We follow the SHEPHERD framework (Adams et al., 2025) using three structurally diverse natural products (NP1, NP2, NP3) from the COCONUT database (Sorokina et al., 2021), aiming to generate analogues with SA scores below 4.5. We generated 2,500 candidate analogues with both *Interpolate–Integrate* and *Replacement Guidance*, conditioning on key pharmacophoric features (Hydrogen Bond Donors, Hydrogen Bond Acceptors, and Aromatic Rings) extracted from each reference NP using the RDKit FeatureFactory. We compare our results against two baselines: MolSnapper, in a ligand-only mode, conditioned on the NP's pharmacophore points, and SHEPHERD, conditioned on its ESP and pharmacophore profiles. Further details on the generation parameters and post-processing are available in Appendix E.

Our results (Table 1) demonstrate that our two conditioning strategies have distinct strengths, making them suitable for different design goals. *Replacement-guidance* consistently generates molecules with the most favorable SA scores across all three targets. In contrast, *Interpolate-integrate* is effective for high-fidelity generation. While its yield of SA-compliant molecules is lower in this task, the candidates it produces tend to show high similarity to the reference seed in both ESP and pharmacophore space. Figure 2 provides a visual summary of these performance trade-offs, showing the relationship between surface and pharmacophore similarity. As the figure highlights, *Interpolate–Integrate* generates analogues with high 3D similarity to the reference. This focus on conservative edits, reflected in its lower overall diversity, suggests its utility as a tool for scenarios where preserving precise interaction patterns is the primary objective. The MOLSNAPPER baseline struggled

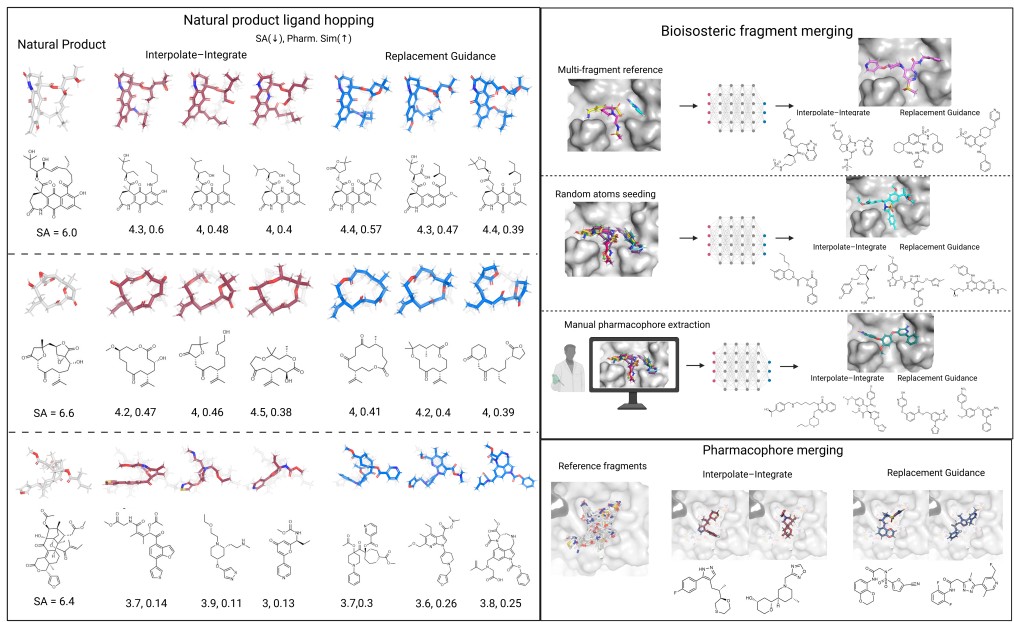

Figure 1: Overview of the three drug-relevant tasks, with top-ranked molecules selected by a medicinal chemist from our conditioning strategies *Interpolate–Integrate* and *Replacement Guidance*. **Left:** The natural product ligand hopping task. The goal is to generate analogues withm-lower SA scores while preserving the key features of the original natural product (leftmost column). **Right (Top):** The bioisosteric fragment merging task is demonstrated on EV-D68 3C protease using three different conditioning setups (multi-fragment, random atom seeding, and manual extraction) to generate a single coherent molecule from multiple inputs. All molecules shown are filtered for SA $< 4.0$. **Right (Bottom):** The pharmacophore merging task on SARS-CoV-2 $M_{pro}$, involves generating a novel ligand that combines the interaction patterns from a diverse set of known fragments. All molecules shown are filtered for SA $< 4.5$. Created in https://BioRender.com.

Table 1: Performance summary for the NP hopping tasks. The table reports overall PoseBusters validity, the average SA score, the percentage of initial samples passing both validity and an SA score $\leq 4.5$ filter, and top-10 similarity scores (ESP and pharmacophore) are calculated only on these successful candidates.

| NP | Method | Valid ↑ | SA Score ↓ | Valid (SA<4.5) ↑ | ESP sim (top 10) ↑ | Pharma sim (top 10) ↑ |
|---|---|---|---|---|---|---|
| NP1 | MolSnapper | 26.24% | $6.43 \pm 0.78$ | 0.32% | $0.51 \pm 0.02$ | $0.16 \pm 0.02$ |
| | ShEPhERD | 57.3% | $4.75 \pm 0.89$ | 24.7% | $0.63 \pm 0.02$ | $0.34 \pm 0.03$ |
| | Interpolate–integrate | **71.5%** | $4.67 \pm 1.02$ | 35.6% | $\mathbf{0.81 \pm 0.02}$ | $0.49 \pm 0.06$ |
| | Replacement–guidance | 60.5% | $\mathbf{3.74 \pm 1.01}$ | **50.2%** | $\mathbf{0.81 \pm 0.02}$ | $\mathbf{0.52 \pm 0.06}$ |
| NP2 | MolSnapper | 19.88% | $6.84 \pm 0.79$ | 0.16% | $0.63 \pm 0.02$ | $0.19 \pm 0.01$ |
| | ShEPhERD | 51.0% | $5.09 \pm 1.2$ | 17.4% | $0.79 \pm 0.01$ | $0.34 \pm 0.02$ |
| | Interpolate–integrate | **93.6%** | $5.41 \pm 1.03$ | 16.0% | $\mathbf{0.86 \pm 0.01}$ | $\mathbf{0.46 \pm 0.02}$ |
| | Replacement–guidance | 87.9% | $\mathbf{4.43 \pm 1.31}$ | **48.8%** | $0.85 \pm 0.01$ | $0.44 \pm 0.04$ |
| NP3 | MolSnapper | 14.20% | $6.32 \pm 0.85$ | 0.32% | $0.46 \pm 0.02$ | $0.14 \pm 0.01$ |
| | ShEPhERD | 44.1% | $4.69 \pm 0.96$ | 20.2% | $0.60 \pm 0.01$ | $\mathbf{0.30 \pm 0.02}$ |
| | Interpolate–integrate | 62.7% | $5.92 \pm 1.03$ | 5.2% | $0.56 \pm 0.05$ | $0.20 \pm 0.03$ |
| | Replacement–guidance | **64.4%** | $\mathbf{4.07 \pm 1.27}$ | **42.1%** | $\mathbf{0.65 \pm 0.03}$ | $0.29 \pm 0.02$ |

on this task; the mechanism of conditioning on discrete points proved overly restrictive for dense pharmacophore profiles that had not been manually filtered (see Appendix E).

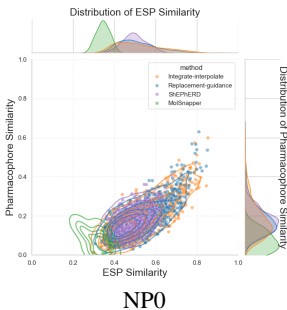 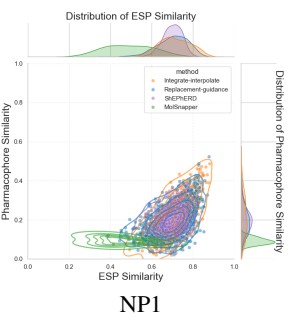 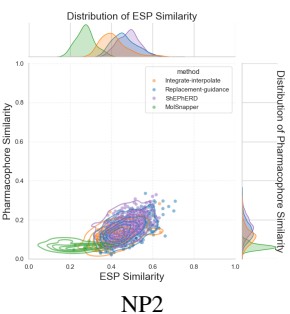

NP0                        NP1                        NP2

Figure 2: ESP similarity vs. pharmacophore similarity for generated analogues (filtered by SA < 4.5) across the three natural product tasks.

### 4.3 BIOISOSTERIC FRAGMENT MERGING.

Fragment-based screening identifies small molecules that bind to distinct regions of a target protein. Traditionally, these fragment hits are grown or linked to form larger ligands that retain the original fragments explicitly. However, recent work by (Wills et al., 2024) introduced a bioisosteric merging strategy, which aims to generate ligands that do not necessarily contain the exact fragment atoms while preserving key binding interactions. This approach enables greater chemical diversity while maintaining pharmacophoric fidelity.

Building on this framework, we perform bioisosteric merging on the EV-D68 3C protease system, using data from the Fragalysis platform[1]. Our input consists of 13 experimentally resolved fragment-bound structures, previously curated for the SHEPHERD study (Adams et al., 2025). We evaluate our methods across the following three conditioning setups (shown in Figure 4.1), which are designed to assess performance in both automated and expert-driven scenarios: **Multi-fragment Reference (Automated)**, where a composite input is formed by combining three known fragment-bound ligands; **Random Atom Seeding (Automated):** A sparse set of pharmacophore features is randomly sampled from the full fragment ensemble. This provides an automated way to leverage collective information from the raw fragment data without the need for manual expert curation; and **Full Interaction Profile (Manually Curated)**, a comprehensive 27-feature set curated for the SHEPHERD study that serves as an ideal but labor-intensive baseline (Adams et al., 2025).

Our goal is to assess how effectively the automated conditioning strategies can approach the performance of the manually curated profile, which represents a key challenge for real-world application.

Following the SHEPHERD study protocol, we generated 1,000 candidates per setup and applied a similar post-processing pipeline (see Appendix F.1 for full details). Since the input fragment poses contain implicit information about the target's binding pocket, a primary goal was to assess in-pocket fit. To do this, we filtered candidates for PoseBusters validity and a favorable SA score (SA $\leq$ 4.0, as used in SHEPHERD study). Passing candidates were then assessed using `AutoDock Vina`. We compare against protein-aware and protein-blind versions of MolSnapper, DiffSBDD, and SHEPHERD. A direct comparison across all methods is only possible on the manually curated 'Full Interaction Profile', as the baseline methods require a single, well-defined conditioning profile. Detailed setups for all methods and an analysis of unfiltered results can be found in Appendix F.3.

A comparison of the three conditioning setups in Table 2 reveals the trade-offs between automated and manually curated approaches. While the manually curated *'Full Interaction Profile'* was most effective at generating candidates with the highest predicted binding affinity (top-10 Vina score of -6.62), *'Multi-fragment reference'* setup produced the highest overall rate of valid candidates with SA¡4 (25.0%). The *'Random atom seeding'*, representing a more challenging scenario, resulted in a lower yield but still produced potent candidates (top-10 Vina score of -5.78). This highlights the robustness of our framework, demonstrating its ability to generate high-quality candidates across a range of conditioning scenarios from fully automated to expert-curated ('Full Interaction Profile').

---

[1]https://fragalysis.diamond.ac.uk/viewer/react/landing

Table 2: Performance summary for the EV-D68 3C bioisosteric merging task across three conditioning setups. The table reports overall PoseBusters validity, the average SA score, the percentage of initial samples passing both validity and an SA score $< 4$ filter, the fraction of successful molecules with zero clashes (No clash), and the top-10 Vina docking score.

| Conditioning setup | Method | Valid ↑ | SA Score ↓ | Valid (SA<4) ↑ | No clash ↑ | Vina top10 ↓ |
|---|---|---|---|---|---|---|
| Multi-fragment ref. | Interpolate–integrate | **59.3%** | $4.65 \pm 0.96$ | 14.8% | 12.1% | $-5.30 \pm 0.49$ |
| | Replacement–guidance | 36.3% | $3.57 \pm 0.85$ | **25.0%** | **19.3%** | $-5.59 \pm 0.21$ |
| Random atom seeding | Interpolate–integrate | 41.0% | $5.18 \pm 1.17$ | 6.8% | 5.3% | $-4.73 \pm 0.24$ |
| | Replacement–guidance | 28.1% | $3.84 \pm 0.81$ | 17.0% | 13.5% | $-5.78 \pm 0.26$ |
| Full interaction profile | Interpolate–integrate | 38.1% | $4.83 \pm 1.17$ | 8.7% | 7.0% | $-4.84 \pm 0.58$ |
| | Replacement–guidance | 31.9% | $\mathbf{3.47 \pm 0.70}$ | 23.5% | 18.7% | $\mathbf{-6.62 \pm 0.42}$ |

Table 3: Performance summary for the EV-D68 3C bioisosteric merging task (ShEPhERD profile setup). The table reports overall PoseBusters validity, the average SA score, the percentage of initial samples passing both validity and an SA score $< 4$ filter, the fraction of successful molecules with zero clashes (No clash), and the top-10 Vina docking score.

| Method | Valid ↑ | SA Score ↓ | Valid (SA<4) ↑ | No clash ↑ | Pharma sim top10 ↑ | Vina top10 ↓ |
|---|---|---|---|---|---|---|
| MolSnapper (ligand-based) | 17.6% | $7.04 \pm 0.54$ | 0% | — | — | — |
| MolSnapper (clash-rate = 0.1) | 14.5% | $6.91 \pm 0.53$ | 0% | — | — | — |
| DiffSBDD (largest frag) | **43.1%** | $6.91 \pm 0.53$ | 0% | — | — | — |
| ShEPhERD | 33.8% | $4.45 \pm 0.99$ | 10.8% | 10.1% | $\mathbf{0.27 \pm 0.02}$ | $-6.16 \pm 0.50$ |
| Interpolate–integrate | 39.5% | $4.96 \pm 1.17$ | 8.7% | 7.0% | $0.18 \pm 0.02$ | $-4.84 \pm 0.58$ |
| Replacement–guidance | 31.9% | $\mathbf{3.47 \pm 0.70}$ | **23.5%** | **18.7%** | $0.22 \pm 0.02$ | $\mathbf{-6.62 \pm 0.42}$ |

The results highlight that the conditioning mechanisms of DiffSBDD and MolSnapper—which are designed to strictly satisfy a set of discrete constraints—are less suited for this flexible bioisosteric merging task. Satisfying constraints while maintaining drug-likeness proved challenging: neither baseline produced candidates passing the combined validity, connectivity, and SA score ($< 4$) filter. Further analysis is provided in Appendix F. Our results demonstrate that *Replacement Guidance* achieves performance competitive with the state-of-the-art SHEPHERD model on the bioisosteric merging task. While SHEPHERD achieves higher pharmacophore similarity (0.27 vs. 0.22), our method produces candidates with superior synthetic accessibility (3.47 vs. 4.45) and a comparable top-10 Vina score ($-6.62$ vs. $-6.16$).

## 4.4 PHARMACOPHORE-MERGING ON SARS-COV-2 M$_{\text{PRO}}$

Fragment screening on the SARS-CoV-2 main protease (M$_{\text{pro}}$) provides a real-world test case. The Diamond/XChem campaign rapidly yielded a dense constellation of binding motifs, explicitly encouraging the merging of these fragment hits into more potent ligands[2]. From public Fragalysis data, we curated a conditioning set of 81 small fragments and a baseline set of 1,426 known binders, with the goal of merging pharmacophoric motifs from the fragments into novel ligands.

We generated 2,500 candidate molecules per method, using the 'random atom seeding' setup (Section 4.3). To identify promising candidates, we filtered all molecules for PoseBusters validity and a favorable SA score before evaluating them against baseline binders with AutoDock Vina (see Appendix G for the detailed setup). While we acknowledge the limitations of this high-throughput pipeline, such as docking to a single rigid receptor, it serves as a standard computational funnel to identify a manageable set of promising candidates from a large generative run.

The results, summarized in Table 4, demonstrate that our methods can successfully merge pharmacophoric motifs from a large set of small fragments into novel, synthetically feasible ligands. In addition, we performed an interaction-recovery analysis to verify that the generated molecules reproduce the key protein–ligand interaction patterns encoded in the conditioning data. Using ProLIF

---

[2] https://fragalysis.diamond.ac.uk/viewer/react/preview/target/CoV-Mpro/tas/lb32627-272

fingerprints (Bouysset & Fiorucci, 2021), we found that both *Interpolate–Integrate* and *Replacement Guidance* recover the full set of interaction types observed in the 81 conditioning fragments, even after applying PoseBusters filtering and docking (see Appendix G for details). Together, these results indicate that our generated molecules not only satisfy geometric and physicochemical criteria but also explore the desired interaction space, making them competitive starting points for a drug discovery campaign.

Table 4: Performance summary for the SARS-CoV-2 Mpro pharmacophore-merging task. The table reports overall PoseBusters validity, the average SA score, the percentage of initial samples passing both validity and an SA score filter, the fraction of successful molecules with zero clashes (No clash), and the top-10 Vina docking score.

| Target | Method | Valid ↑ | SA Score ↓ | Valid (SA<4.5) ↑ | No clash ↑ | Vina top10 ↓ |
|---|---|---|---|---|---|---|
| SARS-CoV-2 $M_{pro}$ | Interpolate–integrate | 41.0% | $4.49 \pm 1.20$ | 23.5% | **15.4%** | $-6.48 \pm 0.24$ |
| | Replacement–guidance | 32.2% | $\mathbf{3.61 \pm 0.90}$ | **27.3%** | 12.4% | $\mathbf{-7.63 \pm 0.42}$ |
| | Known Binders | 100% | $3.70 \pm 0.97$ | 80.1% | 67.5% | $-9.95 \pm 0.07$ |

## 4.5 INFERENCE RUNTIME.

On a single RTX 6000 GPU, generating a batch of 10 takes 2.85 s with *Interpolate–Integrate* ($\tau = 0.75$) and 3.9 s with *Replacement Guidance*, measured on 50-atom molecules. CPU runtimes are 28.3 s and 38.5 s per sample (batch size 10). Batching improves efficiency but currently supports only identical input seeds per batch. In contrast, SHEPHERD requires approximately 3–4 minutes to generate a batch of 10 samples with 400 denoising steps on a single V100 GPU, or 5–10 minutes per sample on CPU (Adams et al., 2025).

## 4.6 VALIDATION OF CONDITIONING MECHANISMS

We performed ablation studies to validate the design principles of our conditioning strategies.

**Replacement Guidance vs. Inpainting.** Standard diffusion inpainting replaces masked regions with a noised version of the target fragment, keeping the input locally in-distribution (Lugmayr et al., 2022). Table 13 compares this approach to our clean hard-replacement strategy on the DIFFLINKER set (Igashov et al., 2024), which includes cases with three or more disconnected fragments. *Replacement Guidance* achieves higher overall validity (58.6% vs. 32.0%) and better similarity than fully-noised inpainting. Further analysis is in Appendix B.2.

**Behaviour of *Interpolate–Integrate* on disconnected seed structures.** We evaluate whether *Interpolate–Integrate* can merge disconnected inputs into a single valid molecule. As shown in Appendix B.1, the method can "heal" disconnected seeds from the DIFFLINKER test set (Igashov et al., 2024). Table 10 shows the relationship between the interpolation time $\tau$ and validity: as $\tau$ decreases (i.e., more noise is injected), the validity increases from 65.3% to 84.4%.

## 5 CONCLUSION

We have introduced two *training-free*, inference-time conditioning strategies—*Interpolate–Integrate* and *Replacement Guidance*—that operate on top of the fast *SemlaFlow* model for ligand-only, bioisosteric design. Both methods can steer 3D molecular generation to satisfy interaction-level constraints such as shape and pharmacophore overlap. We demonstrate their effectiveness across three tasks: natural product ligand hopping, bioisosteric fragment merging, and pharmacophore merging, showing that they generate synthetically accessible molecules that match key reference features. Unlike prior methods that rely on explicit retraining or task-specific encoders, our framework preserves sampling efficiency and generalizes across tasks and conditioning modes. These results establish inference-time conditioning as a powerful and scalable strategy for goal-directed molecular design, whose modularity makes it readily extendable to new scientific domains such as protein design and materials discovery.

ACKNOWLEDGEMENTS

We thank the authors of the SHEPHERD paper for their support, assistance with the experimental setups, and for guidance on code usage. The authors also thank Leo Klarner and Lucy Vost from the Oxford Protein Informatics Group for their helpful comments. We additionally thank Nele Quast, Alona Jurgenson, and Fergus Imrie for their valuable feedback during the rebuttal period. This work was supported by the Centre of Artificial Intelligence in Precision Medicines (CAIPM), King Abdulaziz University, Jeddah, Saudi Arabia.

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

## A    APPENDIX OVERVIEW

This appendix contains material that supports and extends the main text.

- B: Ablation studies on inference-time conditioning parameters
  - B.1: Interpolate–Integrate ablations
  - B.2: Replacement Guidance ablations
- C: Evaluation metrics
- D: Pharmacophore feature definitions
- E: Extended results for natural product hopping
- F: Detailed evaluation of bioisosteric fragment merging
- G: Extended evaluation of pharmacophore-merging on SARS-CoV-2 $M_{pro}$
- H: Limitations
- J: Reproducibility Statement
- I: Statement on Large Language Model (LLM) Usage

## B    ABLATION: CONDITIONING PARAMETERS

We quantify how the principal inference-time hyper-parameters of the two conditioning schemes Interpolate–Integrate and Replacement Guidance, affect validity, diversity, and similarity.

**Validity, uniqueness, and novelty.**    *Validity* refers to passing all PoseBusters (Buttenschoen et al., 2023) intramolecular checks. *Uniqueness* and *novelty* are computed on canonical SMILES obtained via `RDKit MolToSmiles`. A generated molecule is *novel* if its SMILES is different from the seed molecule, and *unique* within a batch if its SMILES appears exactly once.

**Similarity metrics.**    *Shape and color similarity score (SC$_{RDKit}$):* assesses the 3D similarity between generated molecules and a reference molecule, as described in Imrie et al. 2021. SC$_{RDKit}$ scores range from 0 (no match) to 1 (perfect match). The color similarity function evaluates the overlap of pharmacophoric features, while the shape similarity measure involves a simple volumetric comparison between the two conformers. SC$_{RDKit}$ uses two RDKit functions, based on the methods described in Putta et al. 2005 and Landrum et al. 2006.
*ECFP4 Tanimoto* is the Tanimoto coefficient between 2 048-bit Morgan fingerprints (Morgan, 1965) (radius 2) generated with `RDKit GetMorganFingerprintAsBitVect`—a standard measure of 2-D scaffold similarity.

### B.1    INTERPOLATE–INTEGRATE

We analyze the effect of varying the restart time $\tau = t$ and the seed noise scale $\sigma$. All experiments use the first 1,000 molecules from the GEOM-Drugs (Axelrod & Gómez-Bombarelli, 2022) test set as reference seeds.

### B.1.1    EFFECT OF TIME $\tau$

In all ablation experiments, we use the 100-step, log-spaced ODE grid from SEMLAFLOW. In Interpolate–Integrate, the restart time $\tau$ is set by choosing a step index $s$ in this grid; $\tau$ is simply the $s$-th value in the solver's precomputed log-spaced time schedule. Table 5 lists the restart step $s$ (index in the log-spaced 100-step grid) and the corresponding continuous time $\tau$.

We restart the flow at six grid positions ($s \in [70, 80] \rightarrow \tau \in [0.75, 0.6]$). Larger $\tau$ (late restarts) preserves more of the seed; smaller $\tau$ yields greater novelty at the cost of similarity. Unless otherwise stated, the seed-noise scale is fixed to $\sigma = 0.1$.

The choice of the restart time, $\tau$, reveals a clear trade-off between novelty and fidelity. Lower values of $\tau$ produce more novel and unique molecules, while higher values improve chemical validity and 3D similarity (SC$_{RDKit}$) to the seed molecule. These trends are detailed in Tables 6 and 7 and visualized in Figure 4.

Table 5: Mapping between restart steps in the log-spaced ODE grid and the corresponding continuous time $\tau$ used in Interpolate–Integrate.

| Restart steps | $\tau$ (continuous time) |
|---|---|
| 70 | 0.75 |
| 72 | 0.72 |
| 74 | 0.70 |
| 76 | 0.69 |
| 78 | 0.64 |
| 80 | 0.60 |

Table 6: Validity of generated molecules conditioned on a reference at noise level $\sigma = 0.1$. Each row reports the number of molecules attempted and the percentage that were generated, connected, chemically valid, physically valid, and overall valid. A molecule is considered fully valid only if it passes all three criteria: connectedness, chemical validity, and physical validity.

| $s$ | % Generated | % connectedness | % chemical | % physical | % Valid |
|---|---|---|---|---|---|
| 0.70 | 95.8 | 95.3 | 95.8 | 89.5 | 89.0 |
| 0.72 | 94.0 | 92.8 | 94.0 | 87.4 | 86.3 |
| 0.74 | 91.6 | 89.0 | 91.5 | 85.4 | 83.1 |
| 0.76 | 90.0 | 86.7 | 90.0 | 84.7 | 81.6 |
| 0.78 | 90.6 | 87.0 | 90.6 | 86.2 | 82.8 |
| 0.80 | 91.5 | 88.0 | 91.5 | 87.8 | 84.4 |

Table 7: Validity, uniqueness, and novelty of generated molecules conditioned on a reference molecule at noise level $\sigma = 0.10$. Values are reported as proportions of the total number of molecules that were to be generated.

| $s$ | Valid | Valid & Novel | Valid & Unique | Valid & Unique & Novel |
|---|---|---|---|---|
| 0.70 | 89.05 | 85.18 | 20.63 | 20.36 |
| 0.72 | 86.31 | 83.38 | 39.03 | 38.74 |
| 0.74 | 83.08 | 81.41 | 61.99 | 61.70 |
| 0.76 | 81.56 | 80.88 | 76.38 | 76.12 |
| 0.78 | 82.78 | 82.57 | 82.13 | 81.97 |
| 0.80 | 84.45 | 84.34 | 84.37 | 84.24 |

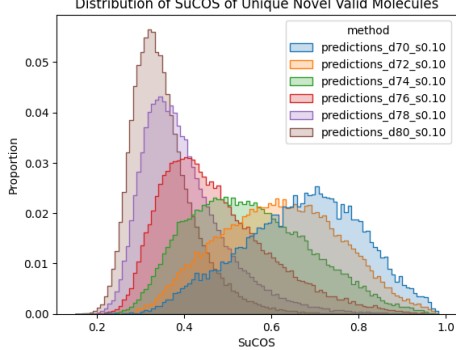
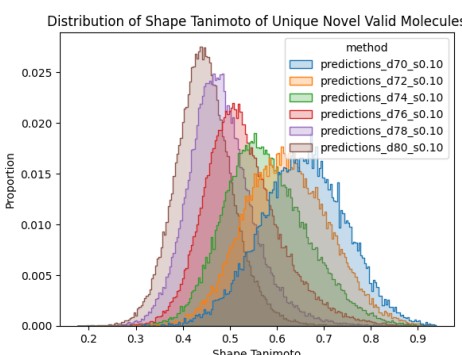

(a) Distribution of $SC_{RDKit}$ between the generated and reference molecules.

(b) Cumulative distribution of ECFP4 Tanimoto similarity. A sharper jump near 1.0 indicates more identical fingerprints.

Figure 3: Chemical and shape/pharmacophoric similarity between generated and reference molecules, measured by ECFP4 Tanimoto similarity and $SC_{RDKit}$, respectively. Curves are shown for different integration step counts at fixed noise level $\sigma = 0.1$. Only valid molecules are included.

### B.1.2 EFFECT OF NOISE LEVEL $\sigma$

We fix $s = 0.72$ and vary the noise $\sigma$ added to the interpolated seed. Higher $\sigma$ injects more coordinate noise at the restart point.

Table 8: Validity of generated molecules conditioned on a reference molecule for restart step $s = 0.72$, across different noise levels $\sigma$. Each row reports the total number of molecules attempted and the percentage that were generated, connected, chemically valid, physically valid, and overall valid. A molecule is considered valid only if it meets all three criteria.

| $s$ | $\sigma$ | Total | % Gen. | % Conn. | % Chem. | % Phys. | % Valid |
|---|---|---|---|---|---|---|---|
| | 0.00 | 94956 | 95.0 | 94.2 | 94.9 | 88.5 | 87.8 |
| | 0.05 | 94765 | 94.8 | 93.9 | 94.8 | 88.1 | 87.3 |
| 0.72 | 0.10 | 94019 | 94.0 | 92.8 | 94.0 | 87.4 | 86.3 |
| | 0.15 | 92807 | 92.8 | 91.0 | 92.8 | 86.4 | 84.7 |
| | 0.20 | 91043 | 91.0 | 88.4 | 91.0 | 84.7 | 82.2 |
| | 0.25 | 89617 | 89.6 | 86.0 | 89.6 | 83.5 | 80.2 |

Table 9: Validity, uniqueness, and novelty of generated molecules conditioned on a reference molecule for $s = 0.72$, across varying noise levels $\sigma$. Values are reported as proportions of the total number of molecules that were to be generated.

| $s$ | $\sigma$ | Valid | Valid & Novel | Valid & Unique | Valid & Unique & Novel |
|---|---|---|---|---|---|
| | 0.00 | 87.83 | 84.42 | 30.80 | 30.52 |
| | 0.05 | 87.35 | 84.04 | 32.73 | 32.45 |
| 0.72 | 0.10 | 86.31 | 83.38 | 39.03 | 38.74 |
| | 0.15 | 84.69 | 82.31 | 49.57 | 49.26 |
| | 0.20 | 82.16 | 80.50 | 60.45 | 60.16 |
| | 0.25 | 80.18 | 79.23 | 69.52 | 69.26 |

Tables 8 and 9 show a clear trade-off: higher $\sigma$ values increase novelty but reduce validity. This suggests $\sigma$ should be tuned for each application based on desired diversity.

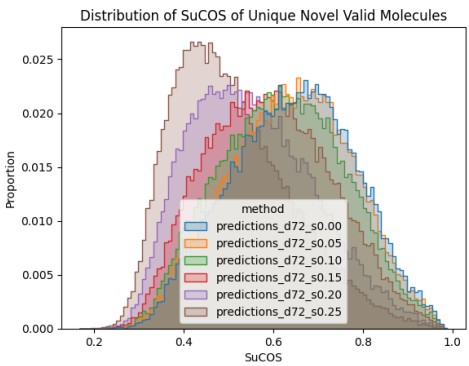
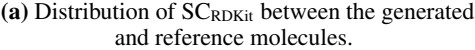

(a) Distribution of SC$_{RDKit}$ between the generated and reference molecules.

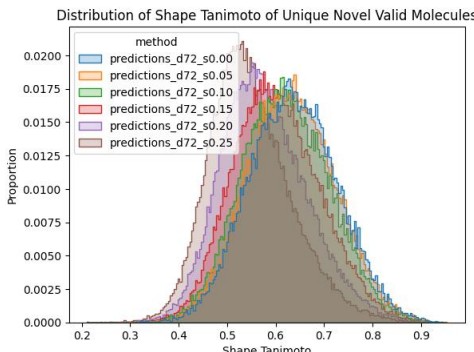

(b) Cumulative distribution of ECFP4 Tanimoto similarity. A sharper jump near 1.0 indicates more identical fingerprints.

Figure 4: Chemical and shape/pharmacophoric similarity between generated and reference molecules across different noise levels $\sigma$ in Interpolate–Integrate. We report SC$_{RDKit}$ (a) and ECFP4 Tanimoto similarity (b), computed only for valid molecules. Curves are shown for different $\sigma$ levels.

### B.1.3 BEHAVIOUR ON DISCONNECTED SEED STRUCTURES

We include an ablation study evaluating the "healing" capability of *Interpolate–Integrate* when provided with disconnected inputs. The experiment is performed on the DIFFLINKER test set (Igashov et al., 2024), using a curated subset of 1,129 examples, each consisting of three spatially separated fragments. This setup provides a deliberately challenging test of whether the model can transform a set of unconnected seed structures into a single, chemically valid molecule. Because the flow preserves dimensionality, *Interpolate–Integrate* always generates a molecule with the same number of atoms as the input fragments.

We measure how generation validity varies as a function of the restart step $s$, which determines the noise level injected into the seed structure. As defined in Table 5, a larger step index $s$ corresponds to a smaller continuous time $\tau$, meaning that more noise is added to the input and the ODE is integrated for a longer duration. This allows us to directly assess how increased noise facilitates the model's ability to "heal" disconnected configurations.

Table 10: Validity of generated molecules conditioned on disconnected 3-fragment seeds from the DIFFLINKER set, as a function of restart step $s$. Each row reports the percentage of molecules that were generated, connected (chemical validity), physically valid, and overall valid.

| Restart Step ($s$) | % Generated | % Chemical | % Physical | % Valid |
|---|---|---|---|---|
| 68 | 94.6 | 62.4 | 86.4 | 58.1 |
| 70 | 93.6 | 66.2 | 85.7 | 61.1 |
| 72 | 94.8 | 68.2 | 87.8 | 63.0 |
| 74 | 94.2 | 77.0 | 88.9 | 72.9 |
| 76 | 95.4 | 83.5 | 90.7 | 79.6 |
| 78 | 93.7 | 84.5 | 89.3 | 80.8 |
| 80 | 94.1 | 86.0 | 89.5 | 82.2 |

The results in Table 10 show a clear monotonic trend: as $s$ increases from 68 to 80, the overall validity rises from 58.1% to 82.2%. This confirms our hypothesis. A larger $s$ (more noise) moves the restart point further from the "broken" disconnected seed, giving the vector field a longer, smoother trajectory to "heal" the structure, generate linkers, and converge to a valid, connected molecule on the data manifold. Conversely, a smaller $s$ (less noise) starts too close to the out-of-distribution input, and the vector field fails to correct the disconnections. Figure 5 provides visual examples of this process.

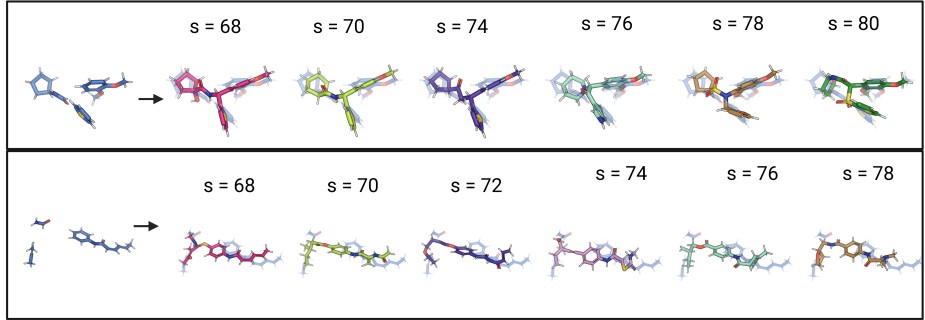

Figure 5: **Effect of the restart step $s$ on the output of *Interpolate–Integrate*.** In each panel, the left column shows the disconnected reference fragments provided as input. The right column shows generated molecules for different values of $s$. Lower values of $s$ inject less noise and leave fewer integration steps, causing the model to stay close to the reference fragments and largely preserve their geometry. Higher values of $s$ add more noise and allow longer integration, giving the model greater freedom to deviate from the seed and produce increasingly modified structures.

### B.1.4 EFFECT OF NUMBER OF SEED FRAGMENTS

To study how the number of input fragments affects generation quality, we performed an ablation in the $M^{pro}$ case where we supplied the model with varying numbers of fragments as the seed. For each configuration, we generated 100 molecules with sizes between 25–35 atoms. We start from 2 fragment till 10. For each configuration, we also estimate the molecular volume of the generated seeds using RDKit's `AllChem.ComputeMolVolume`, which provides an approximate van der Waals volume based on the 3D conformer.

As shown in Table 11, increasing the number of fragments generally degrades performance, confirming that higher fragment counts increase task difficulty. This overall downward trend is not monotonic; the model exhibits fluctuating robustness in the mid-range. The volume measurements help contextualize this behavior. The sharp increase in molecular volume between 5 and 6 fragments corresponds to a pronounced drop in validity, suggesting that sudden expansions in the seed's spatial extent reduce structural coherence and hinder successful reconstruction. Conversely, at 8 fragments the volume-per-fragment ratio is relatively low, which may explain why performance temporarily stabilizes before dropping again for highly fragmented inputs (9–10 fragments).

Table 11: Ablation on the number of seed fragments provided to the model. For each setting, we generate 100 molecules of size 25–35 atoms in the $M^{pro}$ case.

| # Fragments | Volume ($Å^3$) | Vol./Frag | % Generated | % Chemical | % Physical | % Valid |
|---|---|---|---|---|---|---|
| 2 | 211.63 | 105.8 | 91.0 | 90.0 | 86.0 | 84.0 |
| 3 | 282.57 | 94.2 | 92.0 | 85.0 | 88.0 | 81.0 |
| 4 | 290.47 | 72.6 | 87.0 | 79.0 | 80.0 | 72.0 |
| 5 | 393.12 | 78.6 | 86.0 | 80.0 | 84.0 | 77.0 |
| 6 | 513.30 | 85.6 | 85.0 | 78.0 | 82.0 | 75.0 |
| 7 | 559.46 | 79.9 | 90.0 | 80.0 | 84.0 | 74.0 |
| 8 | 586.67 | 73.3 | 86.0 | 83.0 | 82.0 | 79.0 |
| 9 | 628.82 | 69.9 | 72.0 | 65.0 | 69.0 | 61.0 |
| 10 | 741.60 | 74.2 | 77.0 | 68.0 | 74.0 | 65.0 |

## B.2 REPLACEMENT GUIDANCE

We investigate the effect of the relaxation time $t_{\text{free}}$ in Replacement Guidance, where $t_{\text{free}}$ denotes the integration step after which fragment masking is disabled. We also compare this strategy against an inpainting-based conditioning approach.

Experiments are conducted on the DIFFLINKER test set (Igashov et al., 2024), using a curated subset of 1,129 cases in which the objective is to generate a molecule that preserves the shape and pharmacophoric features of three bound fragments.

### B.2.1 EFFECT OF TIME $t_{\text{FREE}}$

The relaxation time, $t_{\text{free}}$, directly affects the properties of the generated molecules, as shown in Table 12. Increasing $t_{\text{free}}$—which holds the fragment constraints for a longer duration—consistently improves the 3D similarity ($SC_{\text{RDKit}}$) to the reference fragments. Conversely, this stronger anchoring can negatively impact PoseBusters' validity across all metrics.

Table 12: Effect of relaxation time $t_{\text{free}}$ in Replacement Guidance. We report the proportion of molecules that are successfully generated, pass all chemical and physical PoseBusters' validity checks, meet the full validity criteria, and the $SC_{\text{RDKit}}$. Larger $t_{\text{free}}$ allows fragment replacement for more steps, leading to greater flexibility but weaker fragment anchoring.

| $t_{\text{free}}$ | % Generated | % Chemical | % Physical | % Valid | $SC_{\text{RDKit}}$ |
|---|---|---|---|---|---|
| 0.85 | 89.8 | 79.0 | 81.5 | 72.2 | $0.49 \pm 0.10$ |
| 0.90 | 89.6 | 79.2 | 82.0 | 72.8 | $0.50 \pm 0.10$ |
| 0.95 | 89.7 | 79.1 | 82.0 | 72.6 | $0.51 \pm 0.10$ |
| 1.00 | 78.0 | 65.4 | 68.4 | 58.6 | $0.55 \pm 0.10$ |

### B.2.2 REPLACEMENT GUIDANCE VS. NOISED INPAINTING CONDITIONING

In this ablation, we compare *Replacement Guidance* to an inpainting-style conditioning strategy. For inpainting, instead of inserting the clean fragment at each step, we interpolate the fragment to time $t$ and use this noised fragment for replacement, making the input locally in-distribution. The inpainting baseline uses the same update schedule and the same $t_{\text{free}} = 1$ relaxation setting as Replacement Guidance, ensuring a fully matched comparison.

To disentangle the effect of noising different components, we evaluate four inpainting variants: (i) noising only the coordinates while hard-replacing atom and bond features,(ii) noising only the atom types, while hard-replacing coordinates and bond logits (iii) noising only the bond logits, while hard-replacing coordinates and atom types, and (iv) noising all components simultaneously (the standard inpainting analogue).

As shown in Table 13, *Replacement Guidance* achieves significantly higher overall validity (58.6%) compared to standard, fully-noised inpainting (32.0%). The steep drop in *Inpaint–Coords* (35.6%) confirms that noising the geometry is the primary failure mode, underscoring the critical role of a stable geometric anchor.

We also note that the *Inpaint–Bonds* configuration—where coordinates and atoms remain hard-constrained—yields validity (59.3%) and shape similarity ($SC_{\text{RDKit}}$ 0.56) that are highly comparable to *Replacement Guidance* (58.6% and 0.55, respectively). This reinforces that a hard coordinate constraint is key to both structural coherence and shape retention. While this specific *Inpaint–Bonds* setup warrants further investigation, this ablation confirms our hypothesis that maintaining a hard, un-noised geometric anchor is superior to standard inpainting.

To further analyze why Replacement Guidance yields higher validity than inpainting, we quantify the geometric stability of both approaches by tracking the L2 distance between the predicted coordinates and the clean seed coordinates throughout the entire reverse trajectory. Figure 6 shows this deviation averaged over 1129 runs with different seeds. Replacement Guidance shows a large drop in L2 distance (Å) at the beginning of the trajectory, after which the distance remains relatively

Table 13: Validity comparison between inpainting configurations and Replacement Guidance ($t_{\text{free}} = 1.0$) on the DIFFLINKER test subset (1,129 cases). We report the percentage of molecules that are successfully generated, chemically valid, physically valid, valid overall (i.e., passing all checks), and $\text{SC}_{\text{RDKit}}$.

| Method | % Generated | % Chemical | % Physical | % Valid | $\text{SC}_{\text{RDKit}}$ |
|---|---|---|---|---|---|
| Inpaint coords (Hard-Replace Atoms + Bonds) | 61.8 | 41.4 | 49.7 | 35.6 | 0.45±0.11 |
| Inpaint atoms (Hard-Replace Coords + Bonds) | 74.8 | 62.8 | 64.9 | 56.1 | 0.52±0.10 |
| Inpaint bonds (Hard-Replace Coords + Atoms) | 80.8 | 66.4 | 70.0 | 59.3 | 0.56±0.11 |
| Inpaint (all) | 57.0 | 38.3 | 44.1 | 32.0 | 0.44±0.10 |
| Replacement Guidance | 78.0 | 65.4 | 68.4 | 58.6 | 0.55±0.10 |

stable and low. In contrast, the inpainting curve decreases much more gradually and stays at a higher L2 distance throughout the trajectory.

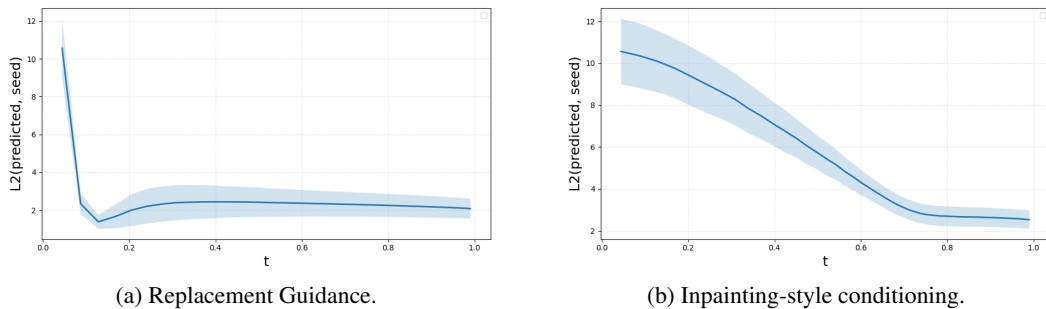

(a) Replacement Guidance.        (b) Inpainting-style conditioning.

Figure 6: Comparison of coordinate stability during generation under (a) *Replacement Guidance* and (b) inpainting-style conditioning. Each curve reports the mean ± standard deviation (shaded) of the L2 distance (Å) between the predicted coordinates and the clean input seed coordinates across 1129 different seeds.

### B.2.3 EFFECT OF CONDITIONING INFORMATION

To assess how different conditioning signals influence generation quality in *Replacement-guidance*, we perform an ablation over three levels of structural information provided to the model: (i) 3D coordinates only (coords), (ii) coordinates with atom-type annotations (coords + atoms), and (iii) coordinates supplemented with both atom types and bond connectivity (coords + atoms + bonds). As shown in Table 14, increasing the amount of conditioning information consistently reduces chemical and physical validity, at the same time, $\text{SC}_{\text{RDKit}}$ similarity increases with richer conditioning, reflecting that more explicit structural guidance leads to molecules that adhere more closely to the provided fragment template, as expected.

### B.2.4 EFFECT OF NUMBER OF SEED FRAGMENTS

Table 15 reports an ablation of *Replacement-guidance* in the $M^{\text{pro}}$ case, where the model is conditioned on between 2 and 10 input fragments. For each configuration, we generate 100 molecules with sizes between 25–35 atoms. We also estimate the molecular volume of each seed using RDKit's `AllChem.ComputeMolVolume`, which provides an approximate van der Waals volume based on the 3D conformer.

The relationship between fragment count and generation quality is not strictly monotonic. Using two to five fragments yields strong performance across all validity metrics, with five fragments achieving the highest full validity (82%). A sharp increase in molecular volume between five and six

Table 14: Effect of conditioning information in Replacement Guidance. We compare three levels of structural context provided to the model: coordinates only (coords), coordinates with atom types (coords + atoms), and coordinates with atom types and bond connectivity (coords + atoms + bonds). We report the proportion of molecules that are successfully generated, pass chemical and physical PoseBusters validity checks, and meet full validity criteria.

| Conditioning | % Generated | % Chemical | % Physical | % Valid | $SC_{RDKit}$ |
|---|---|---|---|---|---|
| coords | 92.5 | 86.4 | 92.5 | 86.3 | 0.35±0.06 |
| coords + atoms | 85.2 | 75.8 | 85.2 | 75.7 | 0.43±0.08 |
| coords + atoms + bonds | 78.0 | 65.4 | 68.4 | 58.6 | 0.55±0.10 |

fragments coincides with a noticeable drop in validity, suggesting that abrupt expansions in spatial extent make reconstruction more difficult. Performance partially recovers at seven and eight fragments—where the volume-per-fragment ratio decreases—but declines again for highly fragmented inputs (9–10 fragments), indicating that excessive fragmentation ultimately reduces structural coherence and hampers the model's ability to reconstruct chemically consistent molecules.

Table 15: Ablation on the number of seed fragments provided to the model under *Replacement-guidance*. For each setting, we generate 100 molecules of size 25–35 atoms in the $M^{pro}$ case. Increasing the number of fragments gradually reduces structural coherence, making reconstruction more challenging for highly fragmented inputs.

| # Fragments | Volume ($Å^3$) | Vol./Frag | % Generated | % Chemical | % Physical | % Valid |
|---|---|---|---|---|---|---|
| 2 | 211.63 | 105.8 | 91.0 | 91.0 | 87.0 | 86.0 |
| 3 | 282.57 | 94.2 | 85.0 | 81.0 | 79.0 | 75.0 |
| 4 | 290.47 | 72.6 | 79.0 | 75.0 | 75.0 | 71.0 |
| 5 | 393.12 | 78.6 | 87.0 | 86.0 | 84.0 | 82.0 |
| 6 | 513.30 | 85.6 | 72.0 | 69.0 | 63.0 | 61.0 |
| 7 | 559.46 | 79.9 | 83.0 | 78.0 | 70.0 | 66.0 |
| 8 | 586.67 | 73.3 | 83.0 | 78.0 | 74.0 | 68.0 |
| 9 | 628.82 | 69.9 | 83.0 | 79.0 | 68.0 | 66.0 |
| 10 | 741.60 | 74.2 | 74.0 | 69.0 | 64.0 | 58.0 |

## C  EVALUATION METRICS

We assess the generated molecules using the following metrics:

**Synthetic Accessibility (SA) Score.** Several scoring systems exist to estimate the synthetic accessibility of a compound (Coley et al., 2018; Thakkar et al., 2021; Ertl & Schuffenhauer, 2009). We use the SA score of Ertl and Schuffenhauer (Ertl & Schuffenhauer, 2009), which combines historical data from synthesized compounds with molecular complexity to assess synthetic feasibility. Lower scores indicate greater ease of synthesis.

**Validity Checks.** Molecules are validated at multiple stages:

- RDKit Validity: The molecule can be successfully parsed and sanitized by RDKit.
- PoseBusters Validity: The molecule passes chemical and physical plausibility checks using PoseBusters (Buttenschoen et al., 2023).

**Similarity Metrics:** We adapted similarity scores introduced in the ShEPhERD paper (Adams et al., 2025) with the same settings (xTB relaxtion in implicit water and ESP alignment with 400 surface points)

- Surface Similarity: Quantifies the volumetric overlap between two atomic point clouds of the generated and reference molecules. High scores indicate preservation of 3D shape.

- Electrostatic Similarity: Measures the similarity between the electrostatic potential fields of the generated and reference molecules, capturing charge complementarity relevant for molecular recognition.

- Pharmacophore Similarity: Assesses the overlap of key pharmacophoric features—hydrogen bond donors (HBD), acceptors (HBA), and aromatic centers—between the generated and reference molecules. Directional alignment is included when necessary to capture interaction specificity.

**Docking Score.** We used AutoDock Vina (Eberhardt et al., 2021) to further assess the quality of the generated molecules. We used the Vina scoring function to assess molecules with binding site conditional poses. Since such scoring functions are sensitive to the exact coordinates and can assign experimental structures poor scores without energy minimization (Weller & Rohs, 2024), we computed energy-minimized scores for generated molecules. This protocol evaluates the plausibility of the generated pose via local energy minimization and does not perform a global pose search (i.e., redocking). For this minimization step, the binding site was defined with a buffer of 5 Å around the reference ligand.

## D    PHARMACOPHORE FEATURE DEFINITIONS AND EXTRACTION PROTOCOLS

Pharmacophores can be used as a representation of chemical interactions crucial for ligand binding to macromolecular targets (Schaller et al., 2020). These interactions include hydrogen bonds, charge interactions, and lipophilic contacts. Pharmacophores can be derived through ligand-based or structure-based approaches, providing versatility in their application. In this study, we employed 3D pharmacophores derived from ground truth molecules.

Throughout this work, when conditioning on pharmacophoric features, we define features using the standard `RDKit FeatureFactory` (based on the `BaseFeatures.fdef` definition file). This implementation is also provided in our supplementary code.

We primarily extract the 3D coordinates of three specific feature types: Hydrogen Bond Donors (HBD), Hydrogen Bond Acceptors (HBA), and Aromatic Rings. These extracted feature points are not used as a post-hoc filter, but rather as the direct input seed structure ($z_1$) for the generative process. After finding the subset of HBD/HBA/Aromatic features, we use their original 3D coordinates to construct the seed.

It is important to distinguish this conditioning protocol (used as input for our model) from the evaluation metric (used to score the output). For the pharmacophore similarity score (from the SHEP-HERD protocol (Adams et al., 2025)), a much broader set of features is used. This includes both directional features (such as HBA, HBD, and Aromatic ring vectors) and non-directional features (such as Hydrophobes, Anions, and Cations) to provide a comprehensive assessment of 3D property overlap.

## E    NATURAL PRODUCT TASK: EXTENDED EVALUATION

To complement the main results, we provide an extended evaluation of model behavior on the natural product (NP) hopping task. We report molecular validity, synthetic accessibility scores, pharmacophore feature recovery, and conformational stability for generated analogues across all three NP targets.

### E.1    EXPERIMENTAL DETAILS

For each seed NP, we generated 2,500 candidate analogues with total atom counts between 36 and 80 (including hydrogens). We used our two conditioning strategies: *Interpolate-Integrate* (restarting at a continuous time of $\tau = 0.75$) and *Replacement Guidance* (with a relaxation time of $t_{\text{relax}} = 1.0$). Both approaches were conditioned by sampling key pharmacophore features—hydrogen bond donors (HBD), acceptors (HBA), and aromatic atoms—from the reference NP. The remaining atoms

in the seed structure were sampled randomly from the full set of atoms in the reference. All generated molecules were post-processed with xTB geometry relaxation and ESP alignment before evaluation.

## E.2 VALIDITY AND FILTERING PIPELINE

Table 16 reports the validity of the generated molecules at each stage of our filtering pipeline. The first stage assesses RDKit validity, followed by a geometry quality check using PoseBusters. The final filter is based on synthetic accessibility (SA < 4.5), where `Replacement Guidance` demonstrates a significantly higher final yield of successful candidates.

Table 16: Reported validity of generated molecules for the natural product (NP) hopping task. For each NP target (NP1, NP2, NP3), 2,500 molecules were generated using two methods: *interpolate–integrate* and *replacement–guidance*. We report the percentage of molecules that (1) are valid according to RDKit parsing, (2) pass all PoseBusters (PB) geometry checks prior xTB relaxation, (3) pass all PoseBusters geometry checks after xTB geometry relaxation and ESP alignment, and (4) additionally have a synthetic accessibility (SA) score < 4.5.

| NP | Method | Valid (RDKit) | PB Valid | xTB + PB Valid | xTB + PB Valid + SA < 4.5 |
|----|--------|---------------|----------|----------------|----------------------------|
| | ShEPhERD | 65.5% | 55.5% | 57.3% | 24.7% |
| | Interpolate–integrate | **89.4%** | **64.9%** | **71.5%** | 35.6% |
| NP1 | Replacement–guidance | 72.5% | 51.2% | 60.5% | **50.2%** |
| | ShEPhERD | 56.5% | 50.0% | 51.0% | 17.4% |
| | Interpolate–integrate | **98.5%** | **91.4%** | **93.6%** | 16.0% |
| NP2 | Replacement–guidance | 93.8% | 78.2% | 87.9% | **48.8%** |
| | ShEPhERD | 55.5% | 39.7% | 44.1% | 20.2% |
| | Interpolate–integrate | **77.1%** | **50.1%** | 62.7% | 5.2% |
| NP3 | Replacement–guidance | 72.9% | 48.2% | **64.4%** | **42.1%** |

## E.3 PERFORMANCE ON FILTERED CANDIDATES (SA < 4.5)

Table 17 presents the primary performance metrics for the final set of candidates that passed all filters, including the SA <4.5 threshold. All molecules were subject to xTB geometry relaxation and ESP alignment before scoring. These results show that *Replacement-guidance* consistently yields the largest fraction of fully valid, synthetically accessible candidates and generates molecules with the best synthetic accessibility scores across all three natural product targets. In contrast, *Interpolate–integrate* is the superior method for achieving the highest 3D similarity scores.

Table 17: Performance summary for molecular generation across NP hopping tasks (NP1, NP2, NP3) after xTB geometry relaxation and ESP alignment. We report PoseBusters Success Rate (Valid and SA < 4.5), similarity metrics (ESP and pharmacophore), and SA score. Only molecules with synthetic accessibility (SA) scores less than 4.5 are included in this analysis.

| NP | Method | Success Rate ↑ | ESP sim ↑ | ESP sim top10 ↑ | Pharma sim ↑ | Pharma sim top10 ↑ | SA ↓ |
|----|--------|----------------|-----------|-----------------|--------------|---------------------|------|
| NP1 | ShEPhERD | 24.7% | $0.48 \pm 0.06$ | $0.63 \pm 0.02$ | $0.17 \pm 0.06$ | $0.34 \pm 0.03$ | $3.95 \pm 0.45$ |
| | Interpolate–integrate | 35.6% | $\mathbf{0.51 \pm 0.11}$ | $\mathbf{0.81 \pm 0.02}$ | $0.18 \pm 0.09$ | $0.49 \pm 0.06$ | $3.81 \pm 0.55$ |
| | Replacement–guidance | **50.2%** | $0.50 \pm 0.09$ | $\mathbf{0.81 \pm 0.02}$ | $0.18 \pm 0.08$ | $\mathbf{0.52 \pm 0.06}$ | $\mathbf{3.36 \pm 0.66}$ |
| NP2 | ShEPhERD | 17.4% | $0.68 \pm 0.06$ | $0.79 \pm 0.01$ | $0.20 \pm 0.06$ | $0.34 \pm 0.02$ | $3.76 \pm 0.57$ |
| | Interpolate–integrate | 16.0% | $\mathbf{0.70 \pm 0.08}$ | $\mathbf{0.86 \pm 0.01}$ | $\mathbf{0.22 \pm 0.10}$ | $\mathbf{0.46 \pm 0.02}$ | $3.90 \pm 0.48$ |
| | Replacement–guidance | **48.8%** | $0.69 \pm 0.07$ | $0.85 \pm 0.01$ | $0.20 \pm 0.07$ | $0.44 \pm 0.04$ | $\mathbf{3.41 \pm 0.71}$ |
| NP3 | ShEPhERD | 20.2% | $\mathbf{0.46 \pm 0.06}$ | $0.60 \pm 0.01$ | $\mathbf{0.17 \pm 0.05}$ | $\mathbf{0.30 \pm 0.02}$ | $3.84 \pm 0.47$ |
| | Interpolate–integrate | 5.2% | $0.40 \pm 0.07$ | $0.56 \pm 0.05$ | $0.11 \pm 0.04$ | $0.20 \pm 0.03$ | $3.98 \pm 0.44$ |
| | Replacement–guidance | **42.1%** | $0.45 \pm 0.06$ | $\mathbf{0.65 \pm 0.03}$ | $0.14 \pm 0.04$ | $0.29 \pm 0.02$ | $\mathbf{3.33 \pm 0.72}$ |

## E.4 PERFORMANCE ON ALL VALID MOLECULES (UNFILTERED)

For a more comprehensive analysis, Table 18 summarizes the same performance metrics for all molecules that passed the initial PoseBusters validity checks, prior to SA filtering. These unfiltered results reveal the inherent trade-offs of the methods: *Interpolate-integrate* is most effective at generating a high percentage of initially valid structures, while *Replacement-guidance* demonstrates a strong intrinsic bias towards producing molecules with better synthetic accessibility.

Table 18: Performance summary for molecular generation across NP hopping tasks (NP1, NP2, NP3) after xTB geometry relaxation and ESP alignment. We report PoseBusters validity, similarity metrics (ESP and pharmacophore), and SA score.

| NP | Method | Valid ↑ | ESP sim ↑ | ESP sim top10 ↑ | Pharma sim ↑ | Pharma sim top10 ↑ | SA ↓ |
|---|---|---|---|---|---|---|---|
| NP1 | ShEPhERD | 57.3% | $0.47 \pm 0.07$ | $0.66 \pm 0.02$ | $0.17 \pm 0.06$ | $0.37 \pm 0.03$ | $4.75 \pm 0.89$ |
| | Interpolate–integrate | **71.5%** | $\mathbf{0.55 \pm 0.13}$ | $\mathbf{0.89 \pm 0.02}$ | $\mathbf{0.21 \pm 0.11}$ | $\mathbf{0.73 \pm 0.09}$ | $4.67 \pm 1.02$ |
| | Replacement–guidance | 60.5% | $0.52 \pm 0.11$ | $0.87 \pm 0.01$ | $0.19 \pm 0.10$ | $0.66 \pm 0.08$ | $\mathbf{3.74 \pm 1.01}$ |
| NP2 | ShEPhERD | 51.0% | $0.68 \pm 0.08$ | $0.81 \pm 0.01$ | $0.21 \pm 0.06$ | $0.39 \pm 0.02$ | $5.09 \pm 1.2$ |
| | Interpolate–integrate | **93.6%** | $\mathbf{0.77 \pm 0.09}$ | $\mathbf{0.93 \pm 0.01}$ | $\mathbf{0.35 \pm 0.17}$ | $\mathbf{0.86 \pm 0.07}$ | $5.41 \pm 1.03$ |
| | Replacement–guidance | 87.9% | $0.72 \pm 0.09$ | $0.91 \pm 0.02$ | $0.26 \pm 0.12$ | $0.82 \pm 0.08$ | $\mathbf{4.43 \pm 1.31}$ |
| NP3 | ShEPhERD | 44.1% | $0.45 \pm 0.06$ | $0.62 \pm 0.01$ | $\mathbf{0.17 \pm 0.04}$ | $0.32 \pm 0.02$ | $4.69 \pm 0.96$ |
| | Interpolate–integrate | 62.7% | $\mathbf{0.47 \pm 0.09}$ | $0.69 \pm 0.02$ | $0.14 \pm 0.06$ | $\mathbf{0.39 \pm 0.05}$ | $5.92 \pm 1.03$ |
| | Replacement–guidance | **64.4%** | $\mathbf{0.47 \pm 0.07}$ | $\mathbf{0.73 \pm 0.02}$ | $0.16 \pm 0.05$ | $\mathbf{0.39 \pm 0.04}$ | $4.07 \pm 1.27$ |

Table 19: Molecular property statistics for molecular generation across NP hopping tasks (NP1, NP2, NP3), after xTB geometry relaxation and ESP alignment. We report molecular weight, Lipinski rule-of-five violations, logP, QED, number of rings, and SMILES uniqueness, computed over PoseBusters-valid molecules.

| NP | Method | Weight [Da] | Lipinski R5 | logP | QED | # Rings | % Unique SMILES |
|---|---|---|---|---|---|---|---|
| NP1 | ShEPhERD | $388.46 \pm 47.64$ | $4.64 \pm 0.51$ | $1.16 \pm 1.59$ | $0.52 \pm 0.18$ | $2.97 \pm 1.27$ | 99.9% |
| | Interpolate–integrate | $412.26 \pm 76.59$ | $4.06 \pm 1.01$ | $1.53 \pm 1.97$ | $0.42 \pm 0.20$ | $2.83 \pm 1.11$ | 100% |
| | Replacement–guidance | $409.36 \pm 73.24$ | $4.37 \pm 0.77$ | $3.23 \pm 1.55$ | $0.50 \pm 0.19$ | $3.33 \pm 1.04$ | 99.7% |
| NP2 | ShEPhERD | $330.59 \pm 45.59$ | $4.86 \pm 0.35$ | $1.53 \pm 1.49$ | $0.66 \pm 0.12$ | $3.01 \pm 1.79$ | 99.9% |
| | Interpolate–integrate | $322.44 \pm 37.92$ | $4.70 \pm 0.59$ | $0.62 \pm 1.56$ | $0.51 \pm 0.16$ | $2.35 \pm 1.18$ | 84.5% |
| | Replacement–guidance | $319.27 \pm 34.13$ | $4.82 \pm 0.39$ | $2.07 \pm 1.31$ | $0.63 \pm 0.15$ | $2.56 \pm 1.13$ | 99.6% |
| NP3 | ShEPhERD | $422.54 \pm 44.14$ | $4.35 \pm 0.61$ | $1.27 \pm 1.57$ | $0.42 \pm 0.17$ | $2.73 \pm 1.10$ | 99.9% |
| | Interpolate–integrate | $431.50 \pm 78.56$ | $3.50 \pm 1.35$ | $-0.36 \pm 2.27$ | $0.35 \pm 0.20$ | $3.81 \pm 1.75$ | 100% |
| | Replacement–guidance | $416.12 \pm 76.61$ | $4.43 \pm 0.82$ | $2.98 \pm 1.44$ | $0.50 \pm 0.19$ | $3.68 \pm 1.44$ | 100% |

## E.5 DISTRIBUTIONS OF KEY METRICS

The figures below provide a visual summary of the performance trade-offs between the methods. Figure 7 demonstrates the clear advantage of *Replacement Guidance* in producing molecules with lower (better) SA scores. While *Interpolate–Integrate* yields more geometrically stable molecules post-relaxation (lower RMSD in Figure 8), Figure 9 confirms that *Replacement Guidance* and SHEPHERD generate a more chemically diverse set of candidates, reflecting broader exploration.

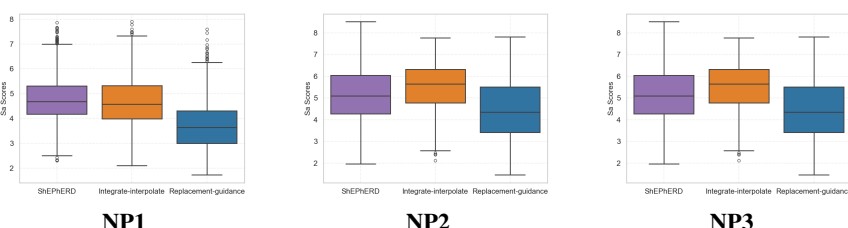

**NP1**          **NP2**          **NP3**

Figure 7: Synthetic accessibility (SA) score distributions for generated analogues across the three natural product tasks.

## E.6 DETAILED POSEBUSTERS VALIDITY

Tables 20, 21, and 22 provide a detailed breakdown of the PoseBusters validity evaluation. Table 20 reports overall generation success and combined chemical/physical validity. Table 21 decomposes chemical validity into sanitization, hydrogen compatibility, valence, and connectivity checks. Table 22 presents results from the PoseBusters suite, including bond lengths, angles, steric clashes, and energetic plausibility.

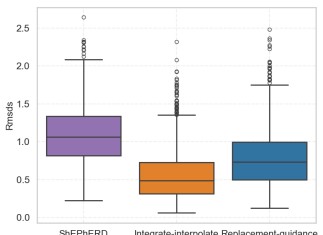 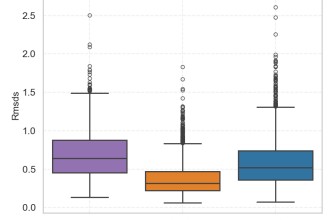 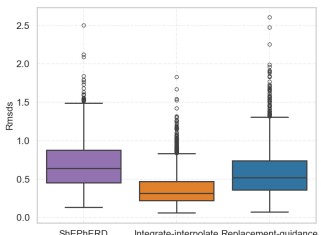

**NP1:** RMSD (aligned vs. original)   **NP2:** RMSD (aligned vs. original)   **NP3:** RMSD (aligned vs. original)

Figure 8:  RMSD between the xTB-relaxed/aligned and original generated conformers for each NP task.

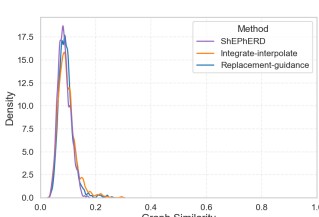 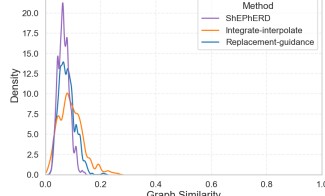 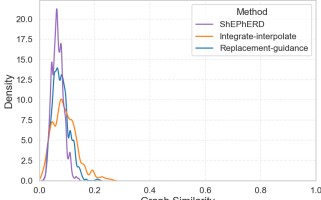

**NP0:** Graph similarity to reference   **NP1:** Graph similarity to reference   **NP2:** Graph similarity to reference

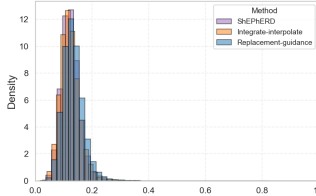 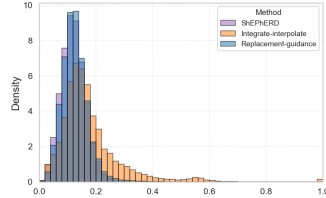 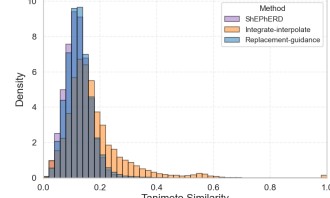

**NP0:** Pairwise Tanimoto similarity   **NP1:** Pairwise Tanimoto similarity   **NP2:** Pairwise Tanimoto similarity

Figure 9:   Graph similarity to the reference (top row) and pairwise Tanimoto similarity among generated ligands (bottom row) for the three NP hopping tasks.

Table 20:   Validity of generated molecules across the natural-product (NP) hopping tasks. Each method attempts 2,500 generations; we report the percentage of molecules that were generated (% Generated) and the proportion of those that pass chemical validity checks, physical PoseBusters checks, or both.

| NP | Method | % Generated | % Chemical | % Physical | % Valid |
|----|--------|-------------|------------|------------|---------|
| NP1 | Interpolate–integrate | 81.6 | 81.6 | 71.6 | 71.5 |
|     | Replacement–guidance | 68.2 | 68.2 | 60.5 | 60.5 |
| NP2 | Interpolate–integrate | 96.7 | 96.7 | 93.6 | 93.6 |
|     | Replacement–guidance | 92.6 | 92.6 | 88.0 | 87.9 |
| NP3 | Interpolate–integrate | 69.0 | 69.0 | 62.8 | 62.7 |
|     | Replacement–guidance | 67.6 | 67.6 | 64.4 | 64.4 |

### E.7    PHARMACOPHORE FEATURE RECOVERY

To better understand how pharmacophoric features are preserved during generation, we analyzed a subset of ligands produced by the *Replacement Guidance* strategy. Molecules were selected based on pharmacophore similarity from the filtered set with SA score $\leq 4.5$, and each recovered reference pharmacophore feature—here focusing on HBD, HBA, and aromatic centers as defined by RDKit's FeatureFactory—was matched within a 1.5Å threshold. While this is not a strict pharmacophore

Table 21: Chemical validity of generated molecules for the natural product (NP) hopping task. Shown are the percentages of generated molecules that pass each chemical validity check. These checks operate on the molecular graph and do not involve 3D conformations. The last column reports the percentage that passed all checks.

| NP | Method | % Gen. | % Sanit. | % H-comp. | % No Rad. | % Conn. | % Chem. |
|----|--------|--------|----------|-----------|-----------|---------|---------|
| NP1 | Interpolate–integrate | 81.6 | 81.6 | 81.6 | 81.6 | 81.6 | 81.6 |
|     | Replacement–guidance | 68.2 | 68.2 | 68.2 | 68.2 | 68.2 | 68.2 |
| NP2 | Interpolate–integrate | 100.0 | 96.6 | 96.6 | 96.6 | 96.6 | 96.7 |
|     | Replacement–guidance | 92.6 | 92.6 | 92.6 | 92.6 | 92.6 | 92.6 |
| NP3 | Interpolate–integrate | 100.0 | 68.9 | 68.9 | 68.9 | 68.9 | 69.0 |
|     | Replacement–guidance | 67.6 | 67.6 | 67.6 | 67.6 | 67.6 | 67.6 |

Table 22: Physical validity of generated molecules for the natural product (NP) hopping task. Shown are the percentages of generated molecules that pass each intramolecular PoseBusters check. The final column shows the percentage passing all physical tests.

| NP | Method | % Gen. | Bond Len. | Bond Ang. | Steric Clash | Ring Flat. | DB Flat. | Energy | % Phys. |
|----|--------|--------|-----------|-----------|--------------|------------|----------|--------|---------|
| NP1 | Interpolate–integrate | 81.6 | 81.4 | 81.6 | 72.3 | 81.6 | 81.0 | 81.3 | 71.6 |
|     | Replacement–guidance | 68.2 | 68.0 | 68.2 | 61.2 | 68.2 | 67.6 | 68.0 | 60.5 |
| NP2 | Interpolate–integrate | 96.7 | 96.5 | 96.6 | 94.0 | 96.6 | 96.5 | 96.3 | 93.6 |
|     | Replacement–guidance | 92.6 | 92.5 | 92.5 | 88.4 | 92.6 | 92.3 | 92.0 | 88.0 |
| NP3 | Interpolate–integrate | 69.0 | 68.3 | 68.8 | 65.1 | 68.9 | 67.7 | 67.8 | 62.8 |
|     | Replacement–guidance | 67.6 | 67.6 | 67.5 | 65.3 | 67.6 | 66.9 | 67.1 | 64.4 |

recovery metric, it allows us to examine which reference features are most consistently reconstructed in high-quality analogues.

Figure 10 presents the per-feature match fractions for each target. Spheres represent the reference pharmacophore features, and their color encodes the recovery rate across the selected molecules: green indicates features recovered in most analogues, yellow indicates moderate recovery, and red indicates rare recovery. In each subpanel, **Top** rows display the recovery rate for the ten highest-ranked analogues by pharmacophore similarity, while **Bottom** rows show rates for all filtered analogues meeting the SA $\leq 4.5$ threshold.

For example, in NP1 (top row, left), most pharmacophoric features are green or yellow, showing high retention in top-ranked molecules. In contrast, the bottom row for NP3 contains more red features, indicating less consistent recovery.

The best-performing molecules recover nearly all pharmacophoric features for NP0 and NP1, including those that were inconsistently reproduced in the general population. This suggests that the model is capable of high-fidelity reconstruction when guided by appropriate selection criteria.

In contrast, the broader set of generated molecules shows greater variability in feature retention, particularly for features located on the molecular periphery. These features are more flexible and less scaffold-constrained, making them harder to consistently reproduce—despite their potential importance for binding interactions.

### E.8 PERFORMANCE OF MOLSNAPPER BASELINE

For the NP hopping task, we also evaluated MolSnapper in ligand-only mode (clash rate = 0) using the NP's HBA and HBD features as input as a secondary baseline to understand how a method conditioned on discrete pharmacophore points would perform. A limitation of this baseline is that it does not explicitly place hydrogens. The results for each natural product target are presented in Table 24.

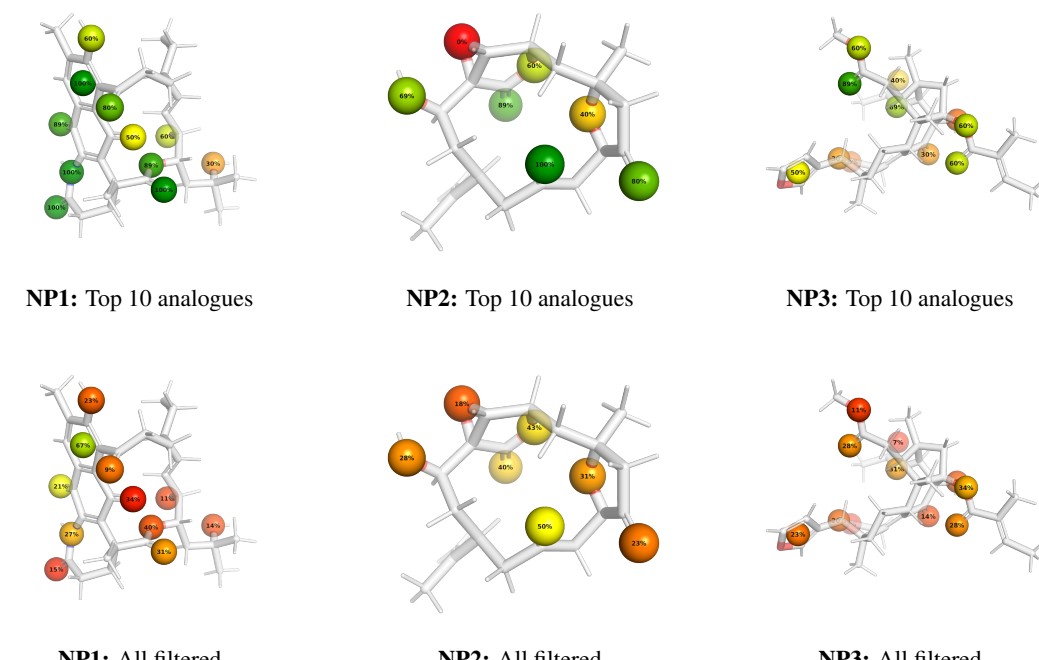

Figure 10: Pharmacophore feature retention for NP1, NP2, and NP3. **Top:** top 10 analogues ranked by pharmacophore similarity to the reference NP. **Bottom:** all filtered analogues with SA ≤ 4.5. Dummy atoms are colored by recovery rate. Reference molecules omitted for clarity.

We hypothesize that MolSnapper's performance is limited in this context due to the nature of its conditioning mechanism. MolSnapper is designed to strictly satisfy a set of discrete, fixed pharmacophore points, which acts as a "hard" constraint on the generation. While this is effective for tasks with a few key constraints, providing a dense set of points derived from a large, complex natural product may be overly restrictive, making it difficult for the model to generate novel and synthetically accessible scaffolds that satisfy all points simultaneously. This highlights the challenge of the NP hopping task and the effectiveness of conditioning strategies that allow flexibility and capture more holistic 3D information.

Table 23: Performance summary for the MolSnapper baseline on the NP hopping tasks after xTB geometry relaxation and ESP alignment. The table reports success rate (percentage of 2,500 samples passing validity and SA ≤ 4.5 filters), SA score, and top-10 similarity scores.

| NP | Method | Success Rate ↑ | SA Score ↓ | ESP sim (top 10) ↑ | Pharma sim (top 10) ↑ |
|-----|------------|----------------|-----------------|--------------------|-----------------------|
| NP1 | MolSnapper | 0.32% | $4.23 \pm 0.36$ | $0.32 \pm 0.04$ | $0.07 \pm 0.05$ |
| NP2 | MolSnapper | 0.16% | $4.22 \pm 0.39$ | $0.46 \pm 0.1$ | $0.09 \pm 0.02$ |
| NP3 | MolSnapper | 0.32% | $4.06 \pm 0.43$ | $0.22 \pm 0.07$ | $0.06 \pm 0.01$ |

Table 24: Performance summary for the MolSnapper baseline on the NP hopping tasks after xTB geometry relaxation and ESP alignment.

| NP | Method | Valid ↑ | SA Score ↓ | ESP sim (top 10) ↑ | Pharma sim (top 10) ↑ |
|-----|------------|----------|-----------------|--------------------|-----------------------|
| NP1 | MolSnapper | 26.24% | $6.43 \pm 0.78$ | $0.51 \pm 0.02$ | $0.16 \pm 0.02$ |
| NP2 | MolSnapper | 19.88 % | $6.84 \pm 0.79$ | $0.63 \pm 0.02$ | $0.19 \pm 0.01$ |
| NP3 | MolSnapper | 14.20 % | $6.32 \pm 0.85$ | $0.46 \pm 0.02$ | $0.14 \pm 0.01$ |

## F  BIOISOSTERIC FRAGMENT MERGING: EXTENDED EVALUATION

This section provides a detailed evaluation of the EV-D68 3C bioisosteric fragment-merging task. We assess molecule quality across synthetic accessibility, conformational stability, chemical validity, and performance against relevant baselines.

### F.1  EXPERIMENTAL DETAILS

**Data and Setups.**  The input data, sourced from the Fragalysis platform[3], consisted of 13 experimentally resolved fragment-bound structures previously curated by SHEPHERD (Adams et al., 2025). Across all setups, generated molecules ranged from 50 to 89 atoms. From this set, the protein conformation from the `D68EV3CPROA-x1071_0A_bound` complex was randomly selected and used for all subsequent docking simulations.

MULTI-FRAGMENT REFERENCE.  For this setup, we created three distinct composite inputs, each by combining three of the known fragment ligands. We generated 400 candidate molecules for each of the three combinations (1,200 total per method). To allow for the generation of scaffolds larger than the combined fragment set, the input for *Replacement-guidance* was padded with additional atoms. Conversely, for *Interpolate-integrate*, the target size was restricted to the exact atom count of the input fragments due to the fixed dimensionality of the interpolation trajectory. This selection was subject to spatial constraints to ensure physical realism: the sampled atoms were kept at least 1 Å apart.

RANDOM ATOM SEEDING.  The conditioning profile was created by randomly selecting a subset of pharmacophore features from the full fragment library. This selection was subject to spatial constraints to ensure physical realism: hydrogen bond features were kept at least 1 Å apart, and aromatic centers at least 2 Å apart. The remaining atoms for the seed were sampled from the broader pool of fragment-derived structures with a minimum distance of 1 Å between all atoms. For this setup, we generated 1,000 candidate ligands per method.

FULL INTERACTION PROFILE.  For the manually curated setup, we adopted the 27-feature pharmacophore profile reported in the SHEPHERD paper (Adams et al., 2025). This profile, which was originally extracted from Fragalysis annotations, represents the aggregate interaction features of the fragment ensemble. (The SHEPHERD baseline also used an averaged ESP field, which our methods do not require). To use this profile as a seed for our generative models, we first converted the 27 abstract features into a concrete atomic representation (an SDF file) by instantiating specific atoms (e.g., a Nitrogen for an HBA) at the given coordinates. This 34-atom set formed the core of our seed. To build a full-sized seed matching our generation range (50–89 atoms), we supplemented these 34 atoms with additional "filler" atoms. These filler atoms were randomly sampled from the original set of 13 fragments, ensuring a minimum distance of 1 Å between all atoms to maintain physical realism. We generated 1,000 candidate ligands per method using this full, supplemented seed.

**Generation Parameters.**  Across all setups, generated molecules ranged from 50 to 89 atoms. We used our standard conditioning parameters: *Interpolate-Integrate* was run with a restart time of $\tau = 0.75$, and *Replacement Guidance* was run with a relaxation time of $t_{\text{relax}} = 1.0$ without conditioning the bonds.

**Evaluation Pipeline.**  All candidate molecules in this section were evaluated using a standardized pipeline. Following generation, each ligand was subject to: (1) geometry relaxation using xTB, (2) alignment to the reference fragment constellation based on electrostatic potential (ESP), and (3) in-pocket energy minimization and scoring using `AutoDock Vina` with a 5 Å buffer.

**Baseline Comparability.**  Existing baselines (e.g., MOLSNAPPER, DIFFSBDD, SHEPHERD) cannot operate directly on raw fragment collections or sparse 3D feature clouds. They require a single, well-defined input structure or pharmacophore profile as conditioning. For this reason, they can be evaluated only under the manually curated *Full Interaction Profile* setup, where the fragment

---

[3] `https://fragalysis.diamond.ac.uk/viewer/react/landing`

information is consolidated into one unified profile. It is not clear how these baselines could be applied to the other two setups—*Multi-fragment Reference* and *Random Atom Seeding*—and therefore they cannot be fairly compared in the automated setups.

## F.2 Comparison of Conditioning Setups (Unfiltered)

We first compared the unfiltered performance of our two methods across the three conditioning setups to understand their inherent properties. Table 25 summarizes the results for all generated molecules that passed initial PoseBusters validity checks, prior to applying the synthetic accessibility filter. This unfiltered analysis reveals the inherent tendencies of our two strategies. *Interpolate–integrate* consistently generates a higher percentage of geometrically valid molecules. In contrast, the molecules generated by *Replacement-guidance*, while fewer in number, tend to have better SA scores and stronger Vina scores. This highlights a trade-off between generating a high quantity of valid structures versus a higher quality of drug-like candidates.

Table 25: Performance comparison of *Interpolate–Integrate* and *Replacement Guidance* across three input conditioning setups for the EV-D68 3C merging task. We report PoseBusters validity, clash rate (fraction of molecules with zero steric clashes), SA and top 10 Vina scores.

| Conditioning setup | Method | Valid (%)↑ | No clash (%)↑ | SA ↓ | Vina top10 ↓ |
|---|---|---|---|---|---|
| Multi-fragment ref. | Interpolate–integrate | 59.3% | 47.2 | $4.65 \pm 0.96$ | $-6.00 \pm 0.36$ |
| | Replacement–guidance | 36.3% | 26.8 | $3.57 \pm 0.85$ | $-5.74 \pm 0.22$ |
| Random atom seeding | Interpolate–integrate | 41.0% | 31.2 | $5.18 \pm 1.17$ | $-5.45 \pm 0.20$ |
| | Replacement–guidance | 28.1% | 20.7 | $3.84 \pm 0.81$ | $-6.00 \pm 0.31$ |
| Full interaction profile | Interpolate–integrate | **38.1%** | 30.2 | $4.83 \pm 1.17$ | $-5.52 \pm 0.18$ |
| | Replacement–guidance | 31.9% | 25.4% | $3.47 \pm 0.70$ | $-6.70 \pm 0.39$ |

## F.3 Detailed Comparison on the Full Interaction Profile

This section provides a deeper analysis on the manually curated 'Full Interaction Profile' setup, which allows for direct comparison against multiple external baselines.

### F.3.1 Baseline Methods

This section benchmarks our methods against the most relevant existing approaches. It is important to note that our task of flexible, abstract bioisosteric merging is novel, and these baselines are not designed for this exact scenario. SHEPHERD is designed for a similar task but requires precomputed grids, while MOLSNAPPER and DIFFSBDD are atom-preserving linkers/inpainters. They are, however, the closest available methods for comparison.

Although our methods are ligand-only, we also benchmark against pocket-conditioned SBDD methods in this section, as our input fragment poses contain implicit information about the binding site.

We evaluated three distinct baselines:

- **SHEPHERD: Adams et al. (2025)** The primary baseline, conditioned on the interaction profile and ESP in a ligand-only setting. It is also designed for bioisosteric merging.

- **MolSnapper: Ziv et al. (2025)** An atom-preserving conditioning mechanism. We evaluated MolSnapper in two settings: a protein-blind mode ('clash-rate=0') and a protein-aware mode ('clash-rate=0.1') using the explicit protein structure.

- **DiffSBDD: Schneuing et al. (2024)** An atom-preserving inpainter. We used the official inpainting script with checkpoints pre-trained on the CrossDocked dataset. As a structure-based method, it conditions on the protein pocket geometry while using the provided pharmacophore points as a reference context to guide the inpainting.

To adapt the atom-based baselines (MolSnapper, DiffSBDD, and ours), we converted the 27-feature abstract profile into the same atom SDF file described in Section F.1. MolSnapper, DiffSBDD were given *only* the SDF file as input (in addition to pocket info)

Both MolSnapper and DiffSBDD do not explicitly generate hydrogens. To account for their limitations, we generated 1,000 molecules for each within a heavy atom range of 36-46. For DiffSBDD, as no fully connected molecules were produced, we selected the largest resulting fragment for evaluation.

### F.3.2 VALIDITY AND FILTERING PIPELINE

Table 26 details the validity rate of the methods at each stage of our filtering pipeline for the 'Full Interaction Profile' setup. While *Interpolate–integrate* shows high initial validity, *Replacement-guidance* ultimately produces a greater final yield of candidates passing all filters.

Table 26: Validity assessment of generated molecules for the bioisosteric merging task for EV-D68 3C (ShEPhERD profile setup). Each method generated 1000 molecules. We report the percentage of molecules that (1) are valid according to RDKit parsing, (2) pass PoseBusters (PB) geometry checks prior to xTB relaxation, (3) pass all PoseBusters checks after xTB geometry relaxation and ESP alignment, and (4) additionally have a synthetic accessibility (SA) score $< 4$.

| Method | Valid (RDKit) | PB Valid | xTB + PB Valid | xTB + PB Valid + SA $< 4$ |
|---|---|---|---|---|
| DiffSBDD (largest frag) | 99.8% | 0% | 43.1% | 0% |
| MolSnapper (ligand-based) | 31.1% | 0% | 17.6% | 0% |
| MolSnapper (clash-rate = 0.1) | 30.4% | 0% | 14.5% | 0% |
| ShEPhERD | 45.1% | 29.9% | 33.8% | 10.8% |
| Interpolate–integrate | 52.2% | 32% | 39.5% | 8.7% |
| Replacement–guidance | 39.7% | 21.8% | 31.9% | 23.5% |

### F.3.3 PERFORMANCE COMPARISON

Table 27 summarizes the unfiltered performance of our methods against the baselines. The results highlight that the conditioning mechanisms of DiffSBDD and MolSnapper are less suited for this bioisosteric merging task, yielding molecules with poor SA and Vina scores. Among the top-performing methods, *Interpolate-integrate*, *Replacement-guidance*, and ShEPhERD all achieve comparable performance in Vina scores and in generating valid, clash-free molecules. However, they exhibit distinct strengths in other metrics: *Replacement-guidance* produces candidates with the best synthetic accessibility (SA score of 3.47), while ShEPhERD achieves the highest pharmacophore similarity.

Table 27: Performance summary for molecular generation in the EV-D68 3C bioisosteric merging task (ShEPhERD profile setup), after xTB geometry relaxation and ESP alignment. We report PoseBusters validity, clash rate (fraction of molecules with zero steric clashes), SA score, top-10 pharmacophore similarity and top 10 Vina scores.

| Target | Method | Valid ↑ | No clash ↑ | SA ↓ | Pharma sim (top 10) ↑ | Vina (top 10) ↓ |
|---|---|---|---|---|---|---|
| EV-D68 3C | DiffSBDD (largest frag) | 43.1% | 38.0% | $6.91 \pm 0.53$ | $0.1 \pm 0.01$ | $-4.35 \pm 0.35$ |
| | MolSnapper (ligand-based) | 17.6% | 16.2% | $7.04 \pm 0.54$ | $0.15 \pm 0.03$ | $-4.56 \pm 0.12$ |
| | MolSnapper (clash-rate = 0.1) | 14.5% | 13.9% | $6.91 \pm 0.53$ | $0.11 \pm 0.01$ | $-4.99 \pm 0.42$ |
| | ShEPhERD | 33.8% | 29.8% | $4.45 \pm 0.99$ | $0.3 \pm 0.02$ | $-6.74 \pm 0.29$ |
| | Interpolate–integrate | 39.5% | 30.8% | $4.96 \pm 1.17$ | $0.19 \pm 0.01$ | $-5.95 \pm 0.39$ |
| | Replacement–guidance | 31.9% | 25.4% | $3.47 \pm 0.70$ | $0.23 \pm 0.02$ | $-6.70 \pm 0.39$ |

Figure 11 shows the distribution of Vina scores for all generated molecules. Panel (a) displays the full range, while panel (b) visualizes the same set of scores but truncated at a Vina score of 5 for clarity, removing only extreme outliers.

Figure 13 provides a detailed comparison of the structural similarity characteristics captured by each method. Panel (a) measures ESP and pharmacophore similarity to the merged fragment template af-

Table 28: Molecular property statistics for molecular generation in the EV-D68 3C bioisosteric merging task (ShEPhERD profile setup), after xTB geometry relaxation and ESP alignment. We report molecular weight, Lipinski violations, logP, QED, number of rings, and SMILES uniqueness.

| Method | Weight [Da] | Lipinski R5 | logP | QED | # Rings | % Unique SMILES |
|---|---|---|---|---|---|---|
| DiffSBDD (largest frag) | $229.02 \pm 43.90$ | $4.97 \pm 0.21$ | $0.61 \pm 0.95$ | $0.24 \pm 0.10$ | $2.06 \pm 0.91$ | 33.7% |
| MolSnapper (ligand-based) | $207.29 \pm 24.51$ | $4.94 \pm 0.23$ | $-0.30 \pm 1.05$ | $0.29 \pm 0.10$ | $3.23 \pm 0.90$ | 48.6% |
| MolSnapper (clash-rate = 0.1) | $214.27 \pm 44.65$ | $4.93 \pm 0.26$ | $-0.42 \pm 1.17$ | $0.24 \pm 0.08$ | $3.21 \pm 0.99$ | 38.9% |
| ShEPhERD | $474.14 \pm 37.92$ | $4.32 \pm 0.75$ | $3.79 \pm 1.71$ | $0.33 \pm 0.11$ | $5.37 \pm 1.25$ | 100% |
| Interpolate–Integrate | $472.09 \pm 48.47$ | $3.88 \pm 1.28$ | $1.48 \pm 2.66$ | $0.31 \pm 0.16$ | $4.08 \pm 1.35$ | 100% |
| Replacement–Guidance | $503.52 \pm 51.09$ | $4.04 \pm 0.87$ | $4.09 \pm 1.52$ | $0.26 \pm 0.11$ | $4.68 \pm 1.04$ | 100% |

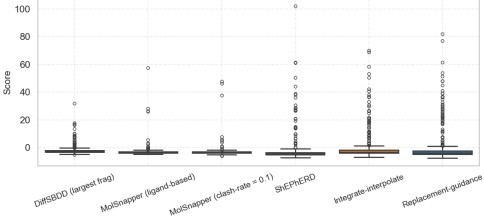

**(a)** Vina score distribution for all generated molecules (full range).

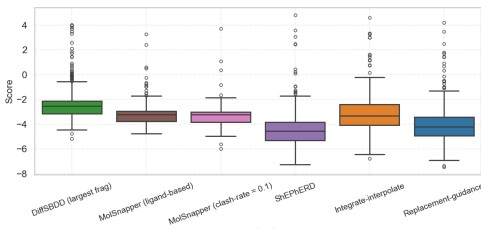

**(b)** Same Vina scores, truncated at a score of 5 to improve visualization.

Figure 11: Docking performance on the EV-D68 3C fragment-merging benchmark. Lower Vina scores are better.

ter xTB relaxation and ESP alignment, capturing how well the generated molecules preserve the electrostatic and shape complementarity implied by the input fragments. Panel (b) shows the pharmacophore similarity to the full target profile of the docked molecules, reflecting whether the final poses recover the key 3D interaction features expected at the binding site.

Finally, Figure 12 characterizes pairwise chemical diversity, synthetic accessibility, and conformational stability. For DIFFSBDD and MOLSNAPPER, RMSD could not be computed because the generated structures changed graph topology during relaxation, which prevented a consistent atom–atom mapping between the pre- and post-relaxed geometries when using the SHEPHERD-SCORE RMSD code. The pairwise Tanimoto distributions confirm that *Interpolate–integrate* and *Replacement-guidance* explore a chemically diverse set of merges. *Replacement-guidance* yields the most synthetically accessible candidates (lowest SA scores), while *Interpolate–integrate* exhibits the lowest RMSD between xTB-relaxed and original geometries, indicating the most geometrically stable poses. Together, these results echo the trends seen in the NP-hopping task: *Replacement-guidance* favors diverse, synthesizable molecules, whereas *Interpolate–integrate* optimizes more aggressively for geometric regularity; SHEPHERD sits between them, offering the strongest pharmacophore recovery.

### F.3.4 ANALYSIS OF CONDITIONING BEHAVIOR

As shown in Table 26, the proportion of fully valid and synthesizable molecules on the 27-feature abstract consensus profile highlights how different conditioning strategies respond respond to this specified manually curated input derived from the underlying fragment set.

For the *Full Interaction Profile* setup, the methods (MOLSNAPPER, DIFFSBDD and ours) received the same conditioning input: an SDF file representing the 27-feature abstract profile, instantiated as a set of atoms to serve as the conditioning seed. The divergent outcomes stem from a fundamental difference in how each method handles these constraints:

Methods like MOLSNAPPER and DIFFSBDD are designed as atom-preserving linkers or inpainters. Their conditioning mechanisms function as "hard constraints," enforcing that the generated molecule contain the exact seed atoms in the output. This approach is effective for tasks like linking two known fragments, growing a fragment, or satisfying constraints from a single known ligand. How-

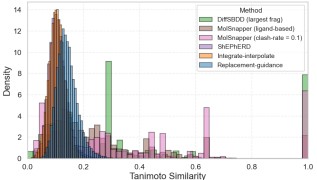

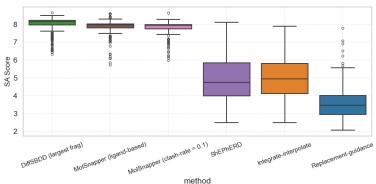

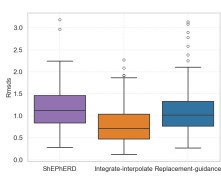

**(a)** Pairwise Tanimoto similarity between generated molecules.

**(b)** SA score distribution across methods.

**(c)** RMSD between xTB-relaxed and aligned conformers vs. original 3D geometries.

Figure 12: Diversity, synthetic accessibility and conformational stability for the EV-D68 3C fragment-merging benchmark.

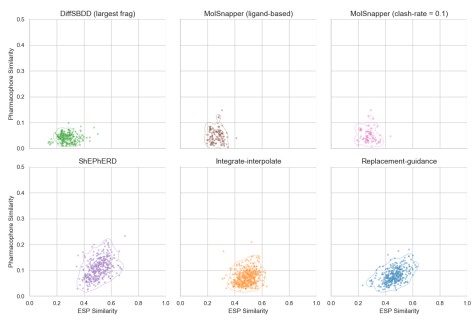

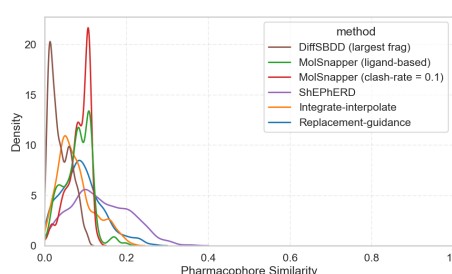

**(a)** ESP vs. pharmacophore similarity to the combined fragments after ESP alignment.

**(b)** Pharmacophore similarity density to the target profile after docking.

Figure 13: Similarity to the reference ligand for the EV-D68 3C fragment-merging task (27-feature profile).

ever, the 27-feature profile is a holistic, abstract consensus of many fragments, not a single, physically plausible molecule. It is therefore chemically challenging to satisfy all 27 dense atomic constraints simultaneously while also forming a valid, connected, low-energy molecule. As shown in Figure 14 (a) and (b), this hard-constraint approach struggles to produce a single, coherent structure from this abstract input, resulting in disconnected fragments and invalid chemistry.

In contrast, our conditioning strategies are designed to handle such abstract, bioisosteric tasks. *Replacement-guidance* uses a "relaxation" mechanism that uses the dense profile as a strong guide during the generative flow, but then releases the hard constraint in the final integration steps. This allows the model to form a final, coherent molecule that is chemically valid and bioisosteric to the abstract profile, rather than attempting to be identical to it (Figure 14 (c)). Similarly, *Interpolate-integrate* treats the input as a "soft" global seed rather than enforcing its exact atomic identity. While SHEPHERD also achieves success by using a more holistic conditioning mechanism (aggregate ESP and pharmacophore grids).

## F.4 DETAILED POSEBUSTERS VALIDITY

Tables 29, 30, and 31 provide a detailed breakdown of molecular validity for the fragment merging task. Table 29 reports overall generation success and the percentage of molecules passing both chemical and physical filters. Table 30 decomposes chemical validity into sanitization, hydrogen compatibility, valence, and connectivity checks. Table 31 presents results from the PoseBusters suite, including bond lengths, angles, steric clashes, and energetic plausibility after xTB relaxation.

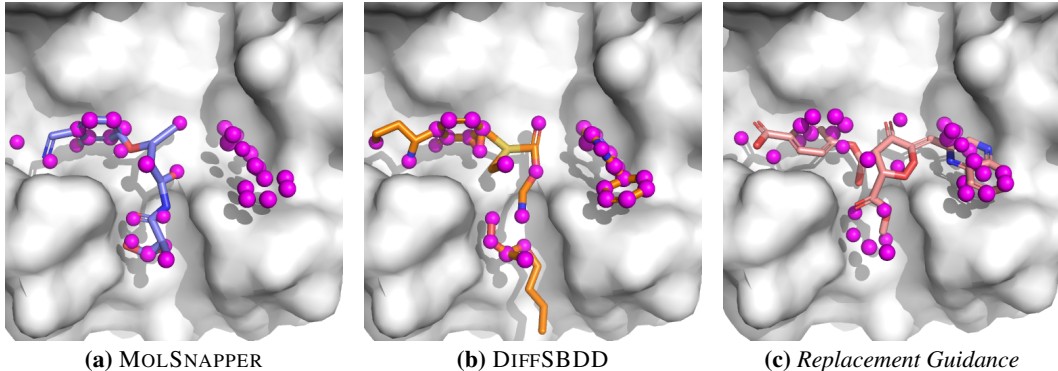

**(a)** MOLSNAPPER      **(b)** DIFFSBDD      **(c)** *Replacement Guidance*

Figure 14: Visualizing the behavior of different conditioning methods on the 27-feature curated profile. Magenta spheres represent the conditioning points. **(a, b)** MOLSNAPPER and DIFFSBDD enforce them rigidly and fail to produce a connected molecule. **(c)** Our *Replacement Guidance* relaxes the constraints and yields a connected bioisosteric molecule.

Table 29: Validity of the generated and ground truth molecules for the bioisosteric fragment merging task for EV-D68 3C (ShEPhERD profile setup). The table contains the total number of molecules that was to be generated or that successfully passed the XTB process and alignment and the proportions of molecules that pass all the chemical validity test, all physical validity tests. The last columns is the percentage of molecules that pass all the tests together.

| Method | % Generated | % Chemical | % Physical | % Valid |
|---|---|---|---|---|
| Interpolate–integrate | 42.4 | 42.4 | 39.5 | 39.5 |
| Replacement–guidance | 35.3 | 35.3 | 31.9 | 31.9 |

Table 30: Components of the chemical validity of the generated molecules for the bioisosteric fragment merging task for EV-D68 3C (ShEPhERD profile setup). Each entry shows the percentage of molecules passing individual chemical validity checks. The final column (**Chemical**) indicates the proportion of molecules that pass *all* checks. These tests assess only the molecular graph (e.g., sanitization, radicals, hydrogen completion), not the 3D conformation.

| Method | % Gen. | % Sanit. | % H-comp. | % No Rad. | % Conn. | % Chem. |
|---|---|---|---|---|---|---|
| Interpolate–integrate | 42.4 | 42.3 | 42.3 | 42.3 | 42.3 | 42.4 |
| Replacement–guidance | 35.3 | 35.2 | 35.2 | 35.2 | 35.2 | 35.3 |

Table 31: Components of the physical validity of the generated molecules for the bioisosteric fragment merging task for EV-D68 3C (ShEPhERD profile setup). The numbers shown are the percentages of the molecules that pass each of the intramolecular PoseBusters tests. The last column is the percentage of molecules that pass all of the intramolecular tests. The physical tests check the 3D conformations of the generated molecules.

| Method | % Gen. | % Bond Len. | % Bond Ang. | % Clash | % Arom. Flat. | % DB Flat. | % Int. Energy | % Physical |
|---|---|---|---|---|---|---|---|---|
| Interpolate–integrate | 42.4 | 41.9 | 42.2 | 41.4 | 42.3 | 41.4 | 41.2 | 39.5 |
| Replacement–guidance | 35.3 | 34.8 | 34.9 | 32.6 | 35.2 | 34.9 | 34.8 | 31.9 |

# G  Pharmacophore-Merging on SARS-CoV-2 M~PRO~: Extended Evaluation

This section provides a detailed evaluation of the SARS-CoV-2 $M_{pro}$ pharmacophore-merging task. We assess the quality of generated molecules by comparing their predicted binding affinity against a strong baseline of larger, known binders for the same target.

## G.1  Experimental Details

**Datasets.**   The data for this task was curated from the Fragalysis platform[4]. We created two distinct sets:

- **Conditioning Set:** A set of 81 small, non-covalent fragments (MW < 300 Da) from the initial Diamond/XChem screen.
- **Baseline Set:** A set of 1,426 larger, known binders (MW > 300 Da) for $M_{pro}$. This set serves as a challenging, real-world baseline for evaluating the predicted binding affinity of our generated molecules.

The full names for the conditioning set fragments, taken directly from the input SDF file, are listed below:

- `5rfp_A_404_1_SARS2_MproA-y0118+A+410+1__LIG`
- `7gf9_A_404_1_SARS2_MproA-y0118+A+410+1__LIG`
- `5rgi_A_404_1_SARS2_MproA-y0118+A+410+1__LIG`
- `5reu_A_404_1_SARS2_MproA-y0118+A+410+1__LIG`
- `7gfv_A_404_1_SARS2_MproA-y0118+A+410+1__LIG`
- `5rf1_A_404_1_SARS2_MproA-y0118+A+410+1__LIG`
- `5r81_A_1001_1_SARS2_MproA-y0118+A+410+1__LIG`
- `5rfh_A_404_1_SARS2_MproA-y0118+A+410+1__LIG`
- `5rgm_A_404_1_SARS2_MproA-y0118+A+410+1__LIG`
- `5rfq_A_404_1_SARS2_MproA-y0118+A+410+1__LIG`
- `5rek_A_404_1_SARS2_MproA-y0118+A+410+1__LIG`
- `5rfw_A_405_1_SARS2_MproA-y0118+A+410+1__LIG`
- `7gcm_A_403_1_SARS2_MproA-y0118+A+410+1__LIG`
- `7gdr_A_406_1_SARS2_MproA-y0118+A+410+1__LIG`
- `5rg3_A_405_1_SARS2_MproA-y0118+A+410+1__LIG`
- `5r82_A_1001_1_SARS2_MproA-y0118+A+410+1__LIG`
- `5rer_A_404_1_SARS2_MproA-y0118+A+410+1__LIG`
- `5rha_A_1001_1_SARS2_MproA-y0118+A+410+1__LIG`
- `5rfm_A_404_1_SARS2_MproA-y0118+A+410+1__LIG`
- `7gc6_A_405_1_SARS2_MproA-y0118+A+410+1__LIG`
- `7gfy_A_404_1_SARS2_MproA-y0118+A+410+1__LIG`
- `7gel_A_404_1_SARS2_MproA-y0118+A+410+1__LIG`
- `7ghi_A_405_1_SARS2_MproA-y0118+A+410+1__LIG`
- `5r80_A_404_1_SARS2_MproA-y0118+A+410+1__LIG`
- `5rg0_A_405_1_SARS2_MproA-y0118+A+410+1__LIG`
- `7gc8_A_404_1_SARS2_MproA-y0118+A+410+1__LIG`
- `5rhd_A_408_1_SARS2_MproA-y0118+A+410+1__LIG`
- `7gb3_A_406_1_SARS2_MproA-y0118+A+410+1__LIG`

---

[4]`https://fragalysis.diamond.ac.uk/viewer/react/preview/target/CoV-Mpro/tas/lb32627-272`

- 5rgh_A_404_1_SARS2_MproA-y0118+A+410+1__LIG
- 5rfl_A_404_1_SARS2_MproA-y0118+A+410+1__LIG
- 5rg2_A_404_1_SARS2_MproA-y0118+A+410+1__LIG
- 5rgo_A_404_1_SARS2_MproA-y0118+A+410+1__LIG
- 7ghh_A_1001_1_SARS2_MproA-y0118+A+410+1__LIG
- 5reh_A_404_1_SARS2_MproA-y0118+A+410+1__LIG
- 5rft_A_404_1_SARS2_MproA-y0118+A+410+1__LIG
- 7ghg_A_1001_1_SARS2_MproA-y0118+A+410+1__LIG
- 5rez_A_404_1_SARS2_MproA-y0118+A+410+1__LIG
- 7gco_A_404_1_SARS2_MproA-y0118+A+410+1__LIG
- 5ren_A_404_1_SARS2_MproA-y0118+A+410+1__LIG
- 5r83_A_404_1_SARS2_MproA-y0118+A+410+1__LIG
- 5r84_A_1001_1_SARS2_MproA-y0118+A+410+1__LIG
- 5re4_A_404_1_SARS2_MproA-y0118+A+410+1__LIG
- 5rel_A_404_1_SARS2_MproA-y0118+A+410+1__LIG
- 5rfx_A_404_1_SARS2_MproA-y0118+A+410+1__LIG
- 5re9_A_404_1_SARS2_MproA-y0118+A+410+1__LIG
- 5res_A_404_1_SARS2_MproA-y0118+A+410+1__LIG
- 5r7z_A_404_1_SARS2_MproA-y0118+A+410+1__LIG
- 5ret_A_404_1_SARS2_MproA-y0118+A+410+1__LIG
- 5rfk_A_404_1_SARS2_MproA-y0118+A+410+1__LIG
- 5rew_A_404_1_SARS2_MproA-y0118+A+410+1__LIG
- 5rgp_A_404_1_SARS2_MproA-y0118+A+410+1__LIG
- 5rex_A_404_1_SARS2_MproA-y0118+A+410+1__LIG
- 7gcp_A_405_1_SARS2_MproA-y0118+A+410+1__LIG
- 7gcy_A_405_1_SARS2_MproA-y0118+A+410+1__LIG
- 5rfn_A_404_1_SARS2_MproA-y0118+A+410+1__LIG
- 5rfz_A_404_1_SARS2_MproA-y0118+A+410+1__LIG
- 7gha_A_1001_1_SARS2_MproA-y0118+A+410+1__LIG
- 5rgk_A_404_1_SARS2_MproA-y0118+A+410+1__LIG
- 5rfo_A_404_1_SARS2_MproA-y0118+A+410+1__LIG
- 7ghj_A_405_1_SARS2_MproA-y0118+A+410+1__LIG
- 5rfs_A_405_1_SARS2_MproA-y0118+A+410+1__LIG
- 7gba_A_404_1_SARS2_MproA-y0118+A+410+1__LIG
- 7gg3_A_404_1_SARS2_MproA-y0118+A+410+1__LIG
- 5rfy_A_404_1_SARS2_MproA-y0118+A+410+1__LIG
- 5rfe_A_404_1_SARS2_MproA-y0118+A+410+1__LIG
- 5rhf_A_405_1_SARS2_MproA-y0118+A+410+1__LIG
- 5rgl_A_404_1_SARS2_MproA-y0118+A+410+1__LIG
- 5rf6_A_404_1_SARS2_MproA-y0118+A+410+1__LIG
- 5rf7_A_404_1_SARS2_MproA-y0118+A+410+1__LIG
- 7gdq_A_406_1_SARS2_MproA-y0118+A+410+1__LIG
- 5rez_A_404_1_SARS2_MproA-y0118+A+410+1__LIG
- 7gco_A_404_1_SARS2_MproA-y0118+A+410+1__LIG
- 5ren_A_404_1_SARS2_MproA-y0118+A+410+1__LIG
- 5r83_A_404_1_SARS2_MproA-y0118+A+410+1__LIG
- 5r84_A_1001_1_SARS2_MproA-y0118+A+410+1__LIG

- `5re4_A_404_1_SARS2_MproA-y0118+A+410+1__LIG`
- `5rel_A_404_1_SARS2_MproA-y0118+A+410+1__LIG`
- `5rfx_A_404_1_SARS2_MproA-y0118+A+410+1__LIG`
- `5re9_A_404_1_SARS2_MproA-y0118+A+410+1__LIG`
- `5res_A_404_1_SARS2_MproA-y0118+A+410+1__LIG`
- `5r7z_A_404_1_SARS2_MproA-y0118+A+410+1__LIG`

**Generation and Evaluation.** We generated 2,500 candidate molecules per method, with total atom counts between 40 and 80, using our two conditioning strategies: *Interpolate-Integrate* (with a restart time of $\tau = 0.75$) and *Replacement Guidance* (with a relaxation time of $t_{\text{relax}} = 1.0$ without conditioning the bonds). The total atom count range of 40 to 80 was chosen to match the core distribution of the 1,426 known Mpro binders, as shown in Figure 15. The generation was conditioned by sampling pharmacophore features from the 81-fragment conditioning set, following the 'random atom seeding' setup (Section 4.3), where each seed was constructed by sampling atoms from the 81-fragment conditioning set: half of the atoms were randomly selected pharmacophore features, while the remaining half were non-pharmacophore atoms. The pharmacophore selection was subject to the spatial constraints described in Appendix F.1 to ensure physical realism.

For evaluation, both our generated molecules and the 1,426 baseline binders were docked using an identical `AutoDock Vina` protocol (`--minimize` mode, 5 Å buffer). For this protocol, the rigid receptor structure was taken from a randomly selected crystal structure, **SARS-a0514b**, which defines the binding pocket for all docking runs. Note that for this task we evaluated the docked geometries directly, without applying xTB relaxation or ESP alignment prior to scoring.

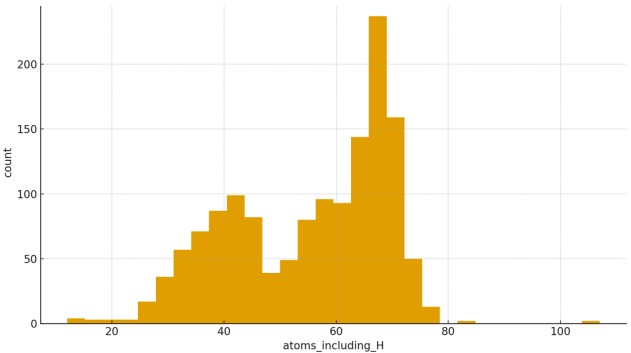

Figure 15: Distribution of total atom counts (including hydrogens) for the baseline set of 1,426 known binders for SARS-CoV-2 Mpro. The generation range of 40–80 atoms was selected to cover the main peaks of this distribution.

### G.2 Validity and Filtering Pipeline

Table 32 details the validity of our methods at each stage of the filtering pipeline. Both methods demonstrate high initial validity, but *Replacement-guidance* produces a higher final yield of candidates that pass the synthetic accessibility score filter.

Table 32: Validity assessment of generated molecules for the pharmacophore-merging task on SARS-CoV-2 $M_{\text{pro}}$. Each method generated 2,500 molecules. We report the percentage of molecules that: (1) are valid according to RDKit parsing, (2) pass all PoseBusters (PB) geometry checks, (3) pass PB and are successfully docked, and (4) additionally satisfy a synthetic accessibility (SA) score $\leq 4.5$.

| Method | RDKit Valid | PB Valid | PB + Docked | PB + Docked + SA < 4.5 |
| --- | --- | --- | --- | --- |
| Interpolate–Integrate | **63.8%** | **46.0%** | **41.1%** | 23.5% |
| Replacement–Guidance | 51.5% | 39.7% | 32.2% | **27.3%** |

## G.3 INTERACTION RECOVERY ANALYSIS WITH PROLIF

To quantitatively assess whether our conditioning strategies effectively transfer interaction patterns from the input data to the generated molecules, we performed an interaction recovery analysis. The objective of this analysis was to verify that the pharmacophoric information present in the conditioning set was successfully encoded and reproduced by the generative models. This was accomplished using the ProLIF (Protein-Ligand Interaction Fingerprints) package (Bouysset & Fiorucci, 2021) to create detailed interaction profiles for both the conditioning set and the generated molecules, allowing for a direct comparison of their interaction patterns (Errington et al., 2025). Following Errington et al. (2025), we considered only hydrogen and halogen bonds (donor and acceptor), $\pi$-stacking, cation–$\pi$ and $\pi$–cation interactions, and ionic interactions (anionic and cationic), while excluding the less specific hydrophobic interactions and Van der Waals contacts.

First, we established a reference interaction profile by characterizing the complete set of protein-ligand interactions for the 81 fragments in the conditioning set against the SARS-CoV-2 $M^{pro}$ target (SARS-a0514b). This step yielded a reference set of 20 distinct interaction types. We then evaluated the entire population of molecules generated by both the *Interpolate-Integrate* and *Replacement Guidance* methods. For each generated molecule, we used ProLIF to determine which of the 20 reference interactions it successfully reproduced.

The analysis confirmed that both conditioning methods were effective at exploring the desired interaction space. We found that the generated populations, in aggregate, were capable of reproducing the complete set of 20 reference interactions found in the original conditioning data. This result indicates that our inference-time guidance can successfully generate a set of candidates for lead generation, serving as starting points for a drug discovery campaign.

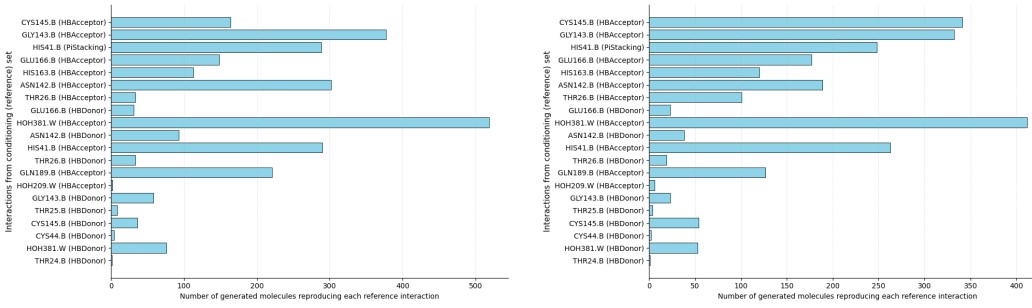

(a) Interaction recovery for *Interpolate-Integrate*.  (b) Interaction recovery for *Replacement Guidance*.

Figure 16: Histograms showing the recovery frequency of 20 reference interactions for molecules generated by (a) Interpolate-Integrate and (b) Replacement Guidance. The y-axis lists the specific protein-ligand interactions from the conditioning set of 81 $M_{pro}$ fragments, and the x-axis shows the number of generated molecules that successfully reproduced each interaction.

We subsequently performed the same interaction recovery analysis on a more stringently filtered subset of molecules, including only those that passed all PoseBusters validity checks and were successfully docked using `AutoDock Vina` in `--minimize` mode, with a 5 Å buffer. This second analysis, shown in Figure 17, serves to confirm that the models reproduce the target interactions not just in any conformation, but specifically in poses that are both chemically plausible and have a favorable predicted binding geometry. The results from this filtered subset further validate the ability of our methods to generate high-quality candidates that retain the essential binding features of the conditioning data.

## G.4 PERFORMANCE ON UNFILTERED CANDIDATES

Table 33 summarizes the final performance metrics for all valid generated molecules, compared against the baseline set of known binders.

## G.5 DETAILED POSEBUSTERS VALIDITY

Table 35 summarizes the validity metrics for the pharmacophore-merging task on SARS-CoV-2 $M_{pro}$. We report the percentage of generated molecules that pass all chemical validity checks (e.g., sanitization, valence), physical plausibility checks via PoseBusters (after xTB relaxation and ESP alignment), and the intersection of both. *Interpolate–Integrate* yields higher overall validity, while *Replacement Guidance* shows competitive chemical correctness and stronger SA-filtered outputs (see Appendix G for extended analysis).

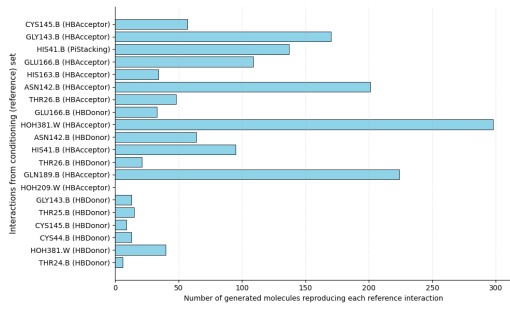 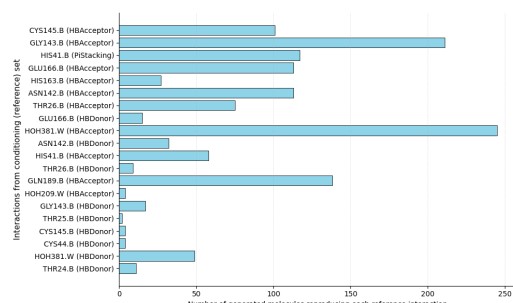

**(a)** Interaction recovery for *Interpolate-Integrate*.     **(b)** Interaction recovery for *Replacement Guidance*.

Figure 17: Histograms showing the recovery frequency of 20 reference interactions for PoseBusters Valid and docked molecules generated by (a) Interpolate-Integrate and (b) Replacement Guidance. The y-axis lists the specific protein-ligand interactions from the conditioning set of 81 $M_{pro}$ fragments, and the x-axis shows the number of generated molecules that successfully reproduced each interaction.

Table 33: Performance summary for the SARS-CoV-2 $M_{pro}$ pharmacophore-merging task. The table reports overall PoseBusters validity (**Valid**), the percentage of initial samples that are valid and clash-free (**No clash**), the average unfiltered SA score, and the top-10 Vina docking score.

| Target | Method | Valid (%) ↑ | No clash (%) ↑ | SA ↓ | Vina top10 ↓ |
|---|---|---|---|---|---|
| | Interpolate–integrate | **41.0%** | **24.0%** | $4.49 \pm 1.20$ | $-6.67 \pm 0.13$ |
| SARS-CoV-2 $M_{pro}$ | Replacement–guidance | 32.2% | 13.6% | $\mathbf{3.61 \pm 0.90}$ | $\mathbf{-7.70 \pm 0.37}$ |
| | Known Binders (Baseline) | 100% | 84.1% | $3.70 \pm 0.97$ | $-10.03 \pm 0.12$ |

Table 34: Molecular property statistics for the SARS-CoV-2 $M_{pro}$ pharmacophore-merging task, computed over PoseBusters-valid molecules. We report molecular weight, Lipinski rule-of-five violations, logP, QED, number of rings, and SMILES uniqueness.

| Method | Weight [Da] | Lipinski R5 | logP | QED | # Rings | % Unique SMILES |
|---|---|---|---|---|---|---|
| Interpolate–integrate | $460.47 \pm 79.92$ | $3.98 \pm 1.22$ | $2.01 \pm 2.40$ | $0.36 \pm 0.19$ | $4.53 \pm 1.51$ | 100% |
| Replacement–guidance | $507.16 \pm 97.87$ | $4.05 \pm 0.98$ | $3.10 \pm 1.72$ | $0.30 \pm 0.17$ | $4.62 \pm 1.40$ | 100% |

Table 35: Validity of the generated molecules for the pharmacophore-merging task on SARS-CoV-2 $M_{pro}$. Each method attempts 2,500 generations. We report the percentage of generated molecules that pass all chemical validity checks, all physical validity checks, and the percentage that pass both.

| Method | % Generated | % Chemical | % Physical | % Valid |
|---|---|---|---|---|
| Interpolate–Integrate | 51.3 | 51.3 | 41.0 | 41.0 |
| Replacement–Guidance | 48.0 | 48.0 | 32.2 | 32.2 |

Although our method operates in a protein-blind setting, we further assessed the spatial plausibility of generated molecules by docking them into SARS-CoV-2 $M_{pro}$ using `AutoDock Vina` in `--minimize` mode, with a 5 Å buffer centered on the reference fragment ensemble. As a post hoc sanity check, we computed the percentage of SA-filtered molecules (i.e., those passing xTB + PB checks and SA $\leq 4.5$) that do not exhibit steric clashes following docking.

# H  LIMITATIONS

Our framework, while flexible, has several limitations that provide avenues for future work.

Table 36: Components of the chemical validity of the generated molecules for the pharmacophore-merging task on SARS-CoV-2 $M_{pro}$. Each entry shows the percentage of molecules passing individual chemical validity checks. The final column indicates the percentage that pass all chemical checks.

| Method | % Gen. | % Sanit. | % H-comp. | % No Rad. | % Conn. | % Chem. |
|---|---|---|---|---|---|---|
| Interpolate–Integrate | 51.3 | 51.2 | 51.2 | 51.2 | 51.2 | 51.3 |
| Replacement–Guidance | 48.0 | 48.0 | 48.0 | 48.0 | 48.0 | 48.0 |

Table 37: Components of the physical validity of the generated molecules for the pharmacophore-merging task on SARS-CoV-2 $M_{pro}$. Each entry shows the percentage of molecules passing individual PoseBusters checks after xTB relaxation and ESP alignment. The final column shows the percentage that pass all physical checks.

| Method | % Gen. | % Bond Len. | % Bond Ang. | % Clash | % Arom. Flat. | % DB Flat. | % Int. Energy | % Physical |
|---|---|---|---|---|---|---|---|---|
| Interpolate–Integrate | 51.3 | 50.0 | 50.3 | 48.4 | 51.2 | 46.4 | 49.4 | 41.0 |
| Replacement–Guidance | 48.0 | 42.5 | 45.8 | 43.0 | 47.6 | 42.1 | 46.5 | 32.2 |

**Dependence on the Conditioning Signal.** Both conditioning strategies depend on the structure and density of the input information. When the seed is well-formed and chemically coherent (e.g., a full ligand in the NP hopping task), validity is naturally higher because the model starts close to the data manifold. Conversely, when the conditioning input is dense, abstract, or highly fragmented (e.g., many disjoint features or large merged profiles), the model must resolve more ambiguity, which can reduce validity. This reflects an inherent trade-off between flexibility of conditioning and difficulty of reconstruction.

**Fixed Dimensionality Requirement in Interpolate–Integrate.** Unlike *Replacement Guidance*, which can operate on variable-sized inputs via masking/padding, *Interpolate–Integrate* requires the seed to share the exact dimensionality of the output. This is because the interpolation formula $x_\tau = \tau\, x_1 + (1 - \tau)x_0$ operates on the global state. This makes *Interpolate–Integrate* less suitable for tasks like linking distant fragments, where the required linker size is unknown or requires dynamic padding that the interpolation path cannot easily accommodate.

**Seed Construction and Fixed Atom Count.** Our methods operate on a fixed tensor size, which requires an explicit sampling step at the beginning of inference to define the target atom count $N$ and composition (e.g., by subsampling a reference molecule or adding "filler" atoms to a fragment set). While this seed construction is automated, it is a prerequisite for generation.

**Evaluation Proxies.** Our evaluation using docking scores relies on rigid-receptor docking (e.g., AutoDock Vina) for both the bioisosteric merging and $M_{pro}$ tasks. This protocol neglects protein flexibility and induced fit. This can unfairly penalize valid binders (including known ligands) that require minor backbone adjustments, or underestimate the potential of generated bioisosteres. Therefore, the reported Vina scores should be interpreted as measures of geometric complementarity rather than definitive binding affinities.

## I  STATEMENT ON LARGE LANGUAGE MODEL (LLM) USAGE

In accordance with the conference policy, we report the use of a large language model (Google's Gemini and GPT, OpenAI) in the preparation of this manuscript. The LLM's role was strictly that of a general-purpose writing and programming assistant; it did not contribute to the core research ideation, experimental design, or the development of the proposed methods.

## J  REPRODUCIBILITY STATEMENT

To ensure reproducibility, we provide all experimental details in the main text and appendices. This includes descriptions of our datasets, evaluation metrics (Section C), and conditioning setups (Appendices E, F, and G). Our open-source code is available at `https://github.com/oxpig/cond-semla`, and all processed

data and generated molecules from our experiments are available on Zenodo at `https://doi.org/10.5281/zenodo.17521477`. The repository includes the scripts required to reproduce our main results.

