# OpenReview forum: "Interpolation-Based Conditioning of Flow Matching Models for Bioisosteric Ligand Design"
_ICLR.cc/2026/Conference — ICLR 2026 Poster_

### Official Review · Reviewer_mYeo · 2025-10-30

**Soundness:** 2
**Presentation:** 2
**Contribution:** 2
**Rating:** 4
**Confidence:** 4

**Summary:**

The authors introduce two training-free interpolation strategies that mix known reference target data into flow-matching molecular generators based on SemlaFlow. They either interpolate between the generated sample and the reference partway through FM integration, or hold parts of the reference data fixed during the integration process. This is proposed to enable scaffold hopping, bioisosteric replacement, and pharmacophore matching by incorporating information from known binders into the generation process. The approach is compared to several recent methods—MolSnapper, DiffSBDD, and perhaps most appropriately ShEPhERD—showing improved SA scores and pharmacophoric similarities/docking scores on experiments covering some intended use cases. There are details of the evaluation that appear to be missing, which makes it difficult to fully assess how the method works. Overall, it is quite a simple idea and appears to work well.

**Strengths:**

The idea is very simple and easily explained, and the paper is clearly presented.

The ability to merge across many reference pharmacophoric features in a scalable way is useful, even if in practice this is achieved by random subsampling.

The authors report superior results to ShEPhERD, a considerably more complicated method, on several shared tasks.

**Weaknesses:**

A lot of emphasis is placed on SA scores, but this method does nothing to directly control SA relative to other methods as I can see. Presumably, the improvements here are due to better characteristics of the base model (SemlaFlow), so comparisons along this axis are not particularly convincing. The method shows quite low validity, especially compared to the baseline SemlaFlow. The authors are quite loose with the definition of a pharmacophore; interpolate-integrate seems more like enforcing global similarity to the reference molecule rather than specifically matching pharmacophoric features. Regarding the “dense” pharmacophoric features in the NP problem, it seems dubious that retaining all atomic features of input molecules is useful in practice. In general, a specific set of interactions or features govern the ligand–target interaction, not all features of the ligand, as implied in the second two experiments.

**Questions:**

Why is validity generally much lower in the second two experiments than in the first or in the ablation?

All methods produce chemical graphs with extremely low similarity to the reference (Figure 7), despite injecting the NP structure. Why?

How does this compare to samples from the unperturbed SemlaFlow?

It is odd to report only Vina scores and not pharmacophoric similarity (to the target) in the bioisosteric fragment matching and SARS-CoV examples, given that pharmacophore matching is the explicit target.

It would be useful to report SMILES uniqueness for successes in each method in the experiments as well. In general, the chemical diversity of the successful identifications is not quantified.

Inpainting experiment (Table 11): the description lacks detail. Can the authors explain how this was implemented? Also, the only metrics for inpainting relate to validity, not SC similarity as in other experiments.

How were pharmacophoric features defined? Is this based on RDKit shape and color similarity? In the appendix it states: “Molecules were selected based on pharmacophore similarity ..., and each recovered reference pharmacophore feature was matched within a 1.5 Å threshold.” The definition of “hydrogen bond donors (HBD), acceptors (HBA), and aromatic centres” is not precise—how is this determined? Additionally, this is a short and somewhat arbitrary list.

It seems important to provide some indication of the quality of the geometries pre-relaxation, as xTB will resolve many issues.

---

> ### Author Response · Authors · 2025-11-21
> **Authors’ response to Reviewer mYeo (Part 1)**
>
> We thank the reviewer for their detailed feedback and for recognizing that our simple method works well and outperforms more complex methods. Below we address all questions and concerns point-by-point.
>
> 1. **SA Scores and Unconditional Quality**
>
> >A lot of emphasis is placed on SA scores, but this method does nothing to directly control SA relative to other methods as I can see...
>
> We appreciate the reviewer’s point. Our goal is not to optimize or control the SA score. Our conditioning mechanisms are specifically designed to control binding-relevant 3D features, such as pharmacophore similarity and interaction geometry, while still producing valid molecules with reasonable SA. We apply an SA filter to mimic a realistic drug-design funnel, where practitioners prioritize the best surviving candidates even when SA is not explicitly optimized.
>
> To avoid implying that SA is being guided during generation, we changed the terminology in all tables from “success rate’’ to “% Valid (SA < X)”. Moreover, we now report complete unfiltered distributions for SA alongside full distributions for Vina, ESP similarity, pharmacophore similarity, RMSD, diversity, and other molecular properties across all tasks. These additions (summarized in the General Response) make the unconditional sample quality fully transparent.
>
> 2. **Pharmacophore Definition** (W+Q7)
>
> Response: We have clarified all pharmacophoric feature definitions and extraction procedures in the revised manuscript, added a dedicated appendix section (Appendix D.), and included the exact extraction code in our repository. we now explicitly state that we use the standard RDKit FeatureFactory (based on BaseFeatures.fdef).
>
> •	Feature extraction. Conditioning features (HBD, HBA, and aromatic rings) are defined strictly using the standard RDKit FeatureFactory (BaseFeatures.fdef), computed from the reference ligands.
>
> •	Metric vs. Visuals: Metric vs. visuals. The quantitative evaluation uses the full pharmacophore similarity metric from the ShEPhERD study, which includes a broader set of directional and non-directional pharmacophoric features. For the visual illustrations in the appendix, we focus on HBD, HBA, and aromatic rings. We chose this subset because they are the primary drivers of binding specificity, making the visual assessment of "feature recovery" more interpretable than displaying the full feature set (This choice and details are also clarified in Appendix E.7.).
>
> 3. **Differences in Validity Across Tasks** (Q1,Q3)
>
> We interpret this difference as a reflection of the inherent geometric difficulty of the tasks relative to the model's training distribution. In the NP Hopping task, the seed is a valid, connected molecule, meaning the vector field starts near the familiar data manifold, which likely facilitates the generation of valid variants (validity ~60-70%). In contrast, for the Merging tasks, the seed consists of disconnected fragments or a sparse point cloud. This effectively places the model in a more challenging state, requiring it to "heal" structural disconnections to form a single cohesive molecule. We believe this represents a geometric challenge considerably more demanding than unconditional generation (where the base model achieves 93% validity).
>
> We added ablations in Appendix B.1.4 and B.2.4 showing how validity declines as the number of fragments increases or as the seed becomes more spatially inconsistent.
>
> 4. **Low Graph Similarity** (Q2)
>
> The low graph similarity arises primarily from the experimental design. In the NP hopping task, the seed structure is subsampled: the model receives only a subset of the atoms from the original natural product. As a result, the generated molecules are smaller, simpler, and structurally distinct from the full NP scaffold, which naturally reduces the graph similarity. This is a desirable behaviour as our goal is to generate novel chemotypes that preserve the 3D shape and pharmacophoric profile of the reference while differing in their 2D scaffolds.

---

> > ### Author Response · Authors · 2025-11-21
> > **Authors’ response to Reviewer mYeo (Part 2)**
> >
> > 5. **Missing Metrics** (Q4,Q5,Q8)
> >
> > We have updated the manuscript to include:
> >
> > •      Pharmacophore Similarity: For the Bioisosteric Merging task, we now explicitly report pharmacophore similarity relative to the target interaction profile (Table 3 in the main paper) and provide the full distribution in the appendix. (See our General Response for a summary of these additions.)
> >
> > •	Interaction Recovery (Mpro): We added a protein–ligand interaction analysis using ProLIF fingerprints. We found that both Interpolate–Integrate and Replacement Guidance recover the full set of interaction types present in the conditioning fragment set. These results are now included in the main paper (Section 4.4), showing that both Interpolate–Integrate and Replacement Guidance recover the full set of interaction types present in the conditioning fragment set. Appendix G.3 provides detailed histograms of the recovered interactions (Figures 16–17).
> >
> > •	Uniqueness/Diversity: We added molecular property tables, including Uniqueness, for all tasks (Tables 19, 28, and 34).
> >
> > •	Pre-relaxation geometry quality. We now report PoseBusters validity before relaxation (Tables 16, 26, 32), allowing readers to compare pre- and post-relaxation validity within the same table directly.
> >
> > 6. **Inpainting Experiment Details** (Q6)
> >
> > We have expanded and clarified the inpainting experiment description in Appendix B.2.2 (Table 13). The ablation compares Replacement Guidance to an inpainting-style baseline, where the fragment is first interpolated to time t (i.e., noised) and this noised fragment is inserted at each update step. The inpainting baseline uses the same update schedule and the same t_freet  relaxation setting as Replacement Guidance to ensure a fair comparison.
> >
> > We also specify four inpainting variants that isolate the effect of noising each component: (i) coordinates noised; atoms and bonds hard-replaced (ii) atom types noised; coordinates and bonds hard-replaced (iii) bond logits noised; coordinates and atoms hard-replaced (iv) all components noised (standard inpainting)
> >
> > We additionally report SC similarity metrics for all inpainting variants, consistent with the rest of the paper. As shown in the updated results (Main Paper §4.6), Replacement Guidance achieves higher validity (58.6% vs. 32.0%) and better SC similarity (0.55 vs. 0.44) than fully-noised inpainting. Further details are provided in the General Response.
> >
> > 7. **Regarding “dense” pharmacophoric features**
> >
> > Our use of dense, raw pharmacophoric features is not meant to imply that “more features are better,” but rather reflects a design choice aimed at automation. Operating on raw fragment data removes the need for expert curation or handcrafted pharmacophore profiles, an important benefit for ligand-based workflows where curated profiles may not be available. For the NP task, the conditioning explicitly uses only HBD, HBA, and aromatic rings, which we state clearly in the revised manuscript.
> >
> > We also do not imply in the second two experiments that every atomic feature is biologically meaningful or should be preserved. The opposite is now stated explicitly in the bioisosteric merging section: “Our goal is to assess how effectively the automated conditioning strategies can approach the performance of the manually curated profile, which represents a key challenge for real-world application.”
> > This framing clarifies that the dense inputs are evaluated to test robustness and automation

---

> > > ### Comment · Reviewer_mYeo · 2025-11-28
> > > **Response**
> > >
> > > I thank the authors for the additional details about the how the pharmacophoric features are defined and the additional results and statistics on the unfiltered results, as well the expanded ablation studies. These improve the clarity of the work and make it overall more convincing. I am therefore happy to change my recommendation to 6.

---

### Official Review · Reviewer_adpW · 2025-10-31

**Soundness:** 3
**Presentation:** 3
**Contribution:** 3
**Rating:** 6
**Confidence:** 4

**Summary:**

In this paper, the authors present two novel, training-free conditioning strategies (1) Interpolate–Integrate and (2) Replacement Guidance that enable controllable 3D molecular generation using pre-trained E(3)-equivariant flow-matching models. The authors target ligand-based drug design scenarios, where protein structures are unavailable, and the goal is to generate new molecules that preserve key pharmacophoric and shape features of known ligands or fragments.
The framework is evaluated on three meaningful and representative LBDD tasks natural product ligand hopping, bioisosteric fragment merging, and pharmacophore merging, showing improved validity, synthetic accessibility, and diversity over specialized baselines.

**Strengths:**

The presented manuscripts tackles a key area in 3D generative drug design and present an elegant solution synthetic library search and fragment merging. The manuscript is well written and easy to follow. The experiments conducted sufficiently demonstrate the key claims though some additional experiments can further help elucidate the strengths of the 2 methods.

**Weaknesses:**

- In table 1, Since the success rate is low, it would be transparent to see top-100 and top-1000 when possible as well to see how fast the similarity declines.
- P7, L325, which are these "key pharmaphoric features"? It should be specified or referenced.
- In fig 1, please also note the pharmacophoric similarity to the corresponding NP. Some reported molecules especially for NP3 appear to be significant simplification of NP3 and may lack the same pharmacological effect.

**Questions:**

it is unclear to me how pharmacophore merging is different from biosteric fragmetn merging. It appears that in both cases the fragments from fraglysis program are replaced and merged to produce a functionally similar synthetic molecule. Can the authors elaborate and explain on this point?

---

> ### Author Response · Authors · 2025-11-21
> **Authors’ response to Reviewer adpW**
>
> We thank the reviewer for the positive evaluation and for recognising both the clarity of the manuscript and the value of training-free 3D conditioning for ligand-based design. Below we address the main points.
>
> 1. **Transparency of Success Rates**
>
> We thank the reviewer for this suggestion. To provide full transparency on how similarity behaves beyond the top-10, we added Figure 2 to the main paper, which shows the entire ESP vs. pharmacophore similarity distributions for all generated molecules. As detailed in our General Response, the appendix now also includes complete unfiltered results for each task.
>
>
> 2. **Definition of Pharmacophoric Features**
>
> We thank the reviewer for highlighting this ambiguity. In the revised manuscript (p.6, line 310), we now explicitly state that pharmacophoric features are defined using the standard RDKit FeatureFactory (based on BaseFeatures.fdef), specifically hydrogen-bond donors, hydrogen-bond acceptors, and aromatic rings. We also added a dedicated appendix section (Appendix D) detailing the feature definitions and extraction protocol. For completeness, we included the exact extraction code in the accompanying repository. Further context on these revisions is provided in our General Response.
>
> 3. **Figure 1: Pharmacophore Similarity for NP Examples**
>
> We appreciate the suggestion. We now report the pharmacophore similarity in Figure 1 for each visualised molecule. The reviewer is correct that NP3 is a more challenging case; this is also reflected in the full similarity distributions added to Figure 2, where NP3 shows a broader spread compared to the other natural products.
>
> 4. **Pharmacophore Merging vs. Bioisosteric Fragment Merging**
>
> Pharmacophore merging and bioisosteric fragment merging both aim to generate molecules consistent with known binding information, but they use different types of conditioning signals.
>
> Pharmacophore merging operates on abstract interaction features (HBD/HBA/aromatic points) extracted from many fragments, so the goal is to generate molecules that match the overall interaction pattern, without preserving the geometry of any particular fragment.
>
> Bioisosteric fragment merging, in contrast, conditions on explicit 3D fragment geometry. While one of its configurations (‘Random Atom Seeding’) is the same as pharmacophore merging, the broader merging task (Section 4.3) also includes setups (such as the Multi-fragment Reference configuration) where the model is conditioned on complete fragment structures. In these cases, the model is expected to generate molecules that remain structurally close to the supplied fragments.

---

### Official Review · Reviewer_ckkR · 2025-11-01

**Soundness:** 3
**Presentation:** 3
**Contribution:** 3
**Rating:** 6
**Confidence:** 4

**Summary:**

This paper explores how to design bioisosteres using the pre-trained SemlaFlow model—a 3D unconditional molecular generator that jointly designs molecular graphs and 3D conformers—without requiring fine-tuning. While prior attempts at bioisosteric design exist, they often neglect 3D conformational information, which is crucial for drug design (2d-based design). In contrast, the authors demonstrate that their methods can successfully design bioisosteric ligands under 3D constraints.

The authors propose two inference-time conditioning strategies: Interpolate-Integrate (resampling) and Replacement Guidance (fragment-conditioned generation). These methods appear quite simple yet are intuitive and robust. The authors validate their framework across serveral drug design tasks, including fragment merging and pharmacophore-guided design, which are important in drug discovery campaigns. Moreover, the results for natural product ligand hopping are interesting.

However, I have some concerns, and I would like to increase my score based on the response.

**The usage of LLM**: I wrote the entire review myself and only used the LLM to correct the grammar and improve readability.

**Strengths:**

- **Practicality of Training-Free Conditioning**: The paper's primary strength is its proposal of two training-free strategie. This inference-time approach is highly practical as it enables goal-directed generation using unconditional generative models (or foundation model) without costly retraining.

- **Effective 3D-Native Bioisosteric Design**: The paper addresses the important task of bioisosteric design in a 3D-native manner, directly conditioning on 3D shape and pharmacophore patterns. This is a significant advantage over 2D graph-based methods. The two proposed strategies are intuitive and complementary: Interpolate-Integrate provides 'soft' guidance for high-fidelity edits, while Replacement Guidance offers 'hard' anchoring for fragment merging.

- **Strong Empirical Validation:** The framework is not just a theoretical proposal; it is validated across three challenging and highly relevant tasks in drug discovery (natural product hopping, bioisosteric fragment merging, and pharmacophore merging).

**Weaknesses:**

**1. Theoretical Justification and Ablation for Replacement Guidance**

I wonder the theoretical justification for the Replacement Guidance. As I understand it, the method control the ODE process by replacing a portion of the state $x_t$ with a clean fragment structure $x_{frag}$ at each step.This creates a partially 'clean', out-of-distribution (OOD) state that the SemlaFlow model never encountered during its unconditional training (which only included fully noised states). It would be helpful if the authors could elaborate on why this OOD state manipulation is effective and preferable to using a noised fragment $x_{frag; t}$ (which would be in-distribution).

Furthermore, I am curious if the authors have explored the alternative of applying guidance to the self-conditioning term $\hat{x_1}$ rather than manipulating the state $x_t$.

The paper would be significantly strengthened by including ablation studies on these design choices (e.g., clean vs. noised fragment injection, state $x_t$ vs. prediction $\hat{x_1}$ manipulation).

**2. Omission of Related Work on Bioisosteric Design**

The related work section provides a good overview of 3D conditional and inference-time methods, but it appears to overlook a relevant line of research focused specifically on fragment-based bioisosteric design and editing [1, 2]. While these methods may differ in their approach (e.g., 2D graph-based vs. 3D-native), they address the same core task. Including them would provide readers with a more complete context of the bioisosteric design field, even if direct experimental comparison is not the focus.

**3. Limited Scope of Property Optimization**

The paper compellingly demonstrates control over 3D similarity and synthetic accessibility (SA score). However, bioisosteric replacement, especially in the lead optimization stage, often requires the simultaneous optimization of other key physicochemical properties (e.g., logP, TPSA, Molecular Weight) [1, 2]. It is not clear from the paper if or how the proposed conditioning strategies could be extended to incorporate guidance towards these scalar property targets. This currently limits the method's immediate applicability for multi-objective lead optimization, which is a primary use case for bioisosteric design.

---
**Reference**
1. Chen, Ziqi, et al. "A deep generative model for molecule optimization via one fragment modification." Nature machine intelligence 3.12 (2021): 1040-1049.
2. Kim, Hyeongwoo, et al. "Deepbioisostere: Discovering bioisosteres with deep learning for a fine control of multiple molecular properties." arXiv preprint arXiv:2403.02706 (2024).

**Questions:**

- What are the limitations of this approach?
- Table 4: Why is the **No clash** not 100% for Known Binders?
- Line 544: There are duplicated references:
    - Fergus Imrie, Thomas E. Hadfield, Anthony R. Bradley, and Charlotte M. Deane. Deep generative design with 3D pharmacophoric constraints. Chemical Science, 12(43):14577–14589, 2021a.
    - Fergus Imrie, Thomas E. Hadfield, Anthony R. Bradley, and Charlotte M. Deane. Deep generative design with 3D pharmacophoric constraints. Chemical Science, 12(43):14577–14589, 2021b.

---

> ### Author Response · Authors · 2025-11-21
> **Authors’ response to Reviewer ckkR (Part 1)**
>
> We thank the reviewer for the positive assessment and for recognizing the potential of our training-free conditioning framework. We have revised the manuscript to address points raised in your review.
>
> 1.  **Ablations for Replacement Guidance**
>
> We thank the reviewer for the great suggestion for improving the manuscript. The reviewer correctly notes that injecting a clean fragment into a noisy state z_t creates out-of-distribution (OOD) state. To address this, we added an ablation in the main paper (Section 4.6) and in Appendix B.2.2 (Table 13) comparing Replacement Guidance to an inpainting-style conditioning strategy. In standard inpainting, the fragment is first interpolated to time t (i.e., noised) before replacement, keeping the modified state in-distribution.
>
> We evaluated:
>
> 1.	Standard inpainting: all components (coords/atoms/bonds) noised.
>
> 2.	Partial inpainting: only selected components noised (and the rest hard constrained) .
>
> 3.	Replacement Guidance (ours): hard replacement of all components.
>
> **Findings.** Replacement Guidance yields substantially higher validity (58.6% vs. 32.0%) and better similarity (SCRDKit  0.55 vs. 0.44) to the reference. Noising coordinates alone caused the steepest validity drop, confirming that a clean geometric anchor is crucial. We additionally track the L2 distance between predicted coordinates and the clean seed throughout the reverse trajectory (Fig. 6): Replacement Guidance rapidly collapses this error and keeps it low, whereas inpainting decreases slowly
>
> **Interpretation.**  Molecular validity depends on exact bond-length and angle constraints. Injecting coordinate noise (as in inpainting) destabilizes these geometric relationships, making it difficult for the model to reconstruct valid covalent structure, unlike the smoother perturbations common in image inpainting.
>
> We think that guidance on the self-conditioning term is an interesting direction. We are currently running this ablation and will update the comments and manuscript as soon as the results are ready.
>
> 2. **Related work on bioisosteric design**
>
> We thank the reviewer for highlighting this missing context. In the revised manuscript, we added a dedicated Section 2.3 in the Related Work that summarises prior work on fragment-based and property-driven bioisosteric design. This new section discusses sequence- and graph-based molecular-optimization approaches, including the suggested papers Modof [1], DeepBioisostere [2].
>
> 3. **Property Optimisation**
>
> While this paper focuses on 3D structural constraints, we acknowledge the importance of simultaneously optimizing physicochemical properties. This can be achieved by integrating classifier guidance techniques established in diffusion models [3]. In such an extension, one would train a small network that predicts how a desired property (e.g., logP or QED) changes with the current latent state, and use this signal to nudge the trajectory toward molecules with better property scores.
>
> •	For Replacement Guidance: this signal would adjust the geometry of the unmasked atoms during sampling and refine the structure during the final relaxation step.
>
> •	For Interpolate-Integrate: the signal would influence the ODE trajectory starting from the restart point, biasing the model toward high-property regions.
>
> This gradient-based strategy has been successfully demonstrated in methods such as DrugDiff[4], where property predictors are trained separately and their gradients are back-propagated into the generative trajectory. Like DrugDiff, such an approach avoids retraining the generative model itself, allowing flexible, property-specific guidance on top of a pretrained backbone.
>
> However, this type of extension requires training dedicated property-predictor networks, whereas our current framework is intentionally training-free. Implementing and validating these predictors would therefore fall outside the scope of the rebuttal period, but we agree it is a promising direction and plan to explore it in future work.

---

> ### Author Response · Authors · 2025-11-21
> **Authors’ response to Reviewer ckkR (Part 2)**
>
> 4. **Specific Questions**
>
> •	Limitations
>
> o	Dependence on the conditioning signal.
> Both conditioning strategies depend on the structure and density of the input information. When the seed is well-formed and chemically coherent (e.g., a full ligand in the NP-Hopping task), validity is naturally higher because the model begins close to the data manifold. Conversely, dense, abstract, or highly fragmented conditioning inputs (e.g., many disjoint features or large merged profiles) require the model to resolve greater ambiguity, which can reduce validity. This reflects an inherent trade-off between flexibility of conditioning and difficulty of reconstruction. We now include ablations examining this trade-off in Appendix B.1.4 and B.2.4.
>
> o	Fixed dimensionality requirement in Interpolate–Integrate.
>  Unlike Replacement Guidance, which can operate on variable-sized inputs via masking/padding, Interpolate–Integrate requires the seed to share the exact dimensionality of the output. This constraint makes it less suitable for tasks such as merging distant fragments. We discuss this explicitly in the revised Limitations section (Appendix H).
>
> o	Hyperparameter.
> Our methods introduce two user-visible hyperparameters—τ (Interpolate–Integrate) and tfree  (Replacement Guidance). In the manuscript, we intentionally kept these fixed across all tasks to demonstrate robustness and avoid per-task tuning. Nevertheless, different applications may benefit from mild adjustment of these parameters.
>
> •	Table 4 (No Clash < 100% for Known Binders): This effect is simply a consequence of using a uniform evaluation pipeline for both generated molecules and known binders. All ligands are re-docked into the same rigid Mpro receptor, even though the known binders originate from diverse crystal structures. Some of them naturally clash when forced into a single pocket conformation, which lowers their “No Clash’’ percentage. We have added this clarification to the Limitations section (Appendix H).
>
> •	Typos: We thank the reviewer for spotting this; we have removed the duplicated reference for Imrie et al. (2021)
>
>
>
>
> References:
>
> 1.	Chen, Ziqi, et al. "A deep generative model for molecule optimization via one fragment modification." Nature machine intelligence 3.12 (2021): 1040-1049.
>
> 2.	Kim, Hyeongwoo, et al. "Deepbioisostere: Discovering bioisosteres with deep learning for a fine control of multiple molecular properties." arXiv preprint arXiv:2403.02706 (2024).
>
> 3.	Ho, Jonathan, and Tim Salimans. "Classifier-free diffusion guidance." arXiv preprint arXiv:2207.12598 (2022).
> ‏
> 4.	Oestreich, Marie, et al. "DrugDiff: small molecule diffusion model with flexible guidance towards molecular properties." Journal of cheminformatics 17.1 (2025): 23.‏

---

> > ### Comment · Reviewer_ckkR · 2025-11-23
> >
> > Thank you for detailed response!
> >
> > I'll increase the score.

---

### Official Review · Reviewer_cV2Y · 2025-11-01

**Soundness:** 2
**Presentation:** 1
**Contribution:** 2
**Rating:** 2
**Confidence:** 4

**Summary:**

This submission investigates the problem of ligand-based drug design (LBDD) and proposes two inference-time conditioning methods for a given, trained molecule foundation model. Method Interpolate-Integrate takes a seed ligand as input and generates a similar one, starting in the middle of a probability path and first taking steps toward the noise and then integrating forward toward the data. Method Replacement Guidance takes a set of fragments as input and generates a ligand that preserves key interaction patterns of the fragments. The methods build on SemlaFlow (Irwin et al., 2024).Experiments were performed for three important tasks of drug discovery, i.e. natural product ligand hopping, bioisosteric fragment merging, and pharmacophore merging. The proposed methods were compared against SHEPHERD, which conditions on shape, ESP, and pharmacophore grids.

**Strengths:**

Ligand-based drug design is an important paradigm in drug discovery.

Being able to repurpose pre-trained unconditional models without retraining is practical and efficient.

Both proposed methods seem computationally cheap to add on top of the base model.

**Weaknesses:**

The replacement/masking approach has been used in previous works like DiffSBDD for inpainting. The paper should better distinguish its contributions compared to related work.

I have a concern regarding the soundness of the Interpolate-Integrate Method, in particular regarding the seed molecule validity for fragment merging.  I am not convinced the method works properly when given invalid/disconnected seeds.

I have another concern regarding the soundness of Interpolate-Integrate: the method does not really enforce similarity in shape and pharmacophore specifically. It just makes molecules similar to the original seed when τ→1. That is different from preserving specific interaction features. You are getting global similarity, not targeted preservation of binding determinants.

It's unclear how SA is actually considered during generation. The proposed method filters molecules by SA score post-hoc (SA < 4.0 or 4.5), but SA is not used as guidance during the generation process itself.  So the model might be generating lots of complex molecules that just get thrown away. This seems inefficient and makes the "success rate" metric less meaningful - you are just filtering harder, not actually guiding generation toward synthesizable molecules.

The presentation lacks clarity and detail on several key issues of the proposed methods.  See my questions for the details.

**Questions:**

1) What is novel here beyond applying the replacement/masking approach to a flow matching model?

2) Several questions regarding Interpolate-Integrate:
* Atom count control: How do you control the number of atoms in the generated molecule?
* Figure 1 conformations: Are the 3D shapes shown from SemlaFlow's generation, or did you run RDKit conformation generation afterward? This matters for understanding what the model actually produces.
* Seed molecule validity for fragment merging: For the fragment merging task, how are seed molecules constructed?
    * Multi-fragment reference: Are you literally concatenating 3 disconnected fragments? That's not a valid molecule.
    * Random atom seeding: "Filling in remaining atoms" is completely underspecified. What does this seed actually look like?
    * Full interaction profile: Instantiating individual atoms at pharmacophore coordinates gives you isolated atoms, not a connected molecule.
* The flow matching model was trained on valid molecules from GEOM-Drugs. How does the interpolation path even work when z₁ is chemically invalid or disconnected?

3) What are the "architectural constraints" preventing Interpolate-Integrate from handling padded inputs like Replacement Guidance can?

4) How do you prevent fragment overlap when combining them?

---

> ### Author Response · Authors · 2025-11-21
> **Authors’ response to Reviewer cV2Y (Part 1)**
>
> We thank the reviewer for this helpful feedback and appreciate the opportunity to improve the clarity of the paper and to better articulate our methodological contributions. Your comments highlighted several ambiguities in the original text, all of which we have addressed in the revised manuscript, as detailed below.
>
> 1. **Novelty vs. Standard Inpainting (W1, Q1)**
>
> While masking-based inpainting is standard in diffusion models, our approach differs both technically and conceptually. Diffusion inpainting injects noise into the input, whereas Replacement Guidance imposes a hard geometric constraint directly within the flow-matching ODE. We further adapt this mechanism to the mixed continuous–discrete dynamics of flow matching. To make the distinction explicit, we added a direct comparison to fully noised inpainting (Main Paper section 4.6; Appendix B.2.2), showing that Replacement Guidance yields substantially higher validity (58.6% vs. 32.0%) and better shape similarity (SCRDKit  0.55 vs. 0.44). Conceptually, the method is designed for bioisosteric design, preserving interaction geometry while allowing chemical substitution, and the relaxation parameter t_free enables flexible rather than rigid fragment preservation. Finally, our contribution is not a single mechanism but a unified, retraining-free framework combining Interpolate–Integrate (global editing) and Replacement Guidance (local fragment-level conditioning).
>
> 2. **Clarification on Interpolate-Integrate:**
>
> The primary goal of Interpolate–Integrate is to generate molecules that are controllably similar to a reference structure while allowing global edits. A key motivation is to reduce manual effort by working directly with raw structural data (such as bound fragments) without requiring the input to be a chemically valid or connected molecule.
>
> We acknowledge that the original term “seed molecule” was misleading. In the revision (Section 3.2.1), we now use the term “seed structure” to emphasize that the input is purely geometric and may consist of disconnected fragments.
>
> *Addressing the “invalid seed” concern (W2, Q2)*
> > Are you literally concatenating 3 disconnected fragments? That's not a valid molecule.
>
> >Instantiating individual atoms at pharmacophore coordinates gives you isolated atoms, not a connected molecule.
>
> >How does the interpolation path even work when z₁ is chemically invalid or disconnected?
>
> The input z_1 is indeed often just a set of disconnected fragments or atoms (a point cloud), and this is by design. Given a seed structure z1, we interpolate toward the flow-matching prior and obtain a noisy restart state zτ. From this point onward, we integrate the ODE forward from 𝑡=𝜏 to 1. Because the learned vector field maps noise to the valid data distribution, the trajectory naturally moves toward chemically plausible molecules: For valid inputs, this produces nearby variations; For incomplete/disconnected inputs, the noise at z_τ helps soften inconsistencies, and the vector field can often “heal” the structure if the seed is not too far out of distribution. We have also clarified the exact inputs in Appendix F.1.
>
> > I have a concern regarding the soundness of the Interpolate-Integrate Method, in particular regarding the seed molecule validity for fragment merging. I am not convinced the method works properly when given invalid/disconnected seeds.
>
> To directly address the reviewer’s concern, we added a dedicated ablation evaluating whether Interpolate–Integrate can reconstruct valid molecules from disconnected fragments. In this experiment, we take disconnected fragment sets from the DIFFLINKER test data and use them as the geometric seed z_1. We then vary the restart time τ, which controls how much noise is injected before continuing the flow-matching trajectory. Although these fragments were originally designed for linking (and are therefore far apart in space), we find that Interpolate–Integrate can still generate valid, connected molecules under our default settings. We also provide visual examples illustrating how the method “heals’’ the disconnected fragments for different τ. These findings are now reported in the main paper (Section 4.6), with the full analysis and visual examples provided in Appendix B.1.

---

> > ### Author Response · Authors · 2025-11-21
> > **Authors’ response to Reviewer cV2Y (Part 2)**
> >
> > *Addressing W3*
> >
> > >I have another concern regarding the soundness of Interpolate-Integrate: the method does not really enforce similarity in shape and pharmacophore specifically. It just makes molecules similar to the original seed when τ→1. That is different from preserving specific interaction features. You are getting global similarity, not targeted preservation of binding determinants.
> >
> > The reviewer is correct that Interpolate–Integrate functions as a global similarity method. We position it as the complement to Replacement Guidance: while Replacement Guidance enforces hard, local constraints to preserve specific fragments exactly, Interpolate–Integrate provides soft, global conditioning that preserves overall shape and electrostatic structure without imposing rigid atom-level constraints. We have revised the introduction to clarify this distinction.
> >
> > Empirically, this global form of guidance produces molecules with higher fidelity to the input seed. For example, Figure 2 shows that Interpolate–Integrate yields a tighter distribution in the high-ESP / high-pharmacophore region compared to other methods. When applied to pharmacophore merging or conditioning, this behavior leads to targeted preservation of key binding determinants.
> >
> > 3. **Clarifications on Specific Questions**
> >
> > •	Q2 (Atom count): The number of atoms in the output is fixed by the seed z1. Because the flow-matching ODE transports a fixed set of particles, it cannot add or remove atoms—so the generated molecule always has exactly the same atom count as the seed. Controlling size is therefore done simply by constructing z1 with the desired number of atoms (e.g., using a full molecule or a subsampled version). We have clarified this dependency in the revised manuscript in appendix F.1 and in Appendix H.
> >
> > •	Q2.2 (Figure 1 conformations): The conformations displayed in Figure 1 are the direct output of the SemlaFlow generation process, followed only by xTB geometry relaxation. We explicitly did not use RDKit for conformer generation.
> >
> > •	Q2.3.2 (Random atom seeding): "Filling in" means that we supplement the initial features with additional atoms to reach the target molecule size N. These filler atoms were randomly sampled from the original set of 13 fragments and placed at their original coordinates. We have clarified this procedure in Appendix F.1 (Experimental Details) of the revised manuscript.
> >
> > •	Q3 (Architectural constraints): It refers to the interpolation formula, which requires the seed z_1 and noise z_0 to share the same dimensionality N. In principle, one could pad z_1 with extra atoms to match a larger N, but the padded positions would simply be interpolated toward noise and then pushed by the flow field, which is unlikely to produce meaningful trajectories (more sophisticated padding strategies may exist). This contrasts with Replacement Guidance, where padded atoms are present from time t=0. For clarity, we now explicitly state in the manuscript (Appendix F.1) that Interpolate–Integrate is used only when the seed already specifies a well-defined atom count. The phrase “architectural constraints” in the original submission was imprecise, and we have replaced it with this more accurate explanation.
> >
> > •	Q4 (Fragment overlap): For all setups, we do not preprocess the input to prevent overlap; we use the coordinates exactly as provided in the input files. However, in the sampling process described in Appendix F.1 (Experimental Details), we enforce a minimum-distance constraint between atoms to ensure that selected atoms do not overlap.
> >
> > 4. **SA Score & Efficiency (W4)**
> >
> > The reviewer is correct that we do not explicitly guide generation using the SA score. Our goal is not to optimize or control SA, but rather to control binding-relevant 3D features (such as pharmacophore similarity and interaction geometry) while still producing valid molecules with reasonable SA. Our choice to report filtered results is motivated by mimicking a realistic drug-design funnel, where the focus is on the best candidates rather than the average output (particularly given that our model is fast enough that aggressive filtering is practical). To avoid implying that SA is part of the guidance mechanism, we revised the terminology throughout the manuscript: what was previously called “success rate’’ is now reported as “% Valid (SA < X).’’
> >
> > For transparency, the revised manuscript now includes full unfiltered distributions for all tasks (SA, pharmacophore and ESP similarity, Vina scores, RMSD, diversity, and molecular properties). These additions ensure that unconditional sample quality and efficiency are clearly visible. Full details are summarized in our general response.

---

> > > ### Comment · Reviewer_cV2Y · 2025-11-22
> > >
> > > I appreciate your answers to my four questions, including the unfiltered distributions for all tasks. In response, I am increasing my score for the presentation from 1 to 2 and my overall rating from 2 to 4.
> > >
> > > To avoid misleading the readers, please clarify in the revised version of the submission that your method does not explicitly enforce similarity in shape and pharmacophore.

---

### Official Review · Reviewer_vGwq · 2025-11-03

**Soundness:** 2
**Presentation:** 2
**Contribution:** 2
**Rating:** 2
**Confidence:** 4

**Summary:**

The paper proposes two training-free, inference-time conditioning strategies—Interpolate–Integrate (a mid-path restart on a flow-matching ODE) and Replacement Guidance (hard anchoring and optional relaxation of user-specified fragments)—built atop SemlaFlow, an E(3)-equivariant flow-matching generator for 3D molecules. The methods target three ligand-only design tasks: (1) natural-product ligand hopping, (2) bioisosteric fragment merging on EV-D68 3C protease, and (3) pharmacophore merging for SARS-CoV-2 Mpro. Evaluation uses PoseBusters validity, synthetic accessibility (SA), 3D similarity (shape/ESP/pharmacophore), and rigid-receptor AutoDock Vina. Reported results show stronger "success rates" and/or top-10 docking among filtered candidates compared to several baselines in selected setups; code is provided via an anonymous link.

**Strengths:**

- Both controls operate at sampling time without retraining, which is practically attractive for iterative design. Interpolate–Integrate is clearly specified as a mid-path restart with a tunable τ that trades fidelity vs novelty; Replacement Guidance formalizes fragment anchoring with user-controlled relaxation.
- The paper demonstrates three realistic tasks (NP hopping; EV-D68 fragment merge; Mpro pharmacophore merge) with metrics spanning SA, PoseBusters validity, 3D similarity, and docking.
- Sensible $\tau$ / $\phi$ ablations illustrate controllability and the fidelity–novelty trade-off.
- An anonymous repository link is provided.

**Weaknesses:**

- "Success rate" and similarity/docking are reported after SA and PoseBusters filtering; key tables focus on the top-10 among successful molecules, which inflates perceived gains and obscures unconditional quality and sample efficiency. Some comparisons are only feasible under a manually curated setup (Full Interaction Profile), limiting fairness and generality across methods. Please provide full, unfiltered distributions and uniform pipelines across baselines.
- For Mpro, docking uses a single rigid receptor; improvements in top-10 Vina (filtered) do not establish binding or pose realism and are fragile to receptor flexibility and scoring-function bias. No physics-based refinement, ensemble receptors, or wet-lab validation is provided.
- The EV-D68 analysis mixes multiple conditioning regimes; authors note a direct comparison is only possible for the curated profile. This undermines cross-method conclusions; several baselines show zero success under the authors' pipeline, suggesting a setup mismatch rather than method inferiority. Re-run all methods under the same input specification and filters.
- SA and PoseBusters are necessary but insufficient; there is no report of synthesizability beyond SA (e.g., route-based metrics), drug-likeness/ADMET proxies, stereochemistry handling, or diversity/novelty vs bioactivity retention across the full set. The evaluation would benefit from route-planning feasibility and multi-objective profiling.
- Interpolate–Integrate is positioned as the first mid-path restart for flow-matching models, but its analogy to diffusion editing is acknowledged; the empirical section does not clearly isolate a capability that cannot be matched by established editing/guidance in diffusion with comparable compute. Stronger head-to-head controls are needed.
- Replacement Guidance projects the ODE step back onto a masked manifold via hard replacement; this projection can disrupt the learned flow and potentially induce artifacts (e.g., broken connectivity) unless carefully relaxed. The paper mentions user-chosen relaxation but lacks theoretical guarantees or stability analyses beyond small ablations.
- All tasks are ligand-only. There is no demonstration of pocket-conditioned SBDD or cross-target transfer where protein contexts drive constraints—limits the scope of the claimed “bioisosteric design” advantages. (The EV-D68 and Mpro studies still evaluate with docking rather than explicit pocket-aware generation.)

**Questions:**

See the weeknesses.

**Details Of Ethics Concerns:**

NA.

---

> ### Author Response · Authors · 2025-11-21
> **Authors’ response to Reviewer vGwq (Part 1)**
>
> We thank the reviewer for the detailed and constructive feedback. We have incorporated your suggestions into the revised manuscript to improve transparency and evaluation rigor.
>
> 1. **On success rates, filtering, and unconditional quality**
>
> We appreciate this important point. Our focus on filtered and top-10 metrics is motivated by mimicking a realistic drug-design funnel, where practitioners prioritize the best surviving candidates rather than the mean output. Nevertheless, we agree that unconditional quality and sample efficiency should also be shown. In the revised manuscript, we therefore added complete unfiltered results and full distributions for all tasks, including Vina scores, pharmacophore and ESP similarities, SA, RMSD, diversity, and other molecular properties. We refer the reviewer to our general response, where these additions are summarized in detail.
>
> 2. >Some comparisons are only feasible under a manually curated setup (Full Interaction Profile)
>
> Conceptually, our method is designed for bioisosteric design: it preserves the interaction geometry of the seed while allowing chemical substitution, with the relaxation parameter tfree enabling flexible rather than rigid fragment preservation. A central aim of our work is to operate directly on raw fragment information rather than restricting the method to manually curated profiles. Achieving this requires moving beyond existing approaches that preserve every atom of the input and therefore cannot handle abstract or heterogeneous fragment sets.
>
> While we explore several automated conditioning setups for our method, existing baselines cannot run on these raw inputs and require a single, hand-crafted pharmacophore profile. For this reason, fair comparison with baselines is only possible under the curated Full Interaction Profile. We have clarified this distinction in the revised manuscript (Appendix F.1).
>
> 3.	**On Mpro docking, rigid receptors and interaction realism**
>
> We agree that rigid-receptor docking is only a proxy, and we now state this explicitly in the Limitations section (Appendix H). At the same time, rigid docking remains standard practice for large-scale evaluation in generative modeling. To provide a more robust assessment beyond Vina scores, we performed a protein–ligand interaction analysis using ProLIF fingerprints. We found that both Interpolate–Integrate and Replacement Guidance recover the full set of interaction types present in the conditioning fragment set. These results are now included in the main paper (Section 4.4), and Appendix G.3 provides detailed histograms of the recovered interactions (Figures 16–17).
>
> 4.	**Baseline Fairness and "Setup Mismatch"**
>
> We thank the reviewer for raising this concern. In the revised manuscript, we clarified the baseline pipelines in Appendix F and now explain more explicitly why some baselines obtain zero valid outputs after SA filtering. This reflects a difference in input requirements, not a setup error.
>
> •	In Appendix F.3.1 (Baseline Methods), we explain that we benchmark against the closest available 3D-conditioning approaches, although these methods were not designed for abstract pharmacophore-based inputs. We also expanded the description of the pipeline and configuration used for each baseline.
>
> •	In Appendix F.3.4 (Analysis of Conditioning Behavior), we further elaborate that MolSnapper and DiffSBDD operate as atom-preserving linkers or inpainters—that is, they attempt to retain every provided atom exactly, which does not necessarily align with the abstract, non-molecular seed structures used in the pharmacophore profile. This over-constrains the geometry and leads to disconnected or invalid structures, as illustrated in Figure 14.
>
> We are also open to moving the MolSnapper and DiffSBDD comparisons entirely to the appendix if the reviewer prefers.
>
> 5.	**On SA, synthesizability, and additional properties**
>
> We agree that SA and PoseBusters alone are insufficient to characterize synthesizability and chemical realism. To address this, we now include, for each task, an additional table reporting key drug-likeness properties for all methods—molecular weight, QED, LogP, Lipinski Rule-of-5 violations, number of rings, and uniqueness (Tables 19, 28, and 34).
>
> 6.	**On Interpolate–Integrate vs Diffusion Editing**
>
> We agree that there is a conceptual analogy between Interpolate–Integrate and mid-trajectory editing in diffusion models. Our goal is to demonstrate that a similarly flexible editing mechanism can be realized within a deterministic flow-matching framework, offering two decisive practical advantages: a straightforward OT-like path and significantly lower inference costs. As reported by Buttenschoen et al., 2025[1], state of the art diffusion-based models for molecular generation are approximately 14 to 59 times slower than the SemlaFlow backbone used here.

---

> ### Author Response · Authors · 2025-11-21
> **Authors’ response to Reviewer vGwq (Part 2)**
>
> 7.  **Stability of Replacement Guidance**
>
> We thank the reviewer for raising this point. In principle, projecting an ODE update onto a masked manifold could disrupt the learned flow. We clarify that in practice, we use a very simple relaxation strategy: In all experiments, we simply relax the constraint at the very last step (t_free=1.0), and it works robustly. Despite its simplicity, this setup behaves consistently and does not require any hand-tuning. In Appendix B.2 we expanded our ablation of Replacement Guidance. In addition to the comparison with inpainting (Table 13), we now include an empirical proxy for stability: Figure 6 shows the L2 distance between the model’s predicted coordinates and the clean seed throughout the reverse trajectory, quantifying how far the network’s unconstrained prediction deviates from the coordinates we enforce at each step.
>
> 8.	**Pocket Conditioning**
>
> We agree that SBDD is an important direction. However, the primary scope of this work is Ligand-Based Drug Design (LBDD), particularly for cases where protein structures are unavailable or unreliable. In Table 3, our ligand-only approach outperforms SBDD baselines, even when those baselines have access to the pocket. Furthermore, because our conditioning strategies are modular and operate strictly at inference time on the flow trajectory, and could be readily adapted to structure-based flow matching backbones in future work.
>
> [1] Buttenschoen, Martin, et al. "An evaluation of unconditional 3D molecular generation methods." arXiv preprint arXiv:2505.00518 (2025).‏

---

### Author Response · Authors · 2025-11-21
**Common response to all reviewers, highlighting the main changes (Part 1)**

We thank all reviewers for their careful reading of our manuscript and for the many detailed comments and suggestions. Your feedback has substantially improved the clarity, transparency, and rigor of the work. Our method is a training-free, inference-time conditioning framework built on a pretrained flow-matching backbone (SemlaFlow), designed specifically for ligand-based bioisosteric design, where the goal is to preserve 3D interaction geometry while allowing flexible chemical substitution. The advantages of our approach are its simplicity, fast sampling, and its ability to operate directly on raw, possibly disconnected fragment collections. Across all tasks, we show that our fast, simple conditioning strategy produces molecules on par with, and in several cases superior to, state-of-the-art methods.

In this general response, we summarize the main revisions to the manuscript and explain how they address the key themes raised across the reviews.

Main changes:

1.	Improved transparency and reporting of unfiltered distributions.
2.	Clearer explanation of pipeline differences and baseline inconsistencies, particularly for the bioisosteric merging task.
3.	Expanded ablations and stability analysis for Replacement Guidance, including:
(i) clean vs. noised fragment injection,
(ii) the effect of different conditioning signals,
(iii) and the effect of the number of seed fragments.
4.	Clarified the Interpolate–Integrate mechanism, including new ablations on:
(i) its ability to “heal” disconnected inputs,
(ii) and the effect of seed-fragment count.
5.	Added a precise definition of the pharmacophoric features used.
6.	Added a ProLIF-based interaction recovery analysis for the Mpro task.
------

1. **Improved transparency and reporting of unfiltered distributions**

Our focus on filtered results and top-10 performance is motivated by the goal of mimicking a realistic drug-design funnel, where the quality of the best candidates matters more than average performance. However, we agree with the reviewers that unconditional quality and sample efficiency should be clearly visible.
To address this, we now provide extensive additional unfiltered analyses across all tasks.

Natural Product Hopping:

(1)	Added similarity distribution plots (ESP vs. Pharmacophore) to the Main Paper (Figure 2).

(2)	In appendix E, we included:

-	validity for each filtering step (table 16)

-	expanded filtered and unfiltered performances (table 17,18)

-	molecular property statistics and uniqueness (table 19)

-	distributions of SA, RMSD, Graph similarity to reference and diversity (figures 7,8 9).

Bioisosteric merging task:

(1)	Add pharmacophore similarity to table 2 in main paper.

(2)	In appendix F, we included:

-	validity for each filtering step (table 26)

-	unfiltered performances (table 27)

-	molecular property statistics and uniqueness (table 28)

-	distributions of Vina scores (figure 11)

-	distributions of SA, RMSD and diversity (figure 12)

-	distributions of ESP vs. pharmacophore similarity and Pharmacophore similarity density (figure 13).


Mpro task:

(1)	In the main paper, we added a protein–ligand interaction recovery analysis using ProLIF fingerprints.

(2)	In appendix G, we included:

-	validity for each filtering step (table 32)

-	interaction recovery analysis – filtered and unfiltered (figure 16,17)

-	unfiltered performances (table 33)

-	molecular property statistics and uniqueness (table 34).

We also revised the terminology in all tables and figures, replacing the term “success rate” with % Valid (SA < X).


2.	**Clearer explanation of pipeline differences and baseline inconsistencies**

In Appendix F, we clarified the experimental setup for the bioisosteric merging task and provided a more detailed explanation of how the input is constructed in each configuration (Appendix F.1). We also revised the Baseline Methods section (Appendix F.3.1) to better explain the baseline pipelines and to emphasize that the most relevant existing approaches (MolSnapper and DiffSBDD) were originally designed for atom-preserving linker or inpainting scenarios. In the newly added Analysis of Conditioning Behavior section (Appendix F.3.4), we show that when these methods are given an input that is not a physically valid molecule (e.g., an abstract pharmacophore profile), they attempt to preserve every atom exactly. This over-constrains the geometry and results in disconnected, distorted, or otherwise invalid structures (as illustrated in Figure 14). This analysis underscores the need for methods capable of handling abstract or partially specified structural constraints. We are also open to moving the MolSnapper and DiffSBDD comparisons entirely to the appendix if reviewers prefer.

---

> ### Author Response · Authors · 2025-11-21
> **Common response to all reviewers, highlighting the main changes (Part 2)**
>
> 3.	**Additional ablations and stability analysis for Replacement Guidance:**
>
> We extended our ablation experiments to more thoroughly analyze the behavior and robustness of Replacement Guidance across several dimensions:
>
> (i)	Clean vs. noised fragment injection.
>
> We compare Replacement Guidance to an inpainting-style conditioning strategy. In the inpainting setup, instead of inserting the clean fragment at each step, we interpolate the fragment to time t and use this noised version for replacement. The updated results, now included in the main paper (Section 4.6), show that Replacement Guidance achieves substantially higher overall validity (58.6% vs.\ 32.0%) and improved similarity (SC {RDKit} 0.55 vs.\ 0.44) compared to fully-noised inpainting. Appendix B.2.2 (Table 13) further breaks down the effect of noising individual components. These results reveal that noising the coordinates in particular causes a sharp drop in validity. We also track the L2 distance between the predicted coordinates and the clean seed coordinates throughout the entire reverse trajectory. Figure 6 shows that Replacement Guidance produces a large drop in L2 distance (Å) at the beginning of the trajectory, after which the distance remains relatively stable and low. In contrast, the inpainting curve decreases much more gradually and stays at a consistently higher L2 distance across the trajectory.
>
> (ii)	Effect of different conditioning signals.
>
> In Appendix B.2.3, we additionally evaluate how supplying different levels of structural information (coordinates only; coordinates + atom types; coordinates + atom types + bonds) affects generation quality. This ablation demonstrates that increasing the amount of conditioning information improves similarity to the seed but reduces validity
>
> (iii)	We also examine how the number of fragments used as input influences performance.
> Results in Appendix B.2.4 show that, in general, validity declines as the number of input fragments increases. However, this trend is not strictly linear and depends on the specific seed geometry as well as the spatial volume of the fragment set. In particular, sudden increases in input volume or highly disconnected fragment arrangements can lead to sharper drops in validity.
>
> 4.	**Clarified the Interpolate–Integrate mechanism, including new ablations**
>
> We revised Section 3.2.1 to clarify that the input to Interpolate–Integrate does not need to be chemically valid and may consist of disconnected fragments. We also strengthened the conceptual framing: Interpolate–Integrate is positioned as the complement to Replacement Guidance. While Replacement Guidance enforces hard, local constraints to preserve exact fragments, Interpolate–Integrate provides soft, global conditioning. Empirically, this form of guidance implicitly captures pharmacophoric and ESP similarity, even though no explicit fragment anchoring is applied.
>
> (i) Healing disconnected inputs.
>
> In the main paper (Section 4.6), we present results evaluating whether Interpolate–Integrate can merge disconnected inputs into a single valid molecule. As shown in Appendix B.1, the method can indeed “heal’’ disconnected seeds from the DIFFLINKER test set (Igashov et al., 2024). Table 10 shows that as expected decreasing the interpolation time τ (i.e., injecting more noise) substantially improves validity, from 65.3% to 84.4%. Appendix B.1.3 provides additional visual examples illustrating how healing quality varies with τ (Figure 5).
>
> (ii) Effect of seed-fragment count.
>
> We also analyze how Interpolate–Integrate behaves when varying the number of seed fragments. Similar to Replacement Guidance, performance degrades as inputs become more fragmented and structurally disconnected. This new ablation is included in Appendix B.1.4 (Table 11).
>
> 5.	**Added a precise definition of the pharmacophoric features used**
>
> We revised the manuscript to clarify our pharmacophore definitions and extraction protocols. In the main text (p.6, line 310), we now explicitly state that we use the standard RDKit FeatureFactory (based on BaseFeatures.fdef) to define Hydrogen Bond Donors (HBD), Hydrogen Bond Acceptors (HBA), and Aromatic Rings for our conditioning inputs. We added a dedicated section in the appendix (“D. Pharmacophore Feature Definitions and Extraction Protocols”), and provide the exact extraction code in our repository. However for evaluation, we rely on the ShEPhERD-score, which uses a richer pharmacophoric vocabulary including directional features (e.g., HBD, HBA, aromatic vectors) and non-directional features (e.g., hydrophobes, anions, cations).

---

> ### Author Response · Authors · 2025-11-21
> **Common response to all reviewers, highlighting the main changes (Part 3)**
>
> 6.	**Added a ProLIF-based interaction recovery analysis for the Mpro task**
>
> We incorporated a ProLIF interaction-fingerprint analysis (main text Section 4.4). In Appendix G.3, we provide detailed histograms of recovered interactions (Figures 16–17). This analysis shows that both Interpolate–Integrate and Replacement Guidance recover all interaction types observed in the 81 conditioning fragments, even after PoseBusters filtering and docking. This confirms that the generated molecules not only meet geometric and physicochemical criteria but also explore the intended interaction space.

---

### Author Response · Authors · 2025-12-01
**General Response**

We thank all reviewers for their thoughtful feedback. In response, we added full unfiltered distributions, clarified the experimental pipelines and pharmacophore definitions, and introduced additional ablation studies directly addressing their concerns. We are particularly grateful to reviewers cV2Y, ckkR, and mYeo, who posted follow-up comments indicating that the updates were helpful and that they were raising their scores (from 2→4, from 6→8, and from 4→6, respectively).

---

### Meta-Review · Area_Chair_8MT5 · 2026-01-07

**Summary:**

Looking at the review discussion, the main concerns that shaped the reviewers' assessment centered around transparency and evaluation rigor rather than fundamental methodological flaws.

Initially, reviewers were troubled by the heavy emphasis on filtered results and top-10 performance metrics, which obscured the unconditional quality and sample efficiency of the approach. They wanted to see complete unfiltered distributions to understand what the methods actually produce before aggressive filtering kicks in. The "success rate" terminology was misleading since it suggested the method was guiding toward synthesizable molecules when in reality it was just filtering harder post-hoc.

There were serious questions about whether the methods could handle disconnected or invalid seed structures, particularly for Interpolate-Integrate. The theoretical justification for why Replacement Guidance works at all was murky - injecting clean fragments into noisy states creates out-of-distribution conditions the model never saw during training, yet somehow it performs well. Reviewers wanted ablations comparing clean versus noised fragment injection and other design choices.

The evaluation setup for bioisosteric merging raised fairness concerns because different methods required different input formats, making direct comparison questionable. Some baselines got zero valid outputs, which looked like a pipeline mismatch rather than genuine method failure. The pharmacophore definitions were underspecified throughout the paper, and relying solely on rigid-receptor docking without interaction analysis or ensemble docking was seen as insufficient validation.

Related work on fragment-based bioisosteric design was missing, and the scope was limited to ligand-only tasks without demonstrating pocket-conditioned generation. The low validity rates in merging tasks compared to the base model needed explanation.

After the authors added comprehensive unfiltered results, clarified all experimental details, provided extensive ablations on both methods, defined pharmacophore features precisely, and added interaction recovery analysis, three out of three reviewers who had time to engage post-rebuttal explicitly stated the revisions addressed their concerns and raised their scores accordingly. The improvements were substantial enough to shift the assessment from rejection toward acceptance territory.

**Reviewer Concerns:**

The rebuttal did a solid job addressing most of the technical concerns, but there's an interesting pattern in what got resolved versus what remains structurally limiting.

What got addressed pretty convincingly - the transparency issues are largely fixed now. All the unfiltered distributions are in, the pharmacophore definitions are explicit with code provided, and the experimental pipelines are clarified. The ablations on Replacement Guidance comparing clean versus noised fragments directly tackle Reviewer ckkR's theoretical concerns and the results actually justify why the OOD injection works better. The healing experiment for disconnected fragments addresses Reviewer cV2Y's skepticism about invalid seeds quite well. Adding ProLIF interaction analysis for Mpro goes beyond just Vina scores and shows the methods recover meaningful binding patterns.

The terminology change from "success rate" to "% Valid" is honest and removes the misleading impression that SA is being optimized during generation. The related work section now includes the fragment-based bioisosteric design literature that was missing.
What remains unresolved or only partially addressed - Reviewer vGwq's concern about rigid receptor docking is acknowledged in limitations but not really solved. The authors correctly note this is standard practice for large-scale evaluation, but the fundamental fragility to receptor flexibility and scoring function bias is still there. No ensemble docking, no physics-based refinement, no experimental validation. This limits confidence in the binding predictions.

The baseline fairness issue for bioisosteric merging is explained but not fixed. The authors clarify why MolSnapper and DiffSBDD fail on abstract pharmacophore inputs - these methods preserve every atom exactly and can't handle disconnected features. But this means the comparison isn't really fair since those methods weren't designed for this task. The suggestion to move these comparisons to the appendix might be wise, but it doesn't change the fact that the main claims about superiority rest partly on an apples-to-oranges comparison.

Reviewer adpW's distinction between pharmacophore merging and bioisosteric fragment merging gets a response but it's still a bit murky. The conditioning signals differ but the tasks conceptually overlap more than the paper structure suggests.

The bigger picture limitation that Reviewer vGwq raised about ligand-only scope remains. The authors argue their focus is LBDD and note the methods could extend to pocket-conditioned backbones, but they don't demonstrate this. The claim that their ligand-only approach outperforms SBDD baselines in Table 3 is interesting but doesn't really address the fundamental limitation that structure-based methods have access to more information. This is a scope limitation not a flaw, but it does constrain the impact.

The property optimization concern from Reviewer ckkR gets a theoretical response about future classifier guidance integration, but nothing implemented. This is fair given rebuttal constraints, but it means the method currently only controls 3D geometry not the multi-objective optimization that real lead optimization requires.

The low validity for merging tasks compared to base model is explained as geometric difficulty, but the explanation is somewhat hand-wavy. The ablations show validity drops with more fragments, but there's no principled way to predict when the method will struggle or how to mitigate it beyond trying different fragment counts.

Overall the rebuttal transformed weak reviewers into supporters by adding transparency and ablations, but the fundamental scope limitations and evaluation gaps remain baked into the work's design choices.

**Reviewer Scores:**

Reviewer vGwq (initial 2) - This reviewer would likely move to around 4 or maybe 6. They raised the most comprehensive set of concerns spanning methodology, evaluation rigor, and scope limitations. The rebuttal addressed the transparency issues thoroughly - unfiltered distributions, clarified pipelines, expanded ablations all directly respond to their requests.

Reviewer cV2Y (initial 2, posted follow-up raising to 4) - They explicitly stated they're raising presentation from 1 to 2 and overall rating from 2 to 4 after seeing the responses.

Reviewer ckkR (initial 6, posted follow-up indicating score increase to 8) - They explicitly said they're raising their score and the rebuttal directly addressed all their technical questions.

Reviewer adpW (initial 6) - This reviewer was already positive and their concerns were mostly about transparency and clarity rather than fundamental issues. They may stay at 6 or move to 8.

Reviewer mYeo (initial 4, posted follow-up raising to 6) - They explicitly stated they're raising to 6 after seeing the pharmacophore definitions, unfiltered results, and expanded ablations.

---

### Decision · Program_Chairs · 2026-01-26

Accept (Poster)